# How to Scale Your EMA

**Dan Busbridge**[*]    **Jason Ramapuram**[*]    **Pierre Ablin**[*]    **Tatiana Likhomanenko**[*]

**Eeshan Gunesh Dhekane**        **Xavier Suau**        **Russ Webb**

Apple

```
{dbusbridge, jramapuram, p_ablin, antares,
   eeshan, xsuaucuadros, rwebb}@apple.com
```

## Abstract

Preserving training dynamics across batch sizes is an important tool for practical machine learning as it enables the trade-off between batch size and wall-clock time. This trade-off is typically enabled by a scaling rule, for example, in stochastic gradient descent, one should scale the learning rate linearly with the batch size. Another important machine learning tool is the model EMA, a functional copy of a target model, whose parameters move towards those of its target model according to an Exponential Moving Average (EMA) at a rate parameterized by a momentum hyperparameter. This model EMA can improve the robustness and generalization of supervised learning, stabilize pseudo-labeling, and provide a learning signal for Self-Supervised Learning (SSL). Prior works have not considered the optimization of the model EMA when performing scaling, leading to different training dynamics across batch sizes and lower model performance. In this work, we provide a scaling rule for optimization in the presence of a model EMA and demonstrate the rule's validity across a range of architectures, optimizers, and data modalities. We also show the rule's validity where the model EMA contributes to the optimization of the target model, enabling us to train EMA-based pseudo-labeling and SSL methods at small and large batch sizes. For SSL, we enable training of BYOL up to batch size 24,576 without sacrificing performance, a 6× wall-clock time reduction under idealized hardware settings.

## 1   Introduction

With data and models becoming progressively larger (Chen et al., 2020; Kaplan et al., 2020; Bommasani et al., 2021; Srivastava et al., 2022), the ability to reduce training wall–clock time is a requirement for practical Machine Learning (ML) at scale. Optimizer scaling rules allow us to find faster learning procedures that produce similar results. For example, the *linear scaling rule* for Stochastic Gradient Descent (SGD) (Krizhevsky, 2014; Goyal et al., 2017), states that the learning rate should be scaled linearly with the batch size. This optimizer scaling works *both ways*. Access to larger computational resources means one can train equivalent models in reduced wall-clock time. Alternatively, with access to limited computational resources, larger distributed computations can be replicated at increased wall-clock time.

---

[*]Primary contributor. For a detailed breakdown of author contributions see Appendix J.

37th Conference on Neural Information Processing Systems (NeurIPS 2023).

Many ML algorithms rely on a *model EMA*, a functional copy of a *target model*[2], whose parameters move towards those of its target model according to an Exponential Moving Average (EMA) (Definition 1.1) at a rate parameterized by a momentum hyperparameter $\rho$.

**Definition 1.1** (EMA Update). *The EMA update for the model EMA parameters $\zeta_t$ following target model parameters $\theta_t$ at iteration t with momentum $\rho \equiv 1 - \beta_\rho$ is*

$$\zeta_{t+1} = \rho\,\zeta_t + (1 - \rho)\,\theta_t \equiv (1 - \beta_\rho)\,\zeta_t + \beta_\rho\,\theta_t. \tag{1}$$

The *model EMA* has a number of desirable properties: i) the model EMA inhabits wider minima than the target model, reducing overfitting and improving generalization (Ruppert, 1988; Polyak & Juditsky, 1992; Huang et al., 2017; Izmailov et al., 2018; He et al., 2022); ii) compared to the target model, the model EMA moves slowly, making it useful as a stabilizer for networks governing Bellman updates in reinforcement learning, (Lillicrap et al., 2016); and iii) the model EMA is relatively cheap to compute, whilst providing a valid model but *different* to the target model. This third property has made the model EMA a common choice for the *teacher* in many distillation setups, from semi-supervised learning (Tarvainen & Valpola, 2017; Sohn et al., 2020; Manohar et al., 2021; Higuchi et al., 2022), to Self-Supervised Learning (SSL) methods like Bootstrap Your Own Latent (BYOL) (Grill et al., 2020), DINO (Caron et al., 2021), and data2vec (Baevski et al., 2022b,a).

Despite its significant role in optimization, a recipe for adapting the EMA Update (Definition 1.1) when changing batch size has, to the best of our knowledge, been absent. To address this, we derive an EMA Scaling Rule (Definition 1.2) which states how the EMA momentum $\rho$ hyperparameter *should* be modified[3].

**Definition 1.2** (EMA Scaling Rule). *When computing the EMA update (Definition 1.1) of a model undergoing stochastic optimization with batch size $\hat{B} = \kappa B$, use a momentum $\hat{\rho} = \rho^\kappa$ and scale other optimizers according to their own scaling rules.*

In Definition 1.2, the momentum $\rho$, which is defined at batch size $B$, typically corresponds to a "good hyperparameter choice", although this does not need to be the case in general. In this paper, we make the following contributions.

1. With the assumptions of Goyal et al. (2017), we derive an EMA Scaling Rule: the EMA update *momentum* should be scaled *exponentially* with the batch size (Definition 1.2).

2. To validate this EMA Scaling Rule theoretically, we propose Stochastic Differential Equation (SDE) approximations of optimization in the presence of a model EMA (Section 2.2). This model EMA contributes to the loss, covering semi-supervised learning and SSL. We prove that these approximations are first order weak approximations, and that our EMA Scaling Rule is correct in the SDE limit under realistic gradient assumptions (Corollary 2.1.1).

3. We empirically validate the EMA Scaling Rule in synthetic settings (Section 3.1) and real-world settings where the model EMA plays an increasingly significant role in optimization: i) where the model EMA is used during inference instead of the target model (Section 3.2); ii) pseudo-labeling, where the model EMA (*teacher*) follows the target model (*student*), and the *student* is optimized on a mixture of a) labeled data and b) data without labels, whose pseudo-labels are produced by the *teacher* (Section 3.3); and iii) self-supervised learning, which is the same as the semi-supervised case, except there is no labeled data (Section 3.4).

4. We observe that pseudo-labeling and SSL training dynamics during optimizer warm-up are not always able to be replicated at large batch sizes using *only* the EMA Scaling Rule. We propose and verify practical methods to overcome this limitation, enabling us to scale to a batch size of 24,576 with BYOL Vision Transformers (ViTs), reducing wall-clock training by 6× under idealized hardware scenarios while maintaining performance of the batch size 4096 baseline.

Finally, to aid practitioners looking to scale, in Appendix C we provide a *Scaling Toolbox*, which gives practical advice on how to scale systematically, collecting known scaling rules, and explaining how to think about the SDE perspective of optimization.

---

[2] The target model usually undergoes gradient-based optimization, but this does not have to be the case.

[3] We stress that the study of momentum in gradient-based optimizers is not the focus of this work. We refer to Smith & Le (2018); Li et al. (2019) for a discussion on scaling rules for these methods.

# 2 The EMA Scaling Rule

We begin with an informal discussion of scaling rules and motivate the existence of an exponential scaling rule for the momentum parameter controlling the update of the model EMA.

## 2.1 Background and an informal discussion of scaling rules

Consider a model with parameters $\theta_t$ at iteration $t$ updated with SGD (Definition 2.1).

**Definition 2.1** (SGD Update). *The SGD update for a model with parameters $\theta_t$ at iteration $t$ given a minibatch $\mathbb{B} = \{x^{(b)} \sim P_\mathbf{x} : b = 1, 2, \ldots, B\}$ of $B = |\mathbb{B}|$ samples with learning rate $\eta$ is*

$$\theta_{t+1} = \theta_t - \eta \times \frac{1}{B} \sum_{x \in \mathbb{B}} \nabla_\theta \mathcal{L}(x; \theta_t), \tag{2}$$

*where $\mathcal{L}$ is the loss function, $\nabla_\theta \mathcal{L}(x; \theta_t)$ is the parameter gradient for the sample $x$ at iteration $t$, and the $x \in \mathbb{B}$ are Independent and Identically Distributed (i.i.d.) from $P_\mathbf{x}$.*

Iterating over a sequence of independent minibatches $\mathbb{B}_0, \mathbb{B}_1, \ldots, \mathbb{B}_{\kappa-1}$ produces model parameters

$$\theta_{t+\kappa} = \theta_t - \eta \times \frac{1}{B} \sum_{j=0}^{\kappa-1} \sum_{x \in \mathbb{B}_j} \nabla_\theta \mathcal{L}(x; \theta_{t+j}). \tag{3}$$

If gradients vary slowly $\nabla_\theta \mathcal{L}(x; \theta_{t+j}) \approx \nabla_\theta \mathcal{L}(x; \theta_t)$, $j = 0, \ldots, \kappa - 1$, *one* SGD step with $\hat{\eta} = \kappa \eta$ on a batch $\widehat{\mathbb{B}} = \cup_i \mathbb{B}_i$ of size $\hat{B} = \kappa B$ results in $\hat{\theta}_{t+1} \approx \theta_{t+\kappa}$, yielding the SGD Scaling Rule (Definition 2.2).

**Definition 2.2** (SGD Scaling Rule). *When running SGD (Definition 2.1) with batch size $\hat{B} = \kappa B$, use a learning rate $\hat{\eta} = \kappa \eta$ (Krizhevsky, 2014; Goyal et al., 2017).*

For clarity in this work, we adopt the naming convention *[Algorithm Name] Scaling Rule*, which means all parameters of those algorithms are appropriately scaled from batch size $B$ to $\kappa B$.

As discussed in Goyal et al. (2017), although the assumption of slowly changing gradients is strong, if it is true, then $\theta_{t+k} \approx \hat{\theta}_{t+1}$ *only* if $\hat{\eta} = \kappa \eta$. The validity of the SGD Scaling Rule has been formally studied. In particular, there was ambiguity regarding whether the scaling should be a square-root or linear (Krizhevsky, 2014). SDE approaches have resolved this ambiguity, and have been used to estimate the scaling $\kappa$ when the SGD Scaling Rule is no longer guaranteed to hold (Li et al., 2021).

To address model parameter EMAs, we first restate the EMA Update (Definition 1.1).

**Definition 1.1** (EMA Update). *The EMA update for the model EMA parameters $\zeta_t$ following target model parameters $\theta_t$ at iteration $t$ with momentum $\rho \equiv 1 - \beta_\rho$ is*

$$\zeta_{t+1} = \rho\, \zeta_t + (1 - \rho)\, \theta_t \equiv (1 - \beta_\rho)\, \zeta_t + \beta_\rho\, \theta_t. \tag{1}$$

The model EMA parameters $\zeta$ do not typically receive gradient information, we take the convention that $\rho$ is close to one, and the $\beta_\rho$ subscript will be omitted where it is clear from the context.

Assuming again that gradients change slowly $\nabla_\theta \mathcal{L}(x; \theta_{t+j}, \zeta_{t+j}) \approx \nabla_\theta \mathcal{L}(x; \theta_t, \zeta_t) \approx g$, for gradient $g$, iterating over $\kappa$ independent minibatches produces model states (see Appendix E.1 for derivation)

$$\begin{bmatrix} \theta_{t+\kappa} \\ \zeta_{t+\kappa} \\ g \end{bmatrix} = \begin{bmatrix} 1 & 0 & -\eta \\ (1-\rho) & \rho & 0 \\ 0 & 0 & 1 \end{bmatrix}^\kappa \cdot \begin{bmatrix} \theta_t \\ \zeta_t \\ g \end{bmatrix} = \begin{bmatrix} \theta_t - \eta\,\kappa\,g \\ \rho^\kappa\,\zeta_t + (1 - \rho^\kappa)\,\theta_t + O\left(\eta \times \beta_\rho\right) \\ g \end{bmatrix}. \tag{4}$$

The first row is the SGD Scaling Rule (Definition 2.2). The third row implements the *slowly changing gradients* assumption for the first row. The second row is equivalent to a single EMA update (Definition 1.1) with momentum $\hat{\rho} = \rho^\kappa$; we can take a *single* SGD update with batch size $\hat{B} = \kappa B$ and learning rate $\hat{\eta} = \kappa \eta$, and a *single* EMA update with momentum $\hat{\rho} = \rho^\kappa$, and we get $(\hat{\theta}_{t+1}, \hat{\zeta}_{t+1}) \approx (\theta_{t+\kappa}, \zeta_{t+\kappa})$ up to terms $O(\eta \times \beta_\rho)$. This yields the EMA Scaling Rule (Definition 1.2).

**Definition 1.2** (EMA Scaling Rule). *When computing the EMA update (Definition 1.1) of a model undergoing stochastic optimization with batch size $\hat{B} = \kappa B$, use a momentum $\hat{\rho} = \rho^\kappa$ and scale other optimizers according to their own scaling rules.*

The EMA Scaling Rule was derived for SGD, and is extended to other optimizers in the following way. An optimizer scaling rule ensures $\hat{\theta}_{t+1} = \theta_{t+\kappa}$, satisfying identification for the first row. Next, the zeroth order term in $\eta \times \beta_\rho$ in the second row in Equation 4 is optimizer-independent, and therefore unchanged. Finally, the first order terms in $\eta \times \beta_\rho$ in the second row, corresponding to the scaling rule error, are an EMA accumulation of target model $\theta$ updates under optimization, which is still $O(\eta \times \beta_\rho)$, although its functional form may be different for different optimizers.

The above discussion is intended to give an intuition for why the EMA momentum should be scaled exponentially. As we have used the same slow-moving gradient assumption as the original SGD Scaling Rule, this may cast doubt on whether our rule is correct. To remove this ambiguity, we will follow Smith & Le (2018); Li et al. (2021); Malladi et al. (2022), and show that the EMA Scaling Rule (Definition 1.2) is correct in the SDE limit under more realistic gradient assumptions.

## 2.2 The EMA Scaling Rule through the lens of stochastic differential equations

SDEs are a tool typically used to obtain scaling rules from first principles (Li et al., 2021; Malladi et al., 2022). In the following, we use SDEs to obtain strong theoretical guarantees for the EMA Scaling Rule found in Section 2.1. We consider the following discrete dynamics for EMA:

$$\begin{aligned}
\theta_{k+1} &= \theta_k - \eta\, g_k, \quad \text{with } g_k = \nabla f(\theta_k, \zeta_k) + \sigma\, \epsilon_k, \text{ and } \epsilon_k \sim \mathcal{E}_\sigma(\theta_k, \zeta_k), \\
\zeta_{k+1} &= \rho\, \zeta_k + (1 - \rho)\, \theta_k,
\end{aligned} \tag{5}$$

where $\sigma > 0$ is the noise scale, $\mathcal{E}_\sigma(\theta_k, \zeta_k)$ is the gradient noise distribution, assumed to be zero-mean and variance $\Sigma(\theta_k, \zeta_k)$ independent of $\sigma$, and $\nabla f(\theta_k, \zeta_k) \equiv \nabla_\theta f(\theta_k, \zeta_k)$. We posit a dependency of the loss $f$ on the EMA $\zeta$ in order to cover semi-supervised (Section 3.3) and SSL (Section 3.4). The case of Polyak-Ruppert averaging (Section 3.2), is covered by letting $f$ be independent of $\zeta$.

We aim to obtain an SDE approximation of Equation 5 as $\eta$ goes to zero. The scaling rule for iterations of $\theta$ is well known (Li et al., 2021): we let $\sigma_0 = \sigma\sqrt{\eta}$. The analysis of Section 2.1 gives the scaling rule $\hat{\eta} = \eta\kappa$ and $\hat{\rho} = \rho^\kappa$. Linearizing this rule near $\eta = 0$ gives $\hat{\rho} = 1 - \kappa \times (1 - \rho)$, which is a linear relationship between $1 - \rho$ and $\eta$. We therefore let $\beta_0 = (1 - \rho)/\eta$ and consider the SDE

$$\begin{aligned}
d\Theta_t &= -\nabla f(\Theta_t, Z_t)\, dt + \sigma_0\, \Sigma(\Theta_t, Z_t)^{\frac{1}{2}}\, dW_t, \quad \text{with } W_t \text{ a Wiener process,} \\
dZ_t &= \beta_0(\Theta_t - Z_t)dt,
\end{aligned} \tag{6}$$

where $\Theta_t$ and $Z_t$ are SDE variables relating to model and EMA parameters respectively. The SDE in Equation 6 approximates the discrete iterations of Equation 5 when the learning rate $\eta$ goes to zero. One way to see this is that an Euler-Maruyama discretization of the SDE with learning rate $\eta$ exactly recovers the discrete iterations. More formally, we have Theorem 2.1, which is in the same spirit as those found in Li et al. (2021); Malladi et al. (2022). In the theorem, $G^\alpha$ is the set of functions with derivatives up to order $\alpha$ that have at most polynomial growth (see Definition D.1).

**Theorem 2.1** (SDE for SGD + EMA; informal see Theorem D.1). *Assume that $f$ is continuously differentiable, with $f \in G^3$. Let $\Theta_t, Z_t$ be solutions of Equation 6, and $\theta_k, \zeta_k$ iterations of Equation 5 with $\Sigma^{\frac{1}{2}} \in G^2$. Then, for any time horizon $T > 0$ and function $g \in G^2$, there exists a constant $c > 0$ independent of $\eta$ such that*

$$\max_{k=0,\dots,\lfloor T/\eta \rfloor} |\mathbb{E}[g(\Theta_{\eta k}, Z_{\eta k})] - \mathbb{E}[g(\theta_k, \zeta_k)]| \le c \times \eta. \tag{7}$$

Theorem 2.1 formalizes the intuition that the SDE is an accurate approximation of the discrete iterations. In turn, it allows validating the scaling rule in the same spirit as in Malladi et al. (2022).

**Corollary 2.1.1** (Validity of the EMA Scaling Rule). *Assume that $f$ is continuously differentiable, with $f \in G^3$ and $\Sigma^{\frac{1}{2}} \in G^2$. Let $\theta_k^{(B)}, \zeta_k^{(B)}$ be iterations of Equation 5 with batch size $B$ and hyperparameters $\eta, \rho$. Let $\theta_k^{(\kappa B)}, \zeta_k^{(\kappa B)}$ be iterates with batch size $\kappa B$, and $\hat{\eta}$ determined by the SGD Scaling Rule (Definition 2.2) and $\hat{\rho}$ determined by the EMA Scaling Rule (Definition 1.2). Then, for any time horizon $T > 0$ and function $g \in G^2$, there exists a constant $c > 0$ independent of $\eta$ such that*

$$\max_{k=0,\dots,\lfloor T/\eta \rfloor} |\mathbb{E}[g(\theta_{\lfloor k/\kappa \rfloor}^{(\kappa B)}, \zeta_{\lfloor k/\kappa \rfloor}^{(\kappa B)})] - \mathbb{E}[g(\theta_k^{(B)}, \zeta_k^{(B)})]| \le c \times \eta. \tag{8}$$

Table 1: The role of the model EMA $\zeta$ in the optimization of $(\theta, \zeta)$ given a target model $\theta$ for different techniques, ordered by increasing influence of the EMA model. All statements assume a momentum $0 \leq \rho < 1$ and that the target model $\theta$ is subject to stochastic optimization at a batch size $B$.

| TECHNIQUE | ROLE OF MODEL EMA |
| --- | --- |
| POLYAK-RUPPERT AVERAGING, SEC. 3.2 | $\theta$ undergoes optimization and is tracked by $\zeta$, which does not affect $\theta$. $\zeta$ is an estimate of $\theta$ with a time horizon and variance determined by $B$ and $\rho$. |
| CONTINUOUS PSEUDO-LABELING, SEC. 3.3 | *Pre-Training* is as above in Polyak-Ruppert Averaging. *After Pre-Training*, $\zeta$ (*teacher*) produces targets for $\theta$ (*student*) from unlabeled data, which is combined with labeled data. The optimization endpoint is dependent on $B$ and $\rho$. |
| SELF-SUPERVISED LEARNING, SEC. 3.4 | As above in *After Pre-Training*, except there is no labeled data. The optimization endpoint is dependent on $B$ and $\rho$. |

Corollary 2.1.1 shows that two trajectories with different batch sizes are close in the limit of small learning rate, demonstrating the validity of Definition 1.2. A natural follow-up question is *what happens when an adaptive optimizer is used instead of SGD?* Malladi et al. (2022) study this without an EMA and characterize how hyperparameters change with the noise scale. In particular, they show that under a high gradient noise hypothesis, there exists a limiting SDE. In Appendix D, we derive the limiting SDEs for RMSProp and Adam with an EMA. Although a formal proof of closeness between the iterations and these SDEs is beyond the scope of this work, these SDEs indicate that the EMA Scaling Rule holds for adaptive algorithms. We demonstrate this empirically in Section 3.

## 3 Experiments

Now that we have derived and shown the validity of the EMA Scaling Rule, we verify it empirically. The experiments validate the EMA Scaling Rule for a variety of uses of EMA and are ordered by increasing influence of the role of EMA on the optimization procedure (see Table 1). The baseline in all of our experiments is *without the EMA Scaling Rule*, which applies all known relevant scaling rules *except* the EMA Scaling Rule, and represents previous best practice.

### 3.1 Polyak-Ruppert averaging in a simple setting

At inference, it is typical to use a model EMA, known as Polyak-Ruppert Averaging (Definition 3.1).

**Definition 3.1** (Polyak-Ruppert Average). *When optimizing model parameters $\theta$, compute their EMA $\zeta$ (Definition 1.1). Use $\zeta$ instead of $\theta$ at inference* (Polyak & Juditsky, 1992; Ruppert, 1988).

We begin by showing the EMA Scaling Rule is *required* to match parameter trajectories in a simple setting. Consider the optimization of $\theta$ in a *noisy parabola* whose loss $\mathcal{L}(\theta)$ is parameterized by coefficients for curvature $a > 0$, scaled additive noise $b \geq 0$, and additive noise $c \geq 0$:

$$\mathcal{L}(\theta) = \frac{a}{2}\,\theta^2, \qquad \theta_{k+1} = \theta_k - \eta\,g_k, \qquad g_k = a\,\theta_k + \epsilon_k, \qquad \epsilon_k \sim \mathcal{N}\left(0, \frac{b\,g_k^2 + c}{\kappa}\right). \qquad (9)$$

The scaling factor $\kappa$ in the covariance denominator implements gradient noise reduction as scaling (i.e. batch size) increases (Jastrzebski et al., 2017). Let $\theta \in \mathbb{R}$ be optimized with SGD (Definition 2.1) and $\zeta \in \mathbb{R}$ be a Polyak-Ruppert average (Definition 3.1) for $\theta$ with momentum $\rho = 1 - \beta$. At scaling $\kappa = 1$, we use $\beta_B = \eta_B = 10^{-4}$ and $I_B = 10^4$ iterations, to yield a total time $T = I_B \times \eta_B = 1$. To keep gradients $O(1)$ and gradient noise non-negligible, we take $a = 1$, $b = 0.5$, and $c = 0$.

First, we observe the effect of scaling on a single run (Figure 1a) by tracking the position of the model EMA. We see that at scaling $\kappa = 8$ or $\kappa = 256$, the runs using the EMA Scaling Rule match the baseline trajectory, whereas the runs using the baseline momentum do not, with a greater deviation induced by greater scaling $\kappa$. Even at $\kappa = 8$, there is a significant difference between scaled and unscaled trajectories, despite the seemingly small numerical difference of their momenta[4].

Second, we consider whether the EMA Scaling Rule is optimal. To do this, inspired by the SDE analysis (Section 2.2), we define the approximation error, $\mathrm{Err}(\rho, \kappa, g)$, of a test function $g$ for a given

---

[4]Momentum enters optimization exponentially; small changes can lead to very different updates.

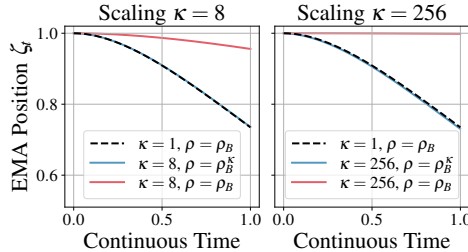
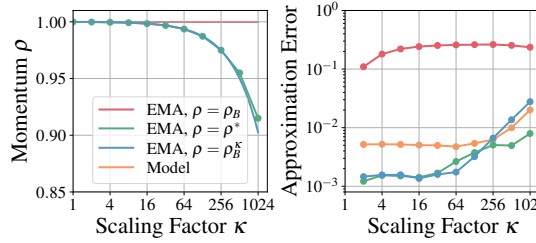

(a) Trajectory of the model EMA $\zeta$ under different scalings $\kappa$, with $1 - \rho_B = \eta_B = 10^{-4}$.

(b) Choices for momentum (left) with corresponding approximation errors (Equation 10) (right).

Figure 1: (a) We show the effect of scaling by comparing model EMA trajectories of the baseline ($\kappa = 1$, black dashed) to $\kappa = 8$ (left) and $\kappa = 256$ (right), with ($\rho = \rho_B^\kappa$, blue) and without ($\rho = \rho_B$, red) the EMA Scaling Rule. (b, left) The momentum according for different scaling rules and the empirically optimal $\rho^*$ (Equation 10). (b, right) The approximation error (Equation 10) of trajectories in (b, left) and the target model (orange).

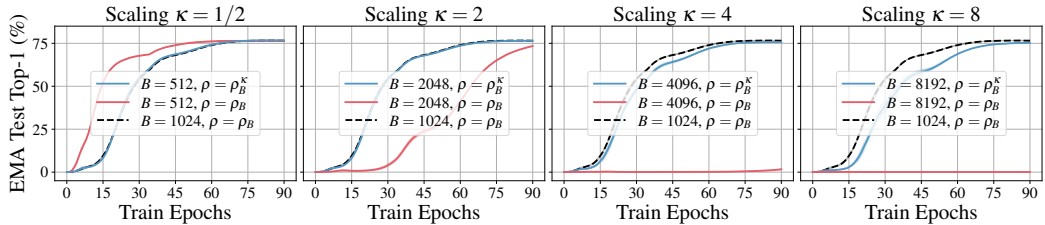

Figure 2: *ResNetv2-50 Polyak-Ruppert averaging on ImageNet1k* for different scalings $\kappa$. The baseline model ($\kappa = 1$, black dashed) uses batch size 1024 and momentum $\rho_B = 0.9999$, is scaled down to a batch size of 512 (left), and up to a batch size of 4096 (right) with (blue, $\rho = \rho_B^\kappa$) and without (red, $\rho = \rho_B$) the EMA Scaling Rule (Definition 1.2). Bands indicate the mean and standard deviation across three runs.

scaling $\kappa$ using momentum $\rho$, and the value of the momentum $\rho^*(\kappa, g)$ that minimizes this error:

$$\rho^*(\kappa, g) = \arg\min_\rho \text{Err}(\rho, \kappa, g), \qquad \text{Err}(\rho, \kappa, g) \equiv \max_{k=0,\ldots,T/\eta} \left| \mathbb{E}\, g(\zeta_k) - \mathbb{E}\, g(\zeta_{k/\kappa}^{(\kappa,\rho)}) \right|. \qquad (10)$$

For scalings $\kappa \in \{1, 2, 4, \ldots, 1024\}$, we determine the optimal momentum $\rho^*$ and compare it to the EMA Scaling Rule (Figure 1b, left). The scaling rule tracks the $\rho^*$ until $\kappa = 256$, when the $\rho^*$ become systematically higher. We see target model error increase at $\kappa = 256$ (Figure 1b, right). As the target model error is EMA-independent, this indicates that the SGD Scaling Rule is breaking. At the lower scaling $\kappa = 64$, there is an inflection point in the EMA Scaling Rule approximation error, before the model error grows. This difference indicates the $O(\eta \times \beta_\rho)$ terms of Equation 4 are beginning to influence the EMA update. Finally, these observations are true in $D = 100$ dimensions, (Appendix F.1), and we stress that *not* changing the momentum at every scaling $\kappa$ induces large approximation error, indicating there is merit to using the EMA Scaling Rule.

### 3.2 Supervised learning on real data with Polyak-Ruppert averaging

We now turn to real-world classification where the target model $\theta$ optimizes a parametric log-likelihood $\max_\theta \log p(\mathbf{y}|\mathbf{x}; \theta)$ with inputs and labels $(\mathbf{x}, \mathbf{y})$ drawn from a joint distribution $p(\mathbf{y}, \mathbf{x})$.

**Image Classification** We consider a variant of the original SGD Scaling Rule result (Goyal et al., 2017) and train a ResNetv2 (He et al., 2016b) on ImageNet1k (Russakovsky et al., 2014) (Figure 2) using a three step learning rate schedule. The base momentum $\rho_B = 0.9999$ at batch size 1024 was found by hyperparameter optimizing for EMA test performance, and we seek to achieve this optimized performance at different batch sizes. We *do not* apply the EMA Scaling Rule on the Batch Normalization (Ioffe & Szegedy, 2015) statistics[5]. We observe that *without* the EMA Scaling Rule, there is a significant drop in model EMA test performance, whereas *with* the EMA Scaling Rule, we

---

[5]Since Batch Normalization statistics use an EMA update, it is reasonable to ask whether the EMA Scaling Rule should be applied. We investigate this in Appendix F.3. We find one *should* apply the scaling rule, however, the effect is less significant than the application of the EMA Scaling Rule to model parameters.

can approximate the baseline model EMA test top-1 performance across all batch sizes. We match baseline EMA statistics across the full trajectory batch size 2048, where the test EMA performance diverges. This is due to non-EMA test performance dropping for high $\kappa$ (see Appendix F.2). We observe that model EMA top-1 is approximately 0.2% to 0.3% higher than the target model.

**Automatic Speech Recognition (ASR)**   We train a transformer (Vaswani et al., 2017) using the Connectionist Temporal Classification (CTC) loss (Graves et al., 2006) and Adam optimizer on the *train-clean-100* subset (100h) of LibriSpeech (Panayotov et al., 2015) (for details see Appendix G). We apply the Adam Scaling Rule (Malladi et al. (2022), Definition C.3) and use dynamic batching (minibatch size × sequence length = const = 290$s$, and $s$ indicates audio duration in seconds).

*Without* the EMA Scaling Rule, there is a significant difference in model EMA test Word Error Rate (WER) trajectories compared to the baseline, whereas *with* the EMA Scaling Rule, trajectories match, as is shown in Figure 3. We note that compared to image classification, in ASR, the model EMA converges to similar final performance irrespective of use of the scaling rule. This convergence is due to the longer training time compared to the EMA horizon as discussed in Table 1 (see Appendix E.2 for a proof sketch). Although in this specific case one can achieve similar *final performance* without the EMA Scaling Rule, it is *necessary* to use the EMA Scaling Rule in order to replicate the full training trajectory, which gives *guarantees* on properties like final performance (see Corollary 2.1.1). We also observe a growing gap between the baseline and EMA-scaled trajectories as we increase $\kappa$. Inspecting the train loss and non-EMA test WER, which *do not* depend on the EMA update (see Figure 14, Appendix G.1), indicates this is due to a breakdown of the Adam Scaling Rule. *In summary, evaluation on ASR shows that the EMA Scaling Rule holds in practice for sequential data with dynamic batch sizes, as well as when using adaptive optimization.*

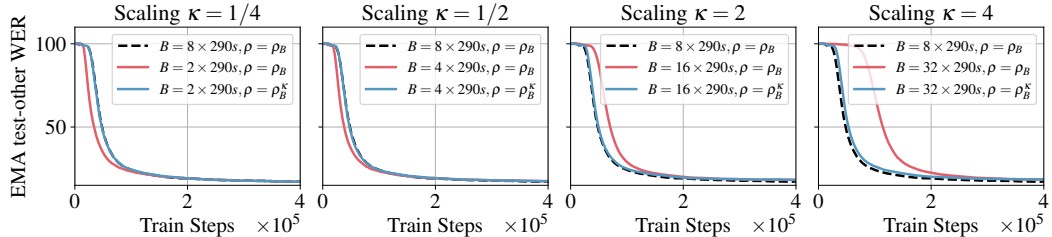

Figure 3: *Transformer Polyak-Ruppert averaging on LibriSpeech (trained on train-clean-100) with different scalings $\kappa$. The baseline ($\kappa = 1$, black dashed) is trained with Adam and momentum $\rho_B = 0.99995$ at a dynamic batch size $B = 8 \times 290s$, which corresponds to a single train step on the x-axis. We investigate dynamic batch sizes down to $B = 2 \times 290s$ (left) and up to $B = 32 \times 290s$ (right), with (blue, $\rho = \rho_B^\kappa$), and without (red, $\rho = \rho_B$) the EMA Scaling Rule. The Adam Scaling Rule (Malladi et al. (2022), Definition C.3) is used throughout.*

### 3.3  Semi-supervised speech recognition via pseudo-labeling

We continue using the same ASR model and training pipeline of Section 3.2. However, we consider semi-supervised learning via continuous pseudo-labeling where labeled (*train-clean-100*, 100h) and unlabeled (the rest of LibriSpeech, 860h) data are given during training, and the model EMA is involved in the overall optimization (Likhomanenko et al., 2021a, 2022; Manohar et al., 2021; Higuchi et al., 2022). We first pre-train a target model (*student*) on a limited labeled set for a short period (e.g. 20k steps of $B = 8 \times 290s$[6]). Concurrently, the student updates a model EMA (*teacher*). After pre-training, we continue training the student with both labeled and unlabeled data, with the teacher first transcribing unlabeled data from the batch producing Pseudo-Labels (PLs). These PLs are treated by the student as ground-truth transcriptions, and standard supervised optimization is performed.

Compared to Polyak-Ruppert Averaging (Section 3.2), where the model EMA plays no role in the joint optimization, we observe that in PL it is *essential* to employ the EMA Scaling Rule in order to match the model trajectories at scaled batch sizes. When the EMA Scaling Rule is not used, Figure 4 reveals a significant difference in PL quality trajectory, leading to a higher test WER.

For $\kappa > 2$, we found the Adam Scaling Rule does not perfectly match the reference trajectory in the pre-training phase. This results in a significantly different PL quality at the start of pseudo-labeling (20k steps of $B = 8 \times 290s$), which affects the training dynamics (Berrebbi et al., 2023). To

---

[6]Note that number of steps is batch size dependent and should be scaled by $1/\kappa$ (see Appendix C).

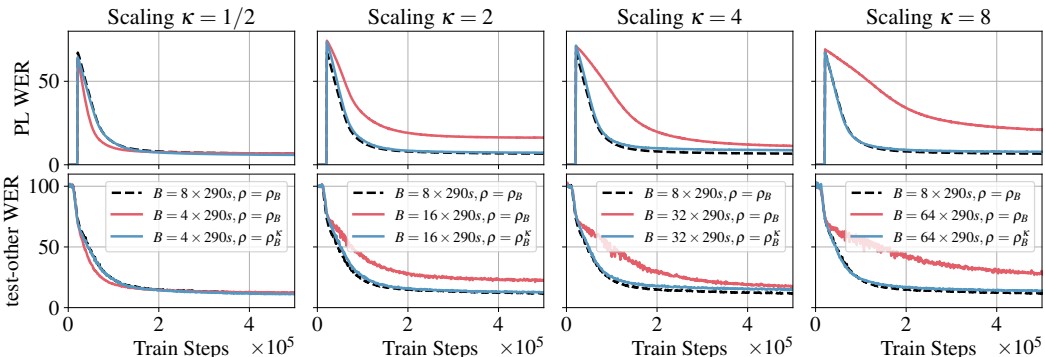

Figure 4: *Transformer pseudo-labeling on LibriSpeech with different scalings $\kappa$. The baseline ($\kappa = 1$, black dashed) is trained with Adam at a dynamic batch size of $8\times290$ seconds, which corresponds to a single train step on the $x$-axis. The model EMA (teacher) is updated with momentum $\rho_B = 0.9999$. We investigate dynamic batch sizes down to $B = 4 \times 290s$ (left) and up to $B = 64 \times 290s$ (right), with (blue, $\rho = \rho_B^\kappa$) and without (red, $\rho = \rho_B$) the EMA Scaling Rule. The Adam Scaling Rule (Malladi et al. (2022), Definition C.3) is used throughout. For $\kappa \leq 2$, we start pseudo-labeling after $20k/\kappa$ training steps; while for $\kappa > 2$, we start when pre-training WER matches the baseline WER.*

alleviate the Adam Scaling Rule mismatch effect for $\kappa > 2$, we postpone the pseudo-labeling until pre-training on labeled data gives similar validation WER, see Appendix G. With this heuristic, we can match the baseline trajectory with the EMA Scaling Rule up to $\kappa = 8$ (Figure 4).

*In summary, (a) model EMA affects the optimization process of pseudo-labeling in ASR resulting in the necessity of EMA Scaling Rule to be applied while scaling the batch size; (b) an optimizer scaling rule breakdown results in the EMA Scaling Rule breakdown but this effect can be alleviated by longer pre-training on labeled data having similar PLs quality at the start across different scalings.*

### 3.4 Self-supervised image representation learning

Finally, we turn our attention to distillation based Self-Supervised Learning (SSL). where the model EMA is the *teacher* (Grill et al., 2020; Niizumi et al., 2023; Caron et al., 2021; Oquab et al., 2023).

We will use BYOL (Grill et al. (2020), Definition 1.1)[7] for our investigation into scaling as it is well-studied (Tian et al., 2021; Richemond et al., 2023), relatively simple to implement due to minimal hyper-parameters, and obtains competitive results (Grill et al., 2020; Koppula et al., 2022). Since BYOL learns through self-referential distillation, momentum plays a significant role in optimization. We analyze: i) a ResNet-18 (He et al., 2016a) on CIFAR10 (Krizhevsky et al., 2014) (Figure 5) using SGD (Definition 2.1); and ii) a ViT-B/16 (Dosovitskiy et al., 2021) on ImageNet1k using AdamW (Loshchilov & Hutter, 2019). A recipe for BYOL using ViTs is provided in Appendix H.3.

**ResNet-18 on CIFAR-10** We begin with a ResNet-18 model and short training duration to enable quick iteration, and an SGD optimizer as it has as *known* scaling rule. This allows us to probe the EMA Scaling Rule without potential confounders like poor gradient-based optimizer scaling[8].

We observe that *without* the EMA Scaling Rule, there is a drop in test top-1 linear probe (Definition H.3) performance compared to the baseline, whereas *with* the EMA Scaling Rule, we closely match the baseline model until batch size 4096. We show that this result is consistent for a range of base learning rates $\eta_B$ and momenta $\rho_B$ in Appendix H.8. At batch size 8192, we see a performance gap between the scaled model using the EMA Scaling Rule and the baseline. We speculate that this is due to dynamics early in the BYOL training process that are challenging to replicate at larger batch sizes. To test, and potentially circumvent this, we introduce *Progressive Scaling* (Definition 3.2).

**Definition 3.2** (Progressive Scaling, informal; see Appendix C.4). *Given batch size B and hyperparameters at B, slowly increase the batch size to the desired largest batch size during training. At any intermediate batch size $\hat{B} = \kappa B$, all hyperparameters are scaled according to their scaling rules.*

---

[7]The BYOL EMA update (Equation 74) uses $\theta_{t+1}$ instead of our analyzed $\theta_t$ (Equation 4). The effect upon the overall EMA update is $O(\eta \times \beta_\rho)$ and so is captured by the EMA Scaling Rule (Definition 1.2).

[8]For competitive performance with the reference BYOL (Grill et al., 2020) using a ResNet-50, adaptive optimization, and longer training duration, see Appendix H.10 and Figure 26.

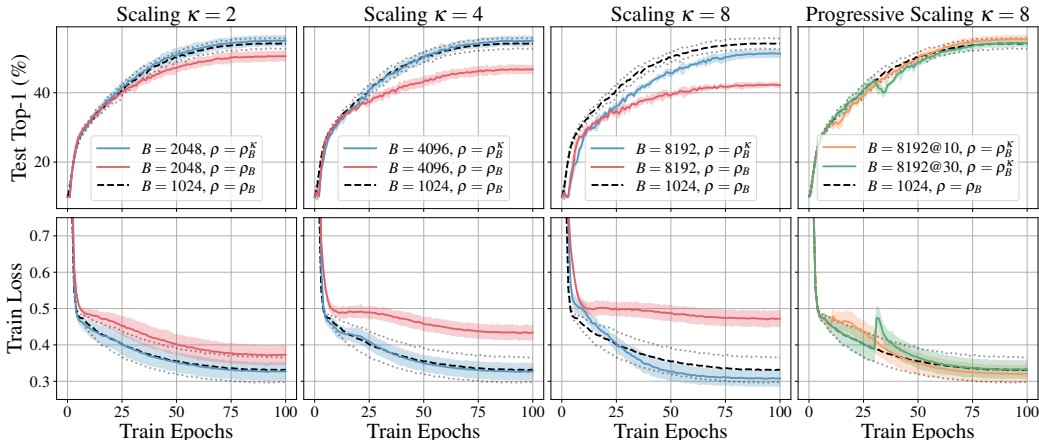

Figure 5: *ResNet-18 BYOL on CIFAR10* for different $\kappa$. The baseline ($\kappa = 1$, black dashed) uses batch size 1024 and momentum $\rho_B = 0.992$, and is scaled from batch size 2048 (left) to 8192 (third) with (blue, $\rho = \rho_B^\kappa$) and without (red, $\rho = \rho_B$) the EMA Scaling Rule. At $\kappa = 8$ we also run *progressive scaling* (right), with transitions at 10 (green) and 30 (orange) epochs. Bands indicate mean and standard deviation across three runs.

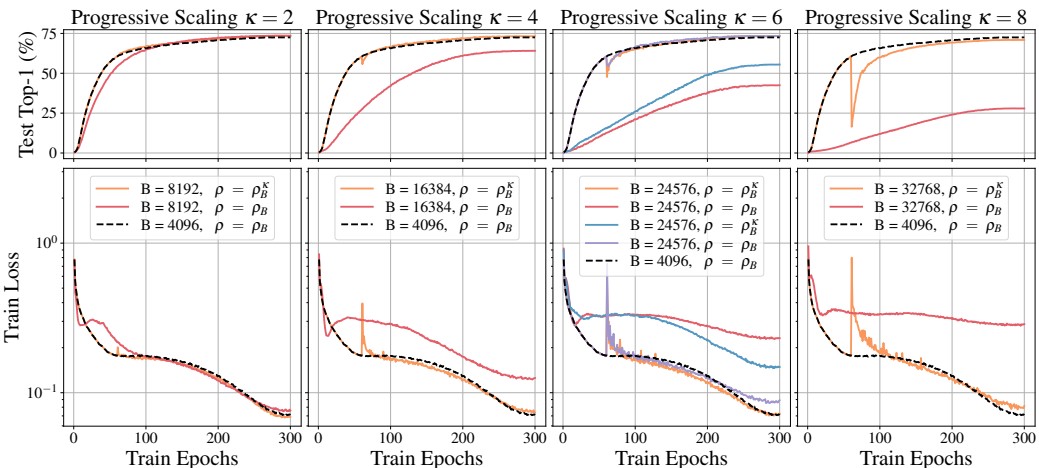

Figure 6: *BYOL ViT-B/16 on ImageNet1k* for different scalings $\kappa$. The baseline model ($\kappa = 1$, black dashed) uses batch size 4096 and teacher momentum $\rho_B = 0.99$, and is scaled from batch size 8192 (left) to 32768 (right) with progressive scaling and the EMA Scaling Rule (Definition 3.2) (orange, $\rho = \rho_B^\kappa$), with the EMA Scaling Rule but without progressive scaling (blue, $\rho = \rho_B^\kappa$), without the EMA Scaling Rule but with progressive scaling (purple, $\rho = \rho_B$), and without either (red, $\rho = \rho_B$). Progressive scaling transitions from the reference model at epoch 60. See Appendix H.6 for a discussion on BYOL progressive scaling.

We see that transitioning to the higher batch size *during* the warmup period results in a model optimization trajectory that diverges from the baseline, whereas transitioning *after* warmup results in matching final trajectories of the scaled and baseline models. In summary, *progressive scaling* allows us to match BYOL dynamics at large batch sizes, provided we transition after the warmup period. This observation is consistent with our hypothesis regarding BYOL dynamics during warmup.

**Vision Transformers on ImageNet1k** Progressive Scaling coupled with the EMA Scaling Rule is required when scaling BYOL ViTs (Figure 6), enabling baseline loss tracking to a batch size of 24,576. Perfect scaling fails at batch size 32,768, consistent with observations in supervised learning (Goyal et al., 2017; Huo et al., 2021). Despite the breakdown, there is only a small drop in 1.6% probe performance when using the EMA Scaling Rule, compared to as 44.56% drop *without* it. We also observe that it is sometimes possible to match test model performance using *only* Progressive Scaling and *not* the EMA Scaling Rule, although this still induces a training loss mismatch. We stress that such an approach is *not* guaranteed to work and discuss when this approach succeeds and fails in Appendix H.6 and Figure 22.

At the transition point between batch sizes, an impulse perturbation[9] is measured at the student, visible from the training loss. This is recovered from by the learning process, and the new model matches the reference batch size. This perturbation happens in both the AdamW and SGD settings, leading us to suspect this is due to the BYOL learning process, rather than an artifact of optimizer or momentum scaling. However, since this is not directly related to the EMA Scaling Rule proposed in this work, we defer this analysis to future investigation.

## 4  Related work

**Optimizer scaling rules from SDEs**  The SDE perspective has uncovered optimizer scaling rules and allowed an understanding of their limitations. Smith & Le (2018) used SDEs to uncover the SGD Scaling Rule, while (Li et al., 2021) used SDEs to explain that rule's breakdown in terms of discretization error. The SDE analysis was extended to adaptive optimization by (Malladi et al., 2022), producing an Adam Scaling Rule (Definition C.3), indicating that along with the learning rate, the $\beta_{1,2}$ and $\epsilon$ parameters transform. The $\beta_{1,2}$ transformation is consistent with the EMA Scaling Rule in the SDE limit. Our work differs as it considers a model EMA that alters the objective.

**Varying the batch size during training**  Smith et al. (2018) investigated the benefits of scheduling the batch size at a fixed learning rate as an alternative to scheduling the learning rate at a fixed batch size. These two are equivalent through the SGD Scaling Rule. The authors *do not* scale the optimizer hyperparameters during this procedure, as they are intentionally replicating the training dynamics of a learning rate schedule. This is in contrast with *Progressive Scaling* (Definition 3.2) which scales the hyperparameters to *maintain* the optimization process at different levels of discretization.

**Large batch training of SSL distillation methods**  SSL methods learn representations without labels, meaning they can take advantage of web-scale data. Large batch optimization is required to make use of this data in a reasonable amount of time. Grill et al. (2020) demonstrated algorithmic robustness when *reducing* the batch size through gradient accumulation and EMA update skipping, which implements an approximation of our EMA Scaling Rule for $\kappa < 1$. Our work provides a recipe to scale down *and up* in $\kappa$. MoCo-v3 (Chen et al., 2021) enables contrastively distilled ViTs up to a batch size of 6144, where the model drops in performance. More recently, methods like DINO (Caron et al., 2020) present a worse scenario, and are unable to scale beyond batch size 1024 (Koppula et al., 2022). In contrast, our work presents practical tools to scale to large batch sizes in the presence of an EMA, enabling practical training of these SSL methods on large scale data.

## 5  Conclusion

We provide an EMA Scaling Rule: when changing the batch size by a factor of $\kappa$, exponentiate the momentum of the EMA update to the power of $\kappa$. This scaling rule should be applied in addition to optimizer scaling rules (for example, linearly scaling the SGD learning rate), and enables the scaling of methods which rely on EMA and are sensitive to the choice of EMA momentum.

We prove the validity of the EMA Scaling Rule by deriving first-order SDE approximations of discrete model optimization when a model EMA is present and can contribute to the model objective. We demonstrate empirical support for a variety of uses of EMA, ordered by increasing influence of the role of EMA on the optimization procedure: supervised model tracking (i.e. Polyak-Ruppert averaging) in speech and vision domains, pseudo-labeling in speech, and self-supervised image representation learning. In almost all scenarios, using the EMA Scaling Rule enables matching of training dynamics under batch size modification, whereas not using it results in significant differences in optimization trajectories. For example, we can scale the BYOL self-supervised method to a batch size of 24,576 without any performance loss *only* when using the EMA Scaling Rule.

While learning rate scaling rules are relatively commonplace in ML, the role of EMA has been overlooked. With this work, we highlight the importance of scaling the EMA momentum, and hope that future works will use the EMA Scaling Rule to scale the EMA momentum correctly, in the same way that learning rates and other optimizer hyperparameters are scaled.

---

[9]Instead of a single large batch transition as in Figure 6 we perform a sequential transition in Appendix H.5. We find that a slow increase in batch size minimizes the magnitude of the perturbation and leads to a final model with higher effective linear probe top-1 than the reference by approximately 1.17%.

# 6 Acknowledgements

We thank Miguel Sarabia del Castillo, Adam Golinski, Pau Rodriguez Lopez, Skyler Seto, Amitis Shidani, Barry Theobald, Vimal Thilak, Floris Weers, Luca Zappella, and Shaungfei Zhai for their helpful feedback and critical discussions throughout the process of writing this paper; Okan Akalin, Hassan Babaie, Denise Hui, Mubarak Seyed Ibrahim, Li Li, Cindy Liu, Rajat Phull, Evan Samanas, Guillaume Seguin, and the wider Apple infrastructure team for assistance with developing and running scalable, fault tolerant code; and Kaifeng Lyu and Abhishek Panigrahi for discussion and details regarding scaling rules for adaptive optimizers. Names are in alphabetical order by last name within group.

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

# Appendices

# A  Broader impact

This work shows how to adapt Machine Learning (ML) optimization in the presence of a model Exponential Moving Average (EMA). There are a number of benefits to this:

1. Scaling rules democratize the training of ML models: they give ML researchers the ability to replicate the optimization of large scale systems, even if those researchers *do not* have access to i) significant parallel computational resources *or* ii) the technical tooling to do so.

2. Our EMA Scaling Rule lowers compute usage as it removes the necessity for a hyperparameter search over momenta; in the case where our scaling assumptions hold, if we know the value of the optimal momentum $\rho_B$ at some batch size $B$ (for example, the momentum that gives the best transfer performance), then the optimal value at another batch size $\hat{B}$ is exactly the one given by the EMA Scaling Rule $\hat{\rho} = \rho_B^\kappa$, for scaling $\kappa = \hat{B}/B$.

3. Our EMA Scaling Rule enables researchers to more quickly iterate through experimental ideas, and opens up access to large-scale training (for example, larger models and larger datasets) for Pseudo-Labeling and Self-Supervised Learning (SSL) techniques.

These points have potential negative consequences:

1. As our EMA Scaling Rule enables researchers to iterate the same experiments more quickly, and perform large-scale training with EMA-based methods, this may encourage a greater number of experiments, or the training of larger models. Either of these possibilities leads to greater energy consumption.

2. As the need to determine momentum hyperparameters has now been removed, researchers who were previously discouraged from attempting to scale these methods due to an *extra* hyperparameter to tune may begin to perform such experiments, leading, once more, to greater energy consumption.

The environmental impact of each of these two points may be significant.

# B  Limitations

The EMA Scaling Rule provides a recipe for producing training dynamics independent of the batch size used in stochastic optimization. The technology underpinning it will not *always* give the desired behavior, however.

The first issue occurs with the wording present in the EMA Scaling Rule: *[...] and scale other optimizers according to their own scaling rules* (Definition 1.2):

1. This statement requires that the given Stochastic Differential Equation (SDE) approximation we are using for the model optimizer is itself providing well-behaved scaling, that is, that in the *absence* of a model EMA, the model optimization trajectories at the batch sizes $B$ and $\kappa B$, with optimizer hyperparameters appropriately scaled, are close. In general we know this is not true. First, we know that the SDE approximation for Stochastic Gradient Descent (SGD) breaks at a given $\kappa$ due to discretization error (Li et al., 2021). Second, we know that if the gradient noise is not sufficiently large, the SDE approximation for Adam does not exist (Malladi et al., 2022), i.e. an SDE motivated scaling rule has no meaning.

2. This statement requires knowledge of how to scale the corresponding model optimizer. We have principled ways to achieve this for SGD (Li et al., 2021), and for the adaptive optimization methods RMSProp and Adam (Malladi et al., 2022). Empirically, a square-root scaling law for LAMB (You et al., 2020) has been observed, however, it has not been derived formally. Problematically, there is no known hyperparameter scaling law or SDE approximation known for LARS (You et al., 2017), which has been used in Bootstrap Your Own Latent (BYOL) (Grill et al., 2020) and many other large-scale training procedures for convolution-based architectures. Despite this, we are able to demonstrate in Appendix H.10 that a combination of the EMA Scaling Rule and progressive scaling can match, or surpass baseline BYOL performance at a batch size of 32,768 using LARS, indicating that although the theoretical guarantees may not hold, there is still practical utility in the tools we provide in this work.

3. It may be the case that the optimal performance attainable by a given model setup exists at a level of discretization/gradient noise where no SDE exists. In this case, SDE-derived scaling rules can never be valid, and no scaling of this dynamics can be achieved with known tools.

The second issue is related to the case when the optimizer scaling rule is valid. In this case, the error for the EMA Scaling Rule at finite learning rate $\eta$ at large $\kappa$ can be considerable. In cases where the model EMA plays a role in the overall optimization, the error introduced by the EMA Scaling Rule can break the preservation of model dynamics.

Put another way, an optimizer scaling rule and the EMA Scaling Rule each introduce their own discretization errors. In the case where EMA plays a role in optimization, as soon as the discretization error of *either* the optimizer scaling rule *or* the EMA Scaling Rule is large, the error for the joint optimization procedure is large. This is *at least* as bad as cases that *do not* use a model EMA during the optimization process.

## C  The scaling toolbox: practical methods for enabling systematic scaling

There are many different components involved in preserving optimization dynamics at different batch sizes. In this appendix we collect into a single place the different concepts and values that we found useful in practice, in an attempt to make the practice of scaling as accessible as possible.

### C.1  The continuous time/SDE perspective

Here we discuss the mindset difference required when trying to preserve training dynamics. In ML we typically use stochastic optimization, leading us to think of the optimization in terms of *performing updates*, or *stepping the optimizer*. This notion has become more common in the era of large datasets, where it may be the case that we only see a fraction of the dataset during optimization.

For dynamics preservation under scaling, we suggest that it is simpler to consider the *amount of data* seen by the training process, or alternatively, the amount of *continuous time* in the discretization of SDEs view. The reason is the following. The SDE scaling rule results (Definition 1.2, Li et al. (2019, 2021); Malladi et al. (2022)) follow from showing that different discretizations of the SDE are close to that SDE, providing we appropriately scale hyperparameters (see Section 2.2). Each of these discretizations shares the *total continuous time* $T = \hat{\eta} \times \widehat{N}_{\text{iter}}$[10] of the underlying SDE, but each discretization has a *different* number of iterations $\widehat{N}_{\text{iter}} = N_{\text{iter}}/\kappa$.

This perspective is already adopted, perhaps by accident in some domains. For example, in Computer Vision (CV), it is typical to compare model performance after optimization on ImageNet1k after a *number of epochs*, whilst also specifing a learning rate warmup after a *number of epochs*. This transforms the schedule into the form *wait until the process meets [condition]*, where here *[condition]* is *when the process has seen sufficiently many samples*.

More generally, we can specify any *condition* that is not a property of the discretization procedure itself. Instead, the discretization procedure should be viewed as a numerical approximation method for the SDE we are evolving, and the properties of that discretization process (like *number of steps*) are not *of specific interest* in the world view where we do decouple optimization from the batch size. A specific example of this more general case is present in Section 3.3, where for scaling $\kappa > 2$ we wait until the pre-training Word Error Rate (WER) is sufficiently low.

There may be cases where one is working with a setup that is explicitly defined in terms of quantities related to the discretization process. Indeed, the optimizer hyperparameters are examples of these, and need to be scaled accordingly with $\kappa$. The other typical example of this is conditions based on the *number of optimizer steps*, rather than the number of epochs. In this case, these quantities should be scaled to achieve the desired condition in the same amount of time, i.e. as above $\widehat{N}_{\text{iter}} = N_{\text{iter}}/\kappa$, where $N_{\text{iter}}$ is the number of iterations specified at the base batch size $B$. Concretely, if training is specified in a number of steps, then doubling the batch size implies you should train for half the number of steps.

---

[10]This is in the case of SGD, for RMSProp and Adam one should use $T = \hat{\eta}^2 \times \widehat{N}_{\text{iter}}$ (Malladi et al., 2022).

## C.2    Scaling rules for optimization

For ease of reference, we collect all the scaling rules related to batch size modification we are aware of. We begin with the most well-known, the SGD Scaling Rule (Definitions 2.2 and C.1).

**Definition C.1** (SGD Scaling Rule). *When running SGD (Definition 2.1) with batch size $\hat{B} = \kappa B$, use a learning rate $\hat{\eta} = \kappa \eta$ (Krizhevsky, 2014; Goyal et al., 2017).*

The SGD Scaling Rule is also known as the Linear Scaling Rule (LSR), although for clarity, this work adopts the naming convention *[Algorithm Name] Scaling Rule*, which means all parameters of those algorithms are appropriately scaled from batch size $B$ to $\kappa B$.

Next, we give the two scaling rules known for the adapative optimizers RMSProp (Tieleman et al., 2012) and Adam (Kingma & Ba, 2015) in Definition C.2 and Definition C.3 respectively.

**Definition C.2** (RMSProp Scaling Rule). *When running RMSProp (Tieleman et al., 2012) with batch size $\hat{B} = \kappa B$, use a learning rate $\hat{\eta} = \sqrt{\kappa}\eta$, beta coefficient $\hat{\beta} = 1 - \kappa \times (1 - \beta)$, and adaptivity parameter $\hat{\epsilon} = \frac{\epsilon}{\sqrt{\kappa}}$ (Malladi et al., 2022).*

**Definition C.3** (Adam Scaling Rule). *When running Adam (Kingma & Ba, 2015) with batch size $\hat{B} = \kappa B$, use a learning rate $\hat{\eta} = \sqrt{\kappa}\eta$, beta coefficients $\hat{\beta}_1 = 1 - \kappa \times (1 - \beta_1)$, $\hat{\beta}_2 = 1 - \kappa \times (1 - \beta_2)$, and adaptivity parameter $\hat{\epsilon} = \frac{\epsilon}{\sqrt{\kappa}}$ (Malladi et al., 2022).*

Next, we present a contribution of this work, the EMA Scaling Rule (Definitions 1.2 and C.4), which extends the above scaling rules to allow the presence of a model EMA which is able to contribute to the overall optimization (see Appendices D and E.1 for derivations).

**Definition C.4** (EMA Scaling Rule). *When computing the EMA update (Definition 1.1) of a model undergoing stochastic optimization with batch size $\hat{B} = \kappa B$, use a momentum $\hat{\rho} = \rho^{\kappa}$ and scale other optimizers according to their own scaling rules.*

Concretely, if we are using SGD in the presence of a model EMA, Definitions C.1 and C.4 state that we should take $\hat{\eta} = \kappa \eta$ and $\hat{\rho} = \rho^{\kappa}$ when scaling by $\kappa = \hat{B}/B$.

The final scaling rule is for weight decay, and follows from the scaling logic discussed in Appendix C.1 and Krizhevsky (2014). If we take the weight decay regularization penalty $\lambda$ defined at batch size $B$, what should the weight decay $\hat{\lambda}$ be for batch size $\hat{B} = \kappa B$? For simplicity, consider $\kappa$ updates of optimization of parameters $\theta_t$ in the presence of weight decay only

$$\theta_{t+\kappa} = \theta_{t+\kappa-1} - \eta\,\lambda\,\theta_{t+\kappa-1} = (1 - \eta\,\lambda)\,\theta_{t+\kappa-1} = (1 - \eta\,\lambda)^{\kappa}\,\theta_t. \tag{11}$$

Therefore, to match the effect of weight decay with a single iteration step, we need to match

$$1 - \hat{\eta}\,\hat{\lambda} = (1 - \eta\,\lambda)^{\kappa}. \tag{12}$$

Solving for $\hat{\lambda}$ and expanding around $\eta \approx 0$ gives

$$\hat{\lambda} = \frac{1 - (1 - \eta\,\lambda)^{\kappa}}{\hat{\eta}} \approx \frac{\eta}{\hat{\eta}} \times \kappa\,\lambda + O(\eta). \tag{13}$$

This leads to the Weight Decay Scaling Rule (Definition C.5).

**Definition C.5** (Weight Decay Scaling Rule). *When using weight decay with batch size $\hat{B} = \kappa B$, use a penalty term $\hat{\lambda} = (\kappa\hat{\eta}/\eta)\,\lambda$, where $\hat{\eta}$ and $\eta$ represent the scaled and unscaled learning rates of the corresponding optimizer (Krizhevsky, 2014; Li et al., 2018; Loshchilov & Hutter, 2019).*

The Weight Decay Scaling Rule implies that using *linear* scaling for the learning rate $\eta$ then the weight decay penalty is automatically scaled, and when using *square-root* scaling for the learning rate $\eta$ (e.g. in the case of the Adam Scaling Rule (Definition C.3)) then the weight decay penalty should also be scaled with a *square-root* as is proposed in Loshchilov & Hutter (2019).

Finally, we see that if the implementation of weight decay does not have an update scaled by the learning rate, i.e. the update is $\theta_{t+1} = (1 - \lambda)\,\theta_t$, then the scaling rule is optimizer-independent, and becomes linear for small weight decay, i.e. $\hat{\lambda} = \kappa\lambda$, and for arbitrary $\lambda$ takes the form $\hat{\lambda} = 1 - (1 - \lambda)^{\kappa}$.

Table 2: Scaled learning rates $\hat{\eta}$ at different batch sizes $\hat{B} = \kappa B$ given reference learning rates $\eta$ defined at batch size $B$. The reference values of each column are boldened. Note that this is only valid when there is a notion of *single sample*. In the sequence learning setup (for example, in Section 3.3), the notion of batch size should be appropriately replaced with the *dynamic batch size*, i.e. total sequence length.

| | $\hat{\eta} = \kappa\eta$ [SGD] | | | $\hat{\eta} = \sqrt{\kappa}\eta$ [RMSProp, Adam] | | |
| | $B = 256$ | | $B = 512$ | $B = 256$ | $B = 4096$ | |
| Batch size $\hat{B}$ | $\eta = 0.1$ | $\eta = 0.3$ | $\eta = 0.1$ | $\eta = 10^{-3}$ | $\eta = 4.8$ | $\eta = 10^{-3}$ |
|---|---|---|---|---|---|---|
| 32 | 0.0125 | 0.0375 | 0.00625 | 0.00035 | 0.42426 | 0.00009 |
| 64 | 0.025 | 0.075 | 0.0125 | 0.0005 | 0.6 | 0.00013 |
| 128 | 0.05 | 0.15 | 0.025 | 0.00071 | 0.84853 | 0.00018 |
| 256 | **0.1** | **0.3** | 0.05 | **0.001** | 1.2 | 0.00025 |
| 512 | 0.2 | 0.6 | **0.1** | 0.00141 | 1.69706 | 0.00035 |
| 1024 | 0.4 | 1.2 | 0.2 | 0.002 | 2.4 | 0.0005 |
| 2048 | 0.8 | 2.4 | 0.4 | 0.00283 | 3.39411 | 0.00071 |
| 4096 | 1.6 | 4.8 | 0.8 | 0.004 | **4.8** | **0.001** |
| 8192 | 3.2 | 9.6 | 1.6 | 0.00566 | 6.78823 | 0.00141 |
| 16384 | 6.4 | 19.2 | 3.2 | 0.008 | 9.6 | 0.002 |
| 32768 | 12.8 | 38.4 | 6.4 | 0.01131 | 13.57645 | 0.00283 |
| 65536 | 25.6 | 76.8 | 12.8 | 0.016 | 19.2 | 0.004 |

Table 3: Scaled EMA momenta $\hat{\rho} = \rho^{\kappa}$ at different batch sizes $\hat{B} = \kappa B$ given reference momenta $\rho$ defined at batch size $B$. The reference values of each column are boldened. Again in the sequence learning setup, batch size should be appropriately replaced with a notion of sequence length.

| | $B = 256$ | | | $B = 4096$ | | | |
| Batch size $\hat{B}$ | $\rho = 0.9999$ | $\rho = 0.999$ | $\rho = 0.99$ | $\rho = 0.996$ | $\rho = 0.992$ | $\rho = 0.99$ | $\rho = 0.97$ |
|---|---|---|---|---|---|---|---|
| 32 | 0.99999 | 0.99987 | 0.99874 | 0.99997 | 0.99994 | 0.99992 | 0.99976 |
| 64 | 0.99997 | 0.99975 | 0.99749 | 0.99994 | 0.99987 | 0.99984 | 0.99952 |
| 128 | 0.99995 | 0.9995 | 0.99499 | 0.99987 | 0.99975 | 0.99969 | 0.99905 |
| 256 | **0.9999** | **0.999** | **0.99** | 0.99975 | 0.9995 | 0.99937 | 0.9981 |
| 512 | 0.9998 | 0.998 | 0.9801 | 0.9995 | 0.999 | 0.99874 | 0.9962 |
| 1024 | 0.9996 | 0.99601 | 0.9606 | 0.999 | 0.99799 | 0.99749 | 0.99241 |
| 2048 | 0.9992 | 0.99203 | 0.92274 | 0.998 | 0.99599 | 0.99499 | 0.98489 |
| 4096 | 0.9984 | 0.98412 | 0.85146 | **0.996** | **0.992** | **0.99** | **0.97** |
| 8192 | 0.9968 | 0.96849 | 0.72498 | 0.99202 | 0.98406 | 0.9801 | 0.9409 |
| 16384 | 0.99362 | 0.93798 | 0.5256 | 0.9841 | 0.96838 | 0.9606 | 0.88529 |
| 32768 | 0.98728 | 0.8798 | 0.27625 | 0.96844 | 0.93776 | 0.92274 | 0.78374 |
| 65536 | 0.97472 | 0.77405 | 0.07632 | 0.93788 | 0.8794 | 0.85146 | 0.61425 |

## C.3 Commonly used values of hyperparameters at different batch sizes

In the literature it is common to give a base learning rate $\eta$ defined at batch size 256, implicitly using the SGD Scaling Rule, even when using the Adam optimizer. Because the scaling of other optimization hyperparameters was not understood until recently, it is also common to just present these *for the experiment*, e.g. the Adam betas and epsilon, and the EMA momentum, implicitly defined at the scale of the experiment, for example at batch size 4096. One way to deal with this in practice is to define a single reference batch size $B$ at which *all* hyperparameters are defined, and then scale from there. In this case, it is easiest to compute *using linear scaling* the learning rate at the redefined base batch size $\eta = \tilde{\kappa}\,\eta_{\mathrm{orig}}$, where $\tilde{\kappa} = B/B_{\mathrm{orig}}$, and then scale this new reference $\eta$ as $\hat{\eta} = \kappa\eta$, $\kappa = \hat{B}/B$, along with e.g. the momentum defined at $B$.

As this process can be slightly frustrating, we have provided tables of typical learning rates in Table 2 and momenta in Table 3.

## C.4 Progressive scaling

In Section 3.4 we introduced Progressive Scaling (Definition 3.2) to test our hypothesis that early in the BYOL training procedure, there are dynamics that are challenging to replicate at larger batch

---

**Algorithm 1** Stochastic Gradient Descent with Progressive Scaling

---

**Require:** Base learning rate $\eta$, base momentum $\rho$ for base batch size $B$
**Require:** Initial target model parameters $\theta$ and model EMA parameters $\zeta$
**Require:** Epochs $E$ and schedule of batch sizes $\mathcal{B} = B_1, B_2, \ldots, B_E$
**Require:** Loss function $\mathcal{L}$
  **for** $e$ in $1, 2 \ldots, E$ **do**
    $\hat{B} \leftarrow \mathcal{B}[e]$                                               ▷ Get current batch size
    $\kappa \leftarrow \hat{B}/B$                                            ▷ Compute scaling factor
    $\hat{\eta} \leftarrow \kappa\eta$                                             ▷ Get scaled learning rate
    $\hat{\rho} \leftarrow \rho^\kappa$                                             ▷ Get scaled momentum
    **for** $b$ in $1, 2 \ldots, \text{floor}(E/\hat{B})$ **do**
      Sample a minibatch of $\hat{B}$ samples $X = \{\boldsymbol{x}^{(1)}, \ldots, \boldsymbol{x}^{(\hat{B})}\}$
      $\theta \leftarrow \theta - (\hat{\eta}/\hat{B}) \sum_{x \in X} \nabla_\theta \mathcal{L}(x; \theta, \zeta)$            ▷ SGD Update
      $\zeta \leftarrow \hat{\rho}\,\zeta + (1 - \hat{\rho})\,\theta$                              ▷ EMA Update
    **end for**
  **end for**

---

sizes. To remove ambiguity, in Algorithm 1 we provide pseudo-code for how to use Progressive Scaling.

In Algorithm 1, the prefactor of the SGD update could also have been written $\eta/B$, although an equivalent use of the base momentum is not possible.

Finally, we outline how to extend Algorithm 1 to more complex setups, like those presented in Section 3.4:

1. Optimizer scaling rules are used appropriately, for example the Adam scaling rule in case of using the Adam optimizer to update parameters $\theta$.

2. Schedules for hyperparameters are computed using the base hyperparameters, and are then modified by application of the scaling law in epoch (outer) loop.

3. Schedules for hyperparameters at the *step* rather than epoch level can be achieved in practice through recomputing the schedule and updating the notion of minibatch index appropriately throughout training.

All of the above techniques are used in Section 3.4. In addition, scheduling batch sizes within epoch is possible, providing one maintains a notion of computation within some fixed continuous time $T_{\text{fixed}}$. We did not investigate this scenario.

## D   EMA approximation theorems with SDEs

### D.1   SGD with model EMA

We will now derive the EMA scaling rule when tracking model parameters and the model is trained using SGD. We employ a strategy similar to Malladi et al. (2022), where we associate to each iterative process a Stochastic Differential Equation (SDE). In order to control the distance between the SDE and the discrete process, we use the tools from Li et al. (2019).

**Definition D.1** (Polynomial growth, Definition 1 in (Li et al., 2019)). *The set $G$ is the set of continuous functions $\mathbb{R}^d \to \mathbb{R}$ with at most polynomial growth, i.e., for $g \in G$ there exists two scalars $\kappa_1, \kappa_2 > 0$ such that for all $\boldsymbol{x} \in \mathbb{R}^d$, we have $|g(\boldsymbol{x})| \leq \kappa_1(1 + \|\boldsymbol{x}\|^{\kappa_2})$.*

*For an integer $\alpha > 0$, $G^\alpha$ is the set of functions $\mathbb{R}^d \to \mathbb{R}$ that are $\alpha$-times continuously differentiable and such that all their derivatives up to order $\alpha$ are in $G$.*

Similarly to Malladi et al. (2022), we use Noisy Gradient Oracle with Scale Parameter (NGOS) to define the update rules on the parameters.

**Definition D.2** (Noisy Gradient Oracle with Scale Parameter (NGOS), adaptation of (Malladi et al., 2022)). *A NGOS is a tuple $\mathcal{G}_\sigma = (f, \Sigma, \mathcal{Z}_\sigma)$. Given a noise scale parameter $\sigma > 0$, the NGOS $\mathcal{G}_\sigma$*

takes as input the parameters $\boldsymbol{\theta}$ and outputs a random vector $\mathbf{g} = \nabla f(\boldsymbol{\theta}, \boldsymbol{\zeta}) + \sigma \boldsymbol{\epsilon}$ where $\nabla f(\boldsymbol{\theta}, \boldsymbol{\zeta})$ is the gradient of $f$ with respect to $\boldsymbol{\theta}$ at $(\boldsymbol{\theta}, \boldsymbol{\zeta})$, and $\boldsymbol{\epsilon}$ is a random vector drawn from the distribution $\mathcal{Z}_\sigma(\boldsymbol{\theta}, \boldsymbol{\zeta})$ with zero mean and covariance $\Sigma(\boldsymbol{\theta}, \boldsymbol{\zeta})$.

Note that in the above definition, the probability distribution $\mathcal{Z}_\sigma(\boldsymbol{\theta}, \boldsymbol{\zeta})$ is allowed to change with the scale $\sigma$, but its first two moments — its mean and its covariance — are fixed with $\sigma$. We have the following theorem for model EMA under optimization with SGD:

**Theorem D.1** (SDE for SGD + EMA). *Consider the couple $\mathbf{x}_k = (\boldsymbol{\theta}_k, \boldsymbol{\zeta}_k)$ where $\boldsymbol{\theta}_k$ are the iterates of SGD with a NGOS (Definition D.2) and $\boldsymbol{\zeta}_k$ is an EMA of $\boldsymbol{\theta}_k$, defined, starting from $\mathbf{x}_0 = \boldsymbol{x}_0$, by*

$$\boldsymbol{\theta}_{k+1} = \boldsymbol{\theta}_k - \eta \mathbf{g}_k, \quad \text{with } \mathbf{g}_k = \nabla f(\boldsymbol{\theta}_k, \boldsymbol{\zeta}_k) + \sigma \boldsymbol{\epsilon}_k, \text{ and } \boldsymbol{\epsilon}_k \sim \mathcal{Z}_\sigma(\boldsymbol{\theta}_k, \boldsymbol{\zeta}_k), \tag{14}$$

$$\boldsymbol{\zeta}_{k+1} = \rho \boldsymbol{\zeta}_k + (1 - \rho) \boldsymbol{\theta}_k \ . \tag{15}$$

*Define $\beta_0 = (1 - \rho)/\eta$, $\sigma_0 = \sigma\sqrt{\eta}$, and define the SDE for $X_t = (\Theta_t, Z_t)$, starting from $X_0 = \boldsymbol{x}_0$, by*

$$d\Theta_t = -\nabla f(\Theta_t, Z_t) dt + \sigma_0 \Sigma(\Theta_t, Z_t)^{\frac{1}{2}} dW_t, \quad \text{with } W_t \text{ a Wiener process} \tag{16}$$

$$dZ_t = \beta_0(\Theta_t - Z_t) dt \ . \tag{17}$$

*Assume that $f$ is continuously differentiable, with $f \in G^3$ and $\Sigma^{\frac{1}{2}} \in G^2$ (Definition D.1). Then, for any time horizon $T > 0$ and test function $g \in G^2$, there exists a constant $c > 0$ such that*

$$\max_{k=0,\ldots,\lfloor T/\eta \rfloor} |\mathbb{E}[g(X_{\eta k})] - \mathbb{E}[g(\mathbf{x}_k)]| \le c \times \eta \ . \tag{18}$$

*Proof.* The proof uses the same tools as in Li et al. (2019). Define $\Delta(\boldsymbol{\theta}, \boldsymbol{\zeta}) = \eta(-\nabla f(\boldsymbol{\theta}, \boldsymbol{\zeta}) + \sigma \boldsymbol{\epsilon}, \beta_0(\boldsymbol{\theta} - \boldsymbol{\zeta}))$ with $\boldsymbol{\epsilon} \sim \mathcal{Z}_\sigma(\boldsymbol{\theta}, \boldsymbol{\zeta})$ the one-step update for the SGD + EMA update, such that $\mathbf{x}_{k+1} = \mathbf{x}_k + \Delta(\mathbf{x}_k)$. We have the first two moments:

$$\mathbb{E}[\Delta(\boldsymbol{\theta}, \boldsymbol{\zeta})] = \eta(-\nabla f(\boldsymbol{\theta}, \boldsymbol{\zeta}), \beta_0(\boldsymbol{\theta} - \boldsymbol{\zeta})) \tag{19}$$

$$\mathbb{V}[\Delta(\boldsymbol{\theta}, \boldsymbol{\zeta})] = \eta \sigma_0^2 \begin{bmatrix} \Sigma(\boldsymbol{\theta}, \boldsymbol{\zeta}) & 0 \\ 0 & 0 \end{bmatrix} \tag{20}$$

and the higher-order moments are $O(\eta^2)$. Similarly, let $\tilde{\Delta}(\boldsymbol{\theta}, \boldsymbol{\zeta})$ be the solution at time $\eta$ of the SDE defined by Equation 6 starting from $X_0 = (\boldsymbol{\theta}, \boldsymbol{\zeta})$. From Ito's formula, we also obtain

$$\mathbb{E}[\tilde{\Delta}(\boldsymbol{\theta}, \boldsymbol{\zeta})] = \eta(-\nabla f(\boldsymbol{\theta}), \beta_0(\boldsymbol{\theta} - \boldsymbol{\zeta})) \tag{21}$$

$$\mathbb{V}[\tilde{\Delta}(\boldsymbol{\theta}, \boldsymbol{\zeta})] = \eta \sigma_0^2 \begin{bmatrix} \Sigma(\boldsymbol{\theta}, \boldsymbol{\zeta}) & 0 \\ 0 & 0 \end{bmatrix} \tag{22}$$

and the higher-order moments are $O(\eta^2)$. Hence, the moments of the discrete iteration and of the SDE match up to second order. Following the same proof technique as in Li et al. (2019) then leads to the advertized theorem. $\square$

This theorem is a simple adaptation of the results of Li et al. (2019). Intuitively, it is expected that $X_t$ and $\mathbf{x}_k$ are close since $\mathbf{x}_k$ is the Euler-Maruyama discretization of $X_t$ with learning rate $\eta$. We then have the corollary.

**Corollary D.1.1** (Validity of the EMA Scaling Rule). *Assume that $f$ is continuously differentiable, with $f \in G^3$ and $\Sigma^{\frac{1}{2}} \in G^2$. Let $\boldsymbol{\theta}_k^B, \boldsymbol{\zeta}_k^B$ the iterates of the Equation 5 with batch size $B$ and hyperparameters $\eta, \rho$. Let $\boldsymbol{\theta}_k^{\kappa B}, \boldsymbol{\zeta}_k^{\kappa B}$ be iterates with batch size $\kappa B$, learning rate $\eta$ determined by the SGD Scaling Rule (Definition 2.2) and momentum determined by the EMA Scaling Rule, linear version (Definition 1.2). Then, for any time horizon $T > 0$ and function $g \in G^2$, there exists a constant $d > 0$ such that*

$$\max_{k=0,\ldots,\lfloor T/\eta \rfloor} |\mathbb{E}[g(\boldsymbol{\theta}_{\lfloor k/\kappa \rfloor}^{\kappa B}, \boldsymbol{\zeta}_{\lfloor k/\kappa \rfloor}^{\kappa B})] - \mathbb{E}[g(\boldsymbol{\theta}_k, \boldsymbol{\zeta}_k)]| \le d \times \eta \ . \tag{23}$$

*Proof.* The proof is similar to Malladi et al. (2022). Under the scaling rule, both $\mathbf{x}_k = (\boldsymbol{\theta}_k, \boldsymbol{\zeta}_k)$ and $\hat{\mathbf{x}}_{\lfloor k/\kappa \rfloor} = (\boldsymbol{\theta}_{\lfloor k/\kappa \rfloor}^{\kappa B}, \boldsymbol{\zeta}_{\lfloor k/\kappa \rfloor}^{\kappa B})$ have the same limiting SDE. Hence we have from the previous theorem that for all test function $g$, we can find $c, c'$ such that

$$\max_{k=0,\ldots,\lfloor T/\eta \rfloor} |\mathbb{E}[g(X_{\eta k})] - \mathbb{E}[g(\mathbf{x}_k)]| \le c \times \eta \text{ and } \max_{k=0,\ldots,\lfloor T/\eta \rfloor} |\mathbb{E}[g(X_{\eta k})] - \mathbb{E}[g(\hat{\mathbf{x}}_{\lfloor k/\kappa \rfloor})]| \le c' \times \eta. \tag{24}$$

The triangle inequality then gives

$$\max_{k=0,\dots,\lfloor T/\eta \rfloor} |\mathbb{E}[g(\hat{\mathbf{x}}_{\lfloor k/\kappa \rfloor})] - \mathbb{E}[g(\mathbf{x}_k)]| \le (c + c') \times \eta. \tag{25}$$

Hence, taking $d = c + c'$ gives the expected result. $\qquad\square$

## D.2 Adaptive gradient methods with model EMA

We now turn to the case where one uses an adaptive gradient method rather than SGD to train the model. We follow derivations similar to those of Malladi et al. (2022), with an added EMA. Like above, we consider that the loss function $f$ also depends on the EMA tracking parameter $\zeta_k$. We begin with RMSProp with EMA, which iterates:

$$\mathbf{v}_{k+1} = \gamma \mathbf{v}_k + (1 - \gamma)\mathbf{g}_k^2, \quad \text{with } \mathbf{g}_k = \nabla f(\theta_k, \zeta_k) + \sigma \epsilon_k, \text{ and } \epsilon_k \sim \mathcal{Z}_\sigma(\theta_k, \zeta_k), \tag{26}$$

$$\theta_{k+1} = \theta_k - \eta(\sqrt{\mathbf{v}_k} + \varepsilon)^{-1} \times \mathbf{g}_k \tag{27}$$

$$\zeta_{k+1} = \rho \zeta_k + (1 - \rho)\theta_k. \tag{28}$$

Like in Malladi et al. (2022), we place ourselves in the high noise regime, in which the term $\mathbf{g}_k^2$ in Equation 26 is approximated by $\mathbf{g}_k^2 \simeq \sigma^2 \text{diag}(\Sigma(\theta_k, \zeta_k))$. We use the same scaling rules, with an additional one for $\rho$:

$$\gamma_0 = (1 - \gamma)/\eta^2, \quad \sigma_0 = \sigma\eta, \quad \varepsilon_0 = \varepsilon\eta, \text{ and } \beta_0 = (1 - \rho)/\eta^2, \tag{29}$$

and we let $\mathbf{u}_k = \mathbf{v}_k/\sigma^2$. The equations for RMSProp with EMA then become, using only these new variables and $\eta$:

$$\mathbf{u}_{k+1} - \mathbf{u}_k = \eta^2 \gamma_0 (\text{diag}(\Sigma(\theta_k, \zeta_k)) - \mathbf{u}_k), \tag{30}$$

$$\theta_{k+1} - \theta_k = -(\sqrt{\mathbf{u}_k} + \varepsilon_0)^{-1}\left(\eta^2 \nabla f(\theta_k, \zeta_k) + \eta\epsilon_k\right) \tag{31}$$

$$\zeta_{k+1} - \zeta_k = \eta^2 \beta_0(\theta_k - \zeta_k). \tag{32}$$

This formulation makes it clear that these iterations can be seen as the discretization of the SDE

$$dU_t = \gamma_0(\text{diag}(\Sigma(\Theta_t, Z_t)) - U_t)dt, \tag{33}$$

$$d\Theta_t = -(\sigma_0\sqrt{U_t} + \varepsilon_0)^{-1}(\nabla f(\Theta_t, Z_t)dt + \sigma_0\Sigma(\Theta_t, Z_t)^{1/2}dWt) \tag{34}$$

$$dZ_t = \beta_0(\Theta_t - Z_t)dt, \tag{35}$$

with step size $\eta^2$. Of course, we recover the SDE of Malladi et al. (2022) in the case where $\beta_0 = 0$. A formal proof of closeness between the iterates and the SDE trajectory is out of the scope of the present paper since it would imply redoing much of the theoretical work developed in Malladi et al. (2022). Still, the previous informal analysis hints that for RMSProp, the scaling rule in Equation 29 should be used. In other words, given a certain set of hyperparameters $\gamma, \eta$ and $\rho$, if the batch size goes from $B$ to $\hat{B} = \kappa \times B$, the noise level becomes $\hat{\sigma} = \sigma/\sqrt{\kappa}$, and keeping the quantities in Equation 29 constant means that we should use as new hyperparameters

$$\hat{\gamma} = 1 - (1 - \gamma) \times \kappa, \quad \hat{\eta} = \eta \times \sqrt{\kappa}, \text{ and } \hat{\rho} = 1 - (1 - \rho) \times \kappa \ .$$

The linear rule $\hat{\rho} = 1 - (1 - \rho) \times \kappa$ is at the first order equivalent to the exponential scaling rule $\hat{\rho} = \rho^\kappa$. Hence, even though the limiting SDE differs greatly from that of SGD, and even though the scaling rule regarding the learning rate differs, we recover for the momentum term $\rho$ the exact same scaling rule as for SGD.

We finish the discussion with the case of Adam, which leads once again to the same rule as for SGD. Adam with EMA tracking of the network parameters iterates

$$\mathbf{m}_{k+1} = \beta_1 \mathbf{m}_k + (1 - \beta_1)\mathbf{g}_k, \quad \text{with } \mathbf{g}_k = \nabla f(\theta_k, \zeta_k) + \sigma\epsilon_k, \text{ and } \epsilon_k \sim \mathcal{Z}_\sigma(\theta_k, \zeta_k), \tag{36}$$

$$\mathbf{v}_{k+1} = \beta_2 \mathbf{v}_k + (1 - \beta_2)\mathbf{g}_k^2 \tag{37}$$

$$\tilde{\mathbf{m}}_{k+1} = \mathbf{m}_{k+1}/(1 - \beta_1^{k+1}) \tag{38}$$

$$\tilde{\mathbf{v}}_{k+1} = \mathbf{v}_{k+1}/(1 - \beta_2^{k+1}) \tag{39}$$

$$\theta_{k+1} = \theta_k - \eta(\sqrt{\tilde{\mathbf{v}}_k} + \varepsilon)^{-1} \times \tilde{\mathbf{m}}_{k+1} \tag{40}$$

$$\zeta_{k+1} = \rho \zeta_k + (1 - \rho)\theta_k \ . \tag{41}$$

Here, we use the same minor modification of the iterations as in Malladi et al. (2022), where we use $\mathbf{v}_k$ instead of $\mathbf{v}_{k+1}$ in the denominator of the $\boldsymbol{\theta}_k$ update.

We consider the following scaling for the hyperparameters

$$c_1 = (1 - \beta_1)/\eta^2, \quad c_2 = (1 - \beta_2)/\eta^2, \quad \sigma_0 = \sigma\eta, \quad \varepsilon_0 = \varepsilon\eta, \text{ and } \beta_0 = (1 - \rho)/\eta^2, \tag{42}$$

and $\gamma_1(t) = 1 - \exp(-c_1 t)$, $\gamma_2(t) = 1 - \exp(-c_2 t)$, and $\mathbf{u}_k = \mathbf{v}_k/\sigma^2$. The SDE for Adam + EMA is given by

$$dM_t = c_1 \left( (\nabla f(\Theta_t, Z_t) - M_t)dt + \sigma_0 \Sigma(\Theta_t, Z_t)^{1/2} dW_t \right) \tag{43}$$

$$dU_t = c_2 (\text{diag}(\Sigma(\Theta_t, Z_t)) - U_t)dt \tag{44}$$

$$d\Theta_t = -\frac{\sqrt{\gamma_2(t)}}{\gamma_1(t)} (\sigma_0 \sqrt{U_t} + \varepsilon_0 \sqrt{\gamma_2(t)})^{-1} \times M_t dt \tag{45}$$

$$dZ_t = \beta_0 (\Theta_t - Z_t)dt. \tag{46}$$

This is once again the same SDE as in Malladi et al. (2022) with the added EMA term. Like previously, this SDE hints at the fact that the scaling rule in eq. 42 should be used. In other words, given a set of hyperparameters $\beta_1, \beta_2, \eta$, and $\rho$, if the batch size goes from $B$ to $\kappa \times B$, then the noise level becomes $\hat{\sigma} = \sigma/\sqrt{\kappa}$ and keeping quantities in eq. 42 constant means that we should use as new hyperparameters

$$\hat{\beta}_1 = 1 - (1 - \beta_1) \times \kappa, \quad \hat{\beta}_2 = 1 - (1 - \beta_2) \times \kappa, \quad \hat{\eta} = \eta \times \sqrt{\kappa}, \text{ and } \hat{\rho} = 1 - (1 - \rho) \times \kappa.$$

We once again recover a linear rule for $1 - \rho$ which is equivalent to the exponential scaling rule $\hat{\rho} = \rho^\kappa$ in the limit $\rho \to 0$.

# E   Additional proofs

## E.1   Iterations of SGD + EMA

Here we derive a critical component of the EMA Scaling Rule, the matrix equation of Equation 4 from which the EMA Scaling Rule (Definition 1.2) follows.

**Theorem E.1** (Iterations of SGD + EMA). *Assuming that gradients change slowly over iterations of SGD (Definition 2.1) and EMA (Definition 1.1):* $\nabla_\theta \mathcal{L}(x; \theta_{t+j}, \zeta_{t+j}) \approx \nabla_\theta \mathcal{L}(x; \theta_t, \zeta_t) \approx \mathbf{g}$, *for $j = 1, 2, \ldots, \kappa$ and representative gradient $\mathbf{g}$, iterating over $\kappa$ independent minibatches produces model states*

$$\begin{bmatrix} \theta_{t+\kappa} \\ \zeta_{t+\kappa} \\ \mathbf{g} \end{bmatrix} = \begin{bmatrix} 1 & 0 & -\eta \\ 1 - \rho & \rho & 0 \\ 0 & 0 & 1 \end{bmatrix}^\kappa \cdot \begin{bmatrix} \theta_t \\ \zeta_t \\ \mathbf{g} \end{bmatrix} = \begin{bmatrix} \theta_t - \eta\,\kappa\,\mathbf{g} \\ \rho^\kappa \zeta_t + (1 - \rho^\kappa) \theta_t + O\left(\eta \times \beta_\rho\right) \\ \mathbf{g} \end{bmatrix}. \tag{47}$$

*Proof.* First note that for matrices of the form

$$A = \begin{bmatrix} 1 & 0 & a_{0,2} \\ 1 - a_{1,1} & a_{1,1} & 0 \\ 0 & 0 & 1 \end{bmatrix}, \tag{48}$$

their multiplication follows

$$A\,B = \begin{bmatrix} 1 & 0 & a_{0,2} \\ 1 - a_{1,1} & a_{1,1} & 0 \\ 0 & 0 & 1 \end{bmatrix} \begin{bmatrix} 1 & 0 & b_{0,2} \\ 1 - b_{1,1} & b_{1,1} & 0 \\ 0 & 0 & 1 \end{bmatrix}$$

$$= \begin{bmatrix} 1 & 0 & a_{0,2} + b_{0,2} \\ 1 - a_{1,1} b_{1,1} & a_{1,1} b_{1,1} & (1 - a_{1,1}) b_{0,2} \\ 0 & 0 & 1 \end{bmatrix}, \tag{49}$$

and

$$A\,B\,C = \begin{bmatrix} 1 & 0 & a_{0,2} + b_{0,2} \\ 1 - a_{1,1} b_{1,1} & a_{1,1} b_{1,1} & (1 - a_{1,1}) b_{0,2} \\ 0 & 0 & 1 \end{bmatrix} \begin{bmatrix} 1 & 0 & c_{0,2} \\ 1 - c_{1,1} & c_{1,1} & 0 \\ 0 & 0 & 1 \end{bmatrix}$$

$$= \begin{bmatrix} 1 & 0 & a_{0,2} + b_{0,2} + c_{0,2} \\ 1 - a_{1,1} b_{1,1} c_{1,1} & a_{1,1} b_{1,1} c_{1,1} & (1 - a_{1,1}) b_{0,2} + (1 - a_{1,1} b_{1,1}) c_{0,2} \\ 0 & 0 & 1 \end{bmatrix}. \tag{50}$$

By induction

$$A^\kappa = \begin{bmatrix} 1 & 0 & \kappa \times a_{0,2} \\ 1 - a_{1,1}^\kappa & a_{1,1}^\kappa & \delta(a_{0,2}, a_{1,1}, \kappa) \\ 0 & 0 & 1 \end{bmatrix}, \tag{51}$$

where

$$\delta(a_{0,2}, a_{1,1}, \kappa) = a_{0,2} \sum_{i=1}^{\kappa-1} \left(1 - a_{1,1}^i\right) = a_{0,2} \left(\kappa - \frac{1 - a_{1,1}^\kappa}{1 - a_{1,1}}\right), \quad \text{for } a_{1,1} \neq 1. \tag{52}$$

It follows that

$$\begin{bmatrix} 1 & 0 & -\eta \\ 1 - \rho & \rho & 0 \\ 0 & 0 & 1 \end{bmatrix}^\kappa = \begin{bmatrix} 1 & 0 & -\kappa\eta \\ 1 - \rho^\kappa & \rho^\kappa & \delta(\eta, \rho, \kappa) \\ 0 & 0 & 1 \end{bmatrix} \tag{53}$$

where the EMA Scaling Rule error

$$\delta(\eta, \rho, \kappa) = (-\eta)\left(\kappa - \frac{1 - \rho^\kappa}{1 - \rho}\right) \approx (-\eta)\left(\kappa - \kappa + O(\beta_\rho)\right) = 0 + O(\eta \times \beta_\rho), \tag{54}$$

where $\beta_\rho \equiv 1 - \rho$ and the approximation is around $\rho = 1$. $\qquad\square$

### E.2 Limiting behavior of Polyak-Ruppert averaging

Here we sketch the asymptotic behavior of a target model $\theta$ and its EMA $\zeta$. Let us assume that $\theta$ converges to the stationary distribution $\lim_{t\to\infty} \theta_t = \theta^*$, $\theta^* \sim p_\infty(\theta)$. We are interested in statistical properties of $\zeta^* = \lim_{t\to\infty} \zeta_t$, as this will formalize the notion of how the EMA depends on the a time-horizon defined by its momentum $\rho$ as discussed in Table 1.

As a warm-up, for $n$ independent random variables $x_1, \ldots, x_2$, we know that the sample mean $\bar{x} = \frac{1}{n}(x_1, x_2, \ldots, x_n)$ has the statistical properties

$$\mathbb{E}[\bar{x}] = \mu, \qquad\qquad \text{Var}[\bar{x}] = \frac{\sigma^2}{n}, \tag{55}$$

where $\mu$ and $\sigma$ are the population mean and variance. This gives us an idea of what to expect. As we will now show, the expectation of $\zeta^*$ should have no time-horizon dependence, whereas the variance of $\zeta^*$ will depend on its time horizon (i.e. the number of samples it integrates over) which is defined by $\rho$.

In the case of a weighted sum

$$\bar{x}^{(w)} = \sum_{i=1}^{n} w_i \, x_i, \tag{56}$$

then if the $x_i$ are Independent and Identically Distributed (i.i.d.), then

$$\mathbb{E}[\bar{x}^{(w)}] = \sum_{i=1}^{n} w_i \, \mathbb{E}[x_i] = n \, \bar{w} \, \mu, \qquad\qquad \bar{w} = \frac{1}{n} \sum_{i=1}^{n} w_i, \tag{57}$$

and for the variance (Kish, 1965)

$$\text{Var}[\bar{x}^{(w)}] = n \cdot \overline{w^2} \cdot \sigma^2 \qquad \overline{w^2} = \frac{1}{n} \sum_{i=1}^{n} w_i^2, \qquad \sigma^2 = \text{Var}[x_i]. \tag{58}$$

We can verify that we reproduce the well-known result in Equation 55 in the case where all weights are equal to $\frac{1}{n}$ as follows

$$\forall i : w_i = \frac{1}{n} \implies \overline{w^2} = \frac{1}{n} \cdot \sum_{i=1}^{n} \left(\frac{1}{n}\right)^2 = \frac{1}{n^2} \implies \text{Var}[\bar{x}^{(w)}] = n \cdot \frac{1}{n^2} \cdot \sigma^2 = \frac{\sigma^2}{n}. \tag{59}$$

In the case of an exponential moving average we have

$$\zeta_{t+1} = \rho \, \zeta_t + (1 - \rho) \, \theta_t = \rho^t \, \zeta_1 + (1 - \rho) \sum_{i=0}^{t-1} \rho^i \theta_{t-i}. \tag{60}$$

Let's consider the specific case where we are at iteration $k$ which is sufficiently large that $\zeta$ and $\theta$ have converged to their stationary distributions. From $k$, the iterations unfold as

$$\zeta_{t+1} = \rho^{t+1-k} \, \zeta_k + (1 - \rho) \sum_{i=0}^{t-k} \rho^i \theta_{t-i}. \tag{61}$$

We rearrange for terms in $\zeta$

$$\zeta_{t+1} - \rho^{t+1-k} \, \zeta_k = (1 - \rho) \sum_{i=0}^{t-k} \rho^i \, \theta_{t-i}, \tag{62}$$

and before proceeding to the final result, using $n = t + 1 - k$, we compute the convenient quantities

$$\bar{\rho} = \frac{1}{n} \sum_{i=0}^{n-1} \rho^i = \frac{1}{n} \times \frac{1 - \rho^n}{1 - \rho} \tag{63}$$

$$\overline{\rho^2} = \frac{1}{n} \sum_{i=0}^{n-1} \rho^{2i} = \frac{1}{n} \times \frac{1 - \rho^{2n}}{1 - \rho^2}. \tag{64}$$

Taking expectation of Equation 62 and setting statistics to their stationary values, we have

$$(1 - \rho^n) \, \mathbb{E}[\zeta^*] = (1 - \rho) \, n \, \bar{\rho} \, \mathbb{E}[\theta^*] = (1 - \rho^n) \, \mathbb{E}[\theta^*], \tag{65}$$

where we have used the result in Equation 57. It follows that for $\rho \neq 1$ we have

$$\mathbb{E}[\zeta^*] = \mathbb{E}[\theta^*], \tag{66}$$

independent of $\rho$. Finally, we can take the variance of Equation 62. First the left hand side

$$\mathrm{Var}\left[\zeta_{t+1} - \rho^n \, \zeta_k\right] = \mathrm{Var}\left[\zeta_{t+1}\right] + \rho^{2n} \, \mathrm{Var}\left[\zeta_k\right] = \left(1 + \rho^{2n}\right) \, \mathrm{Var}\left[\zeta^*\right]. \tag{67}$$

Next the right hand side

$$\mathrm{Var}\left[(1 - \rho) \sum_{i=0}^{n-1} \rho^i \, \theta_{t-i}\right] = (1 - \rho)^2 \, \mathrm{Var}\left[\sum_{i=0}^{n-1} \rho^i \, \theta_{t-i}\right] = (1 - \rho)^2 \cdot \left(\frac{1 - \rho^{2n}}{1 - \rho^2}\right) \cdot \mathrm{Var}[\theta^*]. \tag{68}$$

Finally, equating left and right hand sizes and rearranging for $\mathrm{Var}[\zeta^*]$ gives

$$\mathrm{Var}\left[\zeta^*\right] = \frac{1 - \rho^{2n}}{1 + \rho^{2n}} \cdot \frac{1 - \rho}{1 + \rho} \cdot \mathrm{Var}\left[\theta^*\right] \tag{69}$$

In the limit $t \to \infty$, the momentum-dependent prefactor becomes

$$\lim_{t \to \infty} \left(\frac{1 - \rho^{2n}}{1 + \rho^{2n}} \cdot \frac{1 - \rho}{1 + \rho}\right) = \frac{1 - \rho}{1 + \rho} \implies \lim_{t \to \infty} \mathrm{Var}\left[\zeta^*\right] = \frac{1 - \rho}{1 + \rho} \cdot \mathrm{Var}\left[\theta^*\right]. \tag{70}$$

Equations 69 and 70 validate our intuition. When $\rho \to 0$, then $\zeta$ behaves like $\theta$ independent of $T$, with their variance and expectation matching. When $\rho > 0$, the momentum-dependent prefactor serves as an aggregator over the history when $t$ is sufficiently large compared to $k$, reducing the variance $\mathrm{Var}[\zeta^*]$ but preserving its expectation. This formalizes the notion of time horizon discussed in Table 1.

# F  Additional details and results for Polyak-Ruppert averaging

**Additional background**  Polyak-Ruppert averaging (Definition 3.1) is a simplification of Stochastic Weight Averaging (SWA) (Izmailov et al., 2018) which uses a more complex multi-cycle schedule based weighting of the model parameters. Both Definition 3.1 and SWA present similar favorable properties like wider minima and better generalization (Izmailov et al., 2018). For example, He et al. (2022) observed that a supervised ViT-H/14 overfits on ImageNet1k (Russakovsky et al., 2014) without a model EMA, achieving an accuracy of 80.9%. Equipping a Polyak-Ruppert average ($\rho = 0.9999$) alleviated overfitting and gave a 83.1% accuracy.

**Organization** In this appendix, we look at additional momenta for one-dimensional noisy parabola, as well as extensions to $D$-dimensions (Appendix F.1), provide a more detailed view of the results of Section 3.2 (Appendix F.2), and investigate the scenario where the EMA Scaling Rule (Definition 1.2) is applied to batch normalization (Ioffe & Szegedy, 2015) coefficients (Appendix F.3).

## F.1 Noisy parabola

**Additional one-dimensional examples** First we consider additional one-dimensional examples, investigating the effect of modifying the base momentum $\rho_B$. We present $\rho_B = 0.99$ in Figure 7, and $\rho_B = 0.999$ in Figure 8. The results for $\rho_B = 0.9999$ are presented in main text in Figure 1.

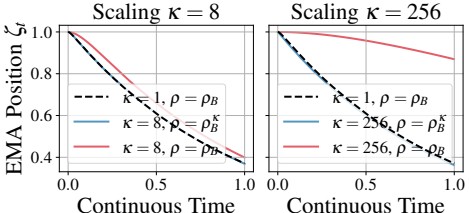 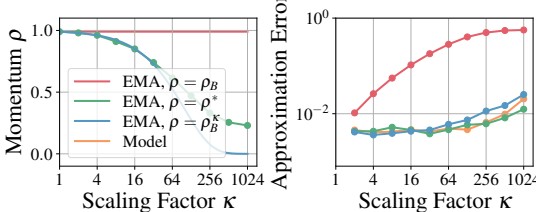

(a) Trajectory of the model EMA $\zeta$ under different scalings $\kappa$, with $\rho_B = 0.99$, $\eta_B = 10^{-4}$.

(b) Choices for momentum (left) with corresponding approximation errors (Equation 10) (right).

Figure 7: (a) We show the effect of scaling by comparing model EMA trajectories of the baseline ($\kappa = 1$, black dashed) to $\kappa = 8$ (left) and $\kappa = 256$ (right), with ($\rho = \rho_B^\kappa$, blue) and without ($\rho = \rho_B$, red) the EMA Scaling Rule. (b, left) The momentum according for different scaling rules and the empirically optimal $\rho^*$ (Equation 10). (b, right) The approximation error (Equation 10) of trajectories in (b, left) and the target model (orange). Error for $\rho^*$ is computed using a hold-out to mitigate overfitting.

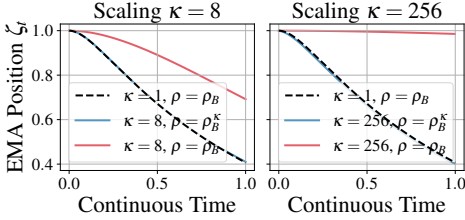 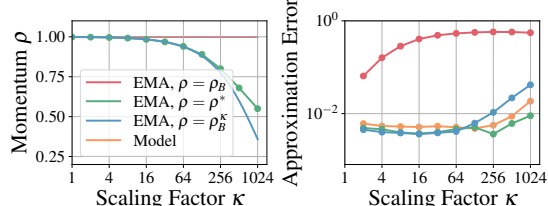

(a) Trajectory of the model EMA $\zeta$ under different scalings $\kappa$, with $\rho_B = 0.999$, $\eta_B = 10^{-4}$.

(b) Choices for momentum (left) with corresponding approximation errors (Equation 10) (right).

Figure 8: (a) We show the effect of scaling by comparing model EMA trajectories of the baseline ($\kappa = 1$, black dashed) to $\kappa = 8$ (left) and $\kappa = 256$ (right), with ($\rho = \rho_B^\kappa$, blue) and without ($\rho = \rho_B$, red) the EMA Scaling Rule. (b, left) The momentum according for different scaling rules and the empirically optimal $\rho^*$ (Equation 10). (b, right) The approximation error (Equation 10) of trajectories in (b, left) and the target model (orange). Error for $\rho^*$ is computed using a hold-out to mitigate overfitting.

As described by the scaling error term in Equation 54, the approximation error at a given $\kappa$ is higher for lower momenta $\rho$. For a large range of scalings $\kappa$, the EMA Scaling Rule and the optimal momenta $\rho^*$ are consistent. In summary, we see the synthetic experiments validate the results of Section 3.1 for a range of momenta $\rho$.

**Examples in higher dimensions** Our final use of the synthetic *noisy* parabola will consider an extension to $D$ dimensions. Consider the optimization of $\theta \in \mathbb{R}^D$ in a *noisy parabola* at the origin:

$$\mathcal{L}(\theta) = \frac{a}{2} \theta^\top \theta, \qquad \theta_{k+1} = \theta_k - \eta \, g_k, \qquad g_k = a \, \theta_k + \epsilon_k, \qquad \epsilon_k \sim \mathcal{N}\left(0, \frac{b \, g_k^2 + c}{\kappa}\right), \qquad (71)$$

for curvature $a > 0$, scaled additive $b > 0$, and additive $c > 0$ noise coefficients. The scaling factor $\kappa$ in the covariance denominator implements the reduction in gradient noise as the scaling (i.e., the batch size) increases (Jastrzebski et al., 2017). Let $\theta \in \mathbb{R}^D$ be optimized with SGD (Definition 2.1)

and let there be a Polyak-Ruppert average (Definition 3.1) $\zeta \in \mathbb{R}^D$ with momentum $\rho = 1 - \beta$ for $\theta$. We consider dimensionalities $D = 2$ (Figure 9), $D = 16$ (Figure 10), and $D = 100$ (Figure 11). We observe no significant differences in the EMA scaling behavior as we vary dimensions.

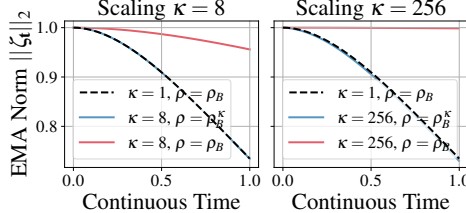 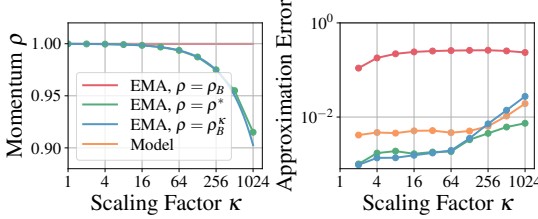

(a) Norm of the model EMA $\zeta$ under different scalings $\kappa$, with $\rho_B = 0.9999$, $\eta_B = 10^{-4}$, $D = 2$.

(b) Choices for momentum (left) with corresponding approximation errors (Equation 10) (right).

Figure 9: (a) We show the effect of scaling by comparing model EMA trajectories of the baseline ($\kappa = 1$, black dashed) to $\kappa = 8$ (left) and $\kappa = 256$ (right), with ($\rho = \rho_B^\kappa$, blue) and without ($\rho = \rho_B$, red) the EMA Scaling Rule. (b, left) The momentum according for different scaling rules and the empirically optimal $\rho^*$ (Equation 10). (b, right) The approximation error (Equation 10) of trajectories in (b, left) and the target model (orange). Error for $\rho^*$ is computed using a hold-out to mitigate overfitting.

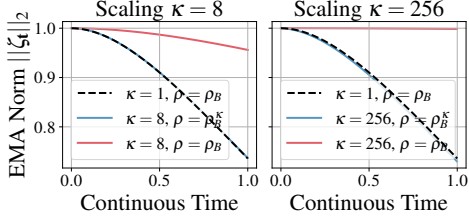 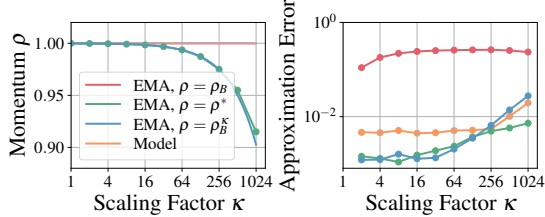

(a) Norm of the model EMA $\zeta$ under different scalings $\kappa$, with $\rho_B = 0.9999$, $\eta_B = 10^{-4}$, $D = 16$.

(b) Choices for momentum (left) with corresponding approximation errors (Equation 10) (right).

Figure 10: (a) We show the effect of scaling by comparing model EMA trajectories of the baseline ($\kappa = 1$, black dashed) to $\kappa = 8$ (left) and $\kappa = 256$ (right), with ($\rho = \rho_B^\kappa$, blue) and without ($\rho = \rho_B$, red) the EMA Scaling Rule. (b, left) The momentum according for different scaling rules and the empirically optimal $\rho^*$ (Equation 10). (b, right) The approximation error (Equation 10) of trajectories in (b, left) and the target model (orange). Error for $\rho^*$ is computed using a hold-out to mitigate overfitting.

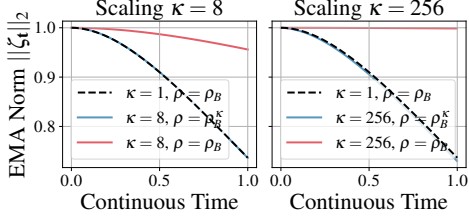 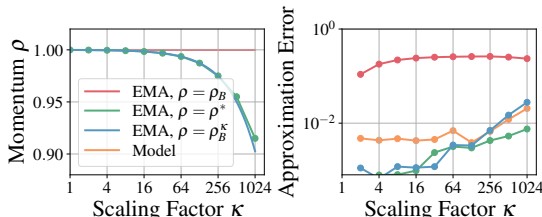

(a) Norm of the model EMA $\zeta$ under different scalings $\kappa$, with $\rho_B = 0.9999$, $\eta_B = 10^{-4}$, $D = 100$.

(b) Choices for momentum (left) with corresponding approximation errors (Equation 10) (right).

Figure 11: (a) We show the effect of scaling by comparing model EMA trajectories of the baseline ($\kappa = 1$, black dashed) to $\kappa = 8$ (left) and $\kappa = 256$ (right), with ($\rho = \rho_B^\kappa$, blue) and without ($\rho = \rho_B$, red) the EMA Scaling Rule. (b, left) The momentum according for different scaling rules and the empirically optimal $\rho^*$ (Equation 10). (b, right) The approximation error (Equation 10) of trajectories in (b, left) and the target model (orange). Error for $\rho^*$ is computed using a hold-out to mitigate overfitting.

**Compute** The compute usage for the noisy parabola experiments is relatively small, with each run taking less than one minute on a single CPU, and so we do not detail this compute usage as we do in the other experimental sections.

Table 4: Supervised ResNetv2-50 hyperparameters used in Polyak-Ruppert Averaging experiments.

| | Supervised ResNetv2-50 |
|---|---|
| ImageNet1k Test Top-1 | 76.27 ± 0.10% |
| ImageNet1k EMA Test Top-1 | 76.55 ± 0.07% |
| Weight initialization | `kaiming_normal(relu)` |
| Backbone normalization | BatchNorm |
| Synchronized BatchNorm over replicas | No |
| Learning rate schedule | Multi step: ×0.1 at [30, 60, 80] epochs |
| Learning rate warmup (epochs) | 5 |
| Learning rate minimum value | $1 \times 10^{-6}$ |
| Training duration (epochs) | 90 |
| Optimizer | SGD + Momentum |
| SGD momentum | 0.9 |
| Optimizer scaling rule | Linear |
| Base learning rate | 0.4 |
| Base batch size | 1024 |
| Base Polyak momentum | 0.9999 |
| Weight decay | $1 \times 10^{-4}$ |
| Weight decay scaling rule | None |
| Weight decay skip bias | Yes |
| Numerical precision | `bf16` |
| Augmentation stack | ImageNet |
| Label smoothing rate | 0.1 |

## F.2 Image Classification

**Hyperparameters**   We present the base hyperparameters for our image experiments in Table 4.

**Data**   For large scale vision evaluation, we use the ImageNet1k dataset (Russakovsky et al., 2014), a widely used dataset containing approximately 1.2 million labeled images, distributed almost uniformly across 1000 different object classes, like animals, plants, and vehicles.

The images in ImageNet1k are are not consistent in resolution. To handle this, they are resized and cropped to a standard size (in our case, $224 \times 224$), before further processing. This is part of the standard ImageNet augmentation stack for convolutional networks mentioned in Table 4.

**Compute usage**   The compute usage image classification Polyak-Ruppert averaging is summarized in Table 5.

Table 5: Compute usage for image classification Polyak-Ruppert averaging in Figures 2 and 13. The three runs for the batch size 1,024 baseline correspond to three seeds, and the nine runs for all other batch sizes correspond to using and not using the EMA Scaling Rule shown in Figure 2, and its application to Batch Normalization shown in Figure 13. All experiments conducted are using 80Gb A100s.

| Batch Size | GPUs | Time (h) | Compute/Run (GPUh) | Runs | Compute (GPUh) |
|---|---|---|---|---|---|
| 512 | 8 | 35.3 | 282.4 | 9 | 2,541.6 |
| 1,024 | 8 | 17.1 | 137.0 | 3 | 410.9 |
| 2,048 | 8 | 13.3 | 106.7 | 9 | 960.6 |
| 4,096 | 8 | 4.2 | 33.5 | 9 | 301.9 |
| 8,192 | 16 | 2.8 | 44.8 | 9 | 403.6 |
| All other compute, e.g. code development, runs with errors, and debugging | | | | | 25,768.3 |
| **Total** | | | | | **30386.8** |

**Additional results**   In Figure 12 we present a more detailed view of the results in Section 3.2. First, we see that for all train metrics, model trajectories match, and that a learning rate step schedule after warmup is present. As discussed in Figure 12, a gap in EMA Test Top-1 trajectories begins at scaling $\kappa = 4$, with a more pronounced effect visible at $\kappa = 8$. From Figure 12 it is clear that the (non-EMA)

Test Top-1 performance trajectory is no longer matching at these scalings, demonstrating that the problem is not due to a breakdown of the EMA Scaling Rule, but rather, that the model is overfitting at larger batch sizes due to batch normalization (Ioffe & Szegedy, 2015).

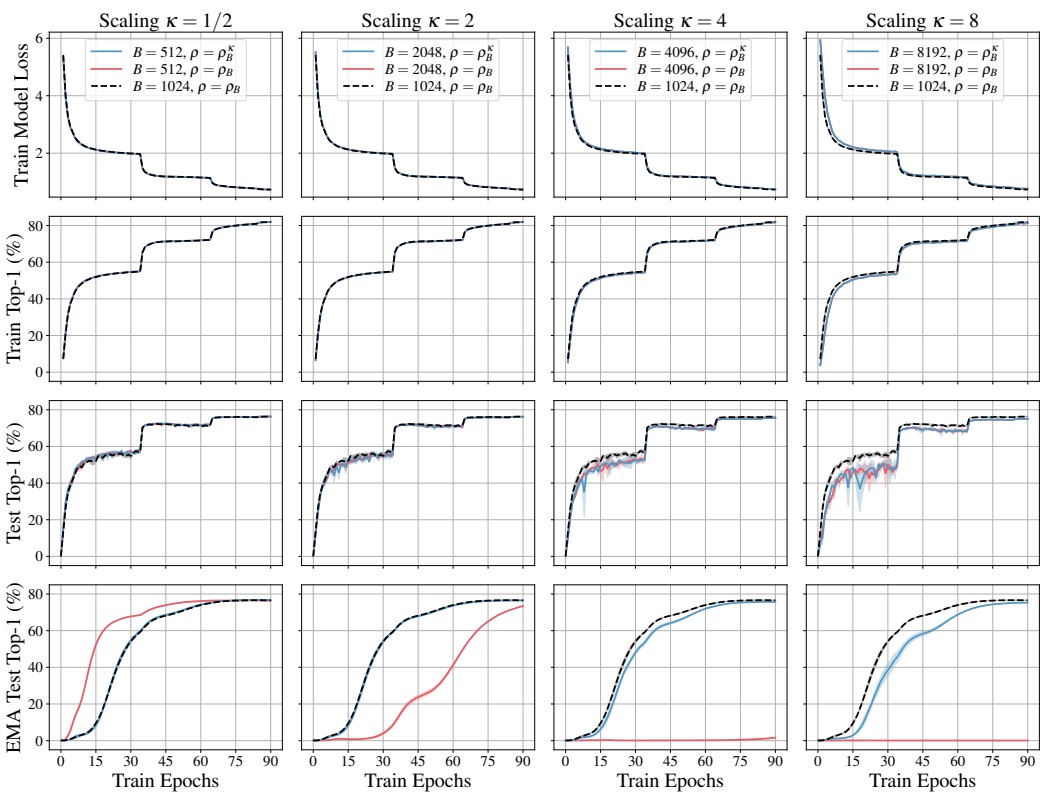

Figure 12: *ResNetv2-50 Polyak-Ruppert averaging on ImageNet1k* for different scalings $\kappa$. The baseline model ($\kappa = 1$, black dashed) uses batch size 1024 and momentum $\rho_B = 0.9999$, is scaled down to a batch size of 512 (left), and up to a batch size of 4096 (right) with (blue, $\rho = \rho_B^\kappa$) and without (red, $\rho = \rho_B$) the EMA Scaling Rule (Definition 1.2). Bands indicate the mean and standard deviation across three runs.

## F.3    Applying the EMA Scaling Rule to Batch Normalization

In Section 3.2 and Appendix F.2, we investigated a range of scalings $\kappa$, *with* and *without* applying the EMA Scaling Rule to the Polyak momentum. In those experiments, we maintained Batch Normalization (Ioffe & Szegedy, 2015) coefficients of $\rho_{BN} = 0.9$ throughout[11], i.e. the EMA Scaling Rule is not applied. The running statistics of Batch Normalization *are* an EMA with values determined by $\rho_{BN}$ and so it is reasonable to suspect we should apply the EMA Scaling Rule to $\rho_{BN}$ also.

In Figure 13 we investigate the effect of applying the EMA Scaling Rule to Batch Normalization coefficients, using $\hat{\rho}_{BN} = \rho_{BN}^\kappa$. We observe that the Test Top-1 trajectories *with* the EMA Scaling Rule applied are slightly closer to the reference trajectories for scalings $\kappa \geq 2$ than those trajectories *without* the EMA Scaling Rule. As the effect is not particularly large, at least in this setup, we do pursue further ablating applications of the EMA Scaling Rule to batch normalization coefficients, and always use $\rho_{BN} = 0.1$ for Batch Normalization, independent of $\kappa$.

---

[11] In many ML frameworks, this value is defined using $\beta_\rho = 1 - \rho$, i.e. the default is 0.1 and corresponds to $\beta_{BN}$ rather than 0.9 corresponding to $\rho_{BN}$. We use $\rho_{BN}$ to maintain consistency across this work.

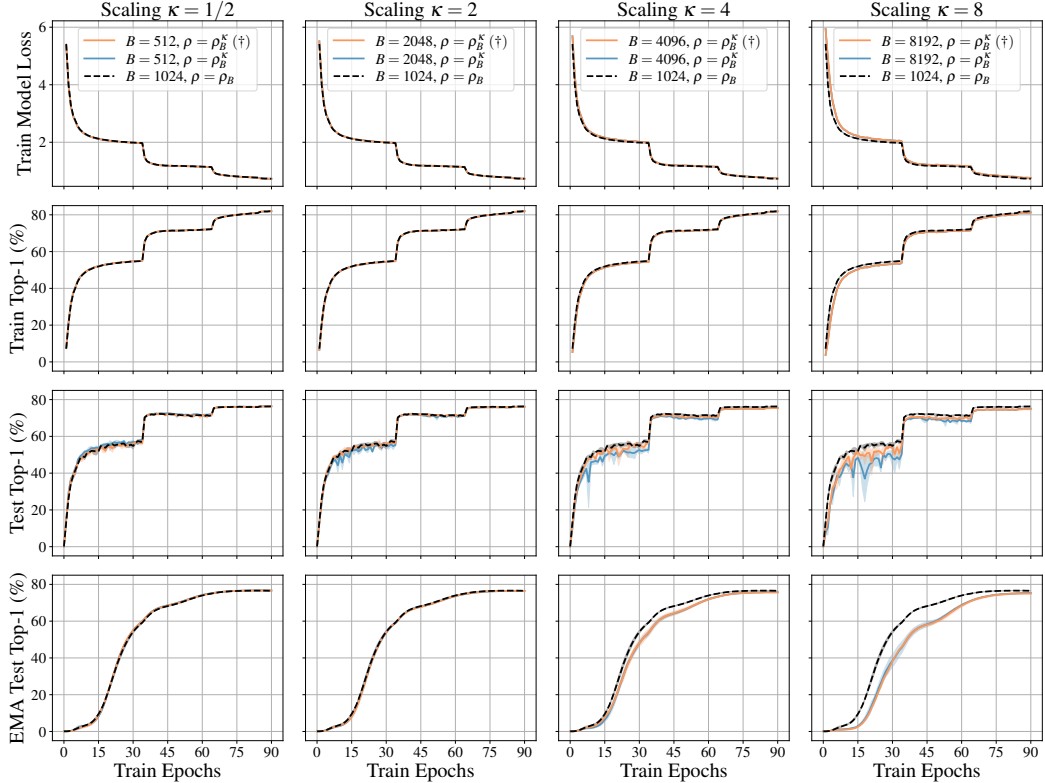

Figure 13: *ResNetv2-50 Polyak-Ruppert averaging on ImageNet1k for different scalings $\kappa$*. The baseline model ($\kappa = 1$, black dashed) uses batch size 1024 and momentum $\rho_B = 0.9999$, is scaled down to a batch size of 512 (left), and up to a batch size of 4096 (right) with the EMA Scaling Rule applied to *only* model parameters (blue, $\rho = \rho_B^\kappa$), and model parameters *and* buffers (orange, $\rho = \rho_B^\kappa$ (†)). Bands indicate the mean and standard deviation across three runs.

## G   Additional details and results for Automatic Speech Recognition (ASR)

In this section we provide additional details for the speech recognition experiments in both the supervised and semi-supervised case.

**Data**   We use the LibriSpeech dataset (Panayotov et al., 2015) which is a dataset of audio-transcription pairs. For supervised Polyak-Ruppert averaging experiments, we use *train-clean-100* as training data, and for semi-supervised pseudo-labeling experiments, we use *train-clean-100* as the labeled and *train-clean-360* and *train-other-500* as the unlabeled data. The standard LibriSpeech validation sets (*dev-clean* and *dev-other*) are used to tune all hyperparameters, as well as to select the best models. Test sets (*test-clean* and *test-other*) are only used for reporting final model performance, measured in WER without an external language model. We maintain the original 16kHz sampling rate, and compute log-mel filterbanks with 80 coefficients for a 25ms sliding window, strided by 10ms, later normalized to zero mean and unit variance for each input sequence.

**Acoustic model**   We employ a vanilla encoder-based only transformer model trained with the Connectionist Temporal Classification (CTC) loss (Graves et al., 2006). We use the training configuration from Likhomanenko et al. (2021a), which has three stages: i) 1D convolutions to perform striding (kernel of 7 with stride of 3); ii) a Transformer encoder with 36 layers, post-LayerNorm, four attention heads, an embedding dimension of 768, an MLP dimension of 3072, a dropout frequency of 0.3, and a layer drop frequency of 0.3; and iii) a linear layer to map to the target vocabulary[12]. To reduce model training time by a factor of approximately $2-3\times$, and to reduce memory footprint,

---

[12]The token set of this vocabulary consists of the 26 English alphabet letters augmented with the apostrophe and a word boundary token.

Table 6: Hyperparameters summary for speech recognition task for supervised (left) and semi-supervised pseudo-labeling (right) training with a vanilla transformer. The $0.3 \to 0.1$ in the dropout and layer drop rates indicates that a rate of 0.3 is used during pre-training, and a rate of 0.1 is used during pseudo-labeling.

| | Supervised | Pseudo-Labeling |
|---|---|---|
| Librispeech test-clean / test-other WER | 7.8/19.1 | 4.8/11.5 |
| Optimizer | Adam | Adam |
| Optimizer scaling rule | Adam | Adam |
| Base $(\beta_1, \beta_2)$ | (0.995, 0.999) | (0.995, 0.999) |
| Base learning rate | 0.0001 | 0.0001 |
| Base learning rate warmup (steps) | 64k | 64k |
| Learning rate schedule | Fixed (no decay) | Fixed (no decay) |
| Learning rate minimum value | 0 | 0 |
| Base training duration (steps) | 400k | 500k |
| Base batch size (dynamic) | $8 \times 290s$ | $8 \times 290s$ |
| Base teacher momentum | 0.99995 | 0.9999 |
| Weight decay | None | None |
| Numerical precision | bf16 | bf16 |
| Augmentation stack | SpecAug | SpecAug |
| Dropout | 0.3 | $0.3 \to 0.1$ |
| Layer drop | 0.3 | $0.3 \to 0.1$ |
| Gradient clipping | 1 | 1 |
| Labeled:unlabeled data ratio | N/A | 1:3 |
| Base pre-training steps | N/A | 20k |
| Base start of EMA accumulation (steps) | N/A | 19k |

we use CAPE positional embeddings (Likhomanenko et al., 2021b) instead of relative positional embeddings (Shaw et al., 2018): both models perform similarly.

**Training** Here we discuss our training procedure for base batch size $B = 8 \times 290s$, which is adapted from Likhomanenko et al. (2021a), and is summarized in Table 6. We use SpecAugment (Park et al., 2019) activated after 5k steps of training: two frequency masks with frequency mask parameter $F = 30$, ten time masks with maximum time-mask ratio $p = 0.1$ and time mask parameter $T = 50$ are used; time warping is not used.

One difference in setup is we use the Adam optimizer, whereas Likhomanenko et al. (2021a) used Adagrad (Duchi et al., 2010). Even though Adagrad can be viewed as a particular limit ($\beta_1 = 0$ and $\beta_2 \to 1$) of Adam (Kingma & Ba, 2015), we were unable to produce reasonable optimization in practice when applying the Adam Scaling Rule of Malladi et al. (2022) in this limit. As a consequence, we chose to work with the Adam optimizer, where its scaling rule has been shown to work (Malladi et al., 2022), and we take $\beta_1 = 0.995$, $\beta_2 = 0.999$, and $\epsilon = 10^{-8}$. We obtained similar results for $\beta_1 = 0.99$. Finally, we use a linear learning rate warmup (64k steps) after which the learning rate is kept constant until convergence. This performance can be improved further by using a step decay schedule as shown in prior work. We also apply gradient clipping of 1, and do not use weight decay.

**Pseudo-Labeling** The pseudo-labeling process comprises of two stages: i) The pre-training phase, where we train model on labeled data for 20k steps with model EMA accumulation starting after 19k steps; and ii) the pseudo-labeling phase, where we involve unlabeled data by generating pseudo-labels from the model EMA (teacher) and provide them to the model (student) as if they were ground-truth labels. Pseudo-labels are generated without any dropout applied to the teacher, and no data augmentation is applied for the corresponding inputs. To produce the pseudo-label, we use *hard transcription* (Definition G.1)

**Definition G.1** (Hard Transcription). *For a sequence of frames, select the most probable token per frame, removing repetitions and the CTC blank token. For example, "h##eelll##ll###oo" is transformed into "hello", where "#" is the CTC blank token.*

These hard transcriptions are then used as transcription for student optimization. We use a 1:3 proportion of labeled to unlabeled data as this was found to be optimal in Likhomanenko et al. (2021a), and we decrease model dropout and layer drop rates to 0.1 after pre-training phase. As we have access to the ground-truth labels on the data being treated as unlabeled, we can track

pseudo-label quality by computing pseudo-labels on this data, and compute the WER against their ground-truth. Pseudo-label quality is the primary metric to evaluate progress on unlabeled data, as loss on pseudo-labeled data is unreliable when a teacher model and pseudo-labels are evolving with each time step.

**Scaling of batch size** Sequential data is typically processed using dynamic batching as it is more computationally efficient than using a fixed number of sequences (Ott et al., 2019). In our work, we use dynamic batching of ~290s audio per GPU. Moreover, for CTC we do not apply any additional sequence normalization. We experimented with fixed batching, but did not observe any significant differences in conclusions compared with the dynamic batching.

We note that dynamic batching is a more challenging setting for achieving systematic scaling, as the number of independent sequences in any given batch may change, and the i.i.d. assumption does not hold at the frame level. Despite these violations of the assumptions of Section 2.2, our results demonstrate that the Adam Scaling Rule (Definition C.3, Malladi et al. (2022)) holds in the case of dynamic batches, as does our EMA Scaling Rule (Definition 1.2).

The base batch size is set to $B = 8 \times 290s$, and in our experiments we scale down to batch size of $B = 2 \times 290s$ and up to batch size of $B = 128 \times 290s$. The number of warmup and pre-training steps, steps before SpecAugment is turn on and model EMA is accumulated are scaled according to Appendix C.1.

**Compute** All experiments are done using A100 80GB 8GPU nodes with `bfloat16` precision training. While for supervised training evaluation of different EMA decay values is done in parallel during a single run, for pseudo-labeling every EMA decay value needs separate training. Final models training compute is detailed in Tables 7 and 8. Total compute, including e.g. code development, runs with errors, and debugging, is **61k** GPUh.

Table 7: Compute usage for supervised model for speech recognition task in Figure 14. Values *include* node allocation times (typically a small % of corresponding total runtime), giving a practical estimate of reproduction cost. All experiments conducted are using 80Gb A100s with fast interconnect.

| Batch Size | GPUs | Time (h) | Compute/Run (GPUh) | Runs | Compute (GPUh) |
|---|---|---|---|---|---|
| $2 \times 290s$ | 2 | 222 | 444 | 1 | 222 |
| $4 \times 290s$ | 4 | 108 | 432 | 1 | 432 |
| $8 \times 290s$ | 8 | 64 | 512 | 1 | 512 |
| $16 \times 290s$ | 16 | 54 | 864 | 1 | 896 |
| $32 \times 290s$ | 32 | 37 | 1,184 | 1 | 1,184 |
| **Total** | | | | | **3,436** |

Table 8: Compute usage for continuous pseudo-labeling for the speech recognition task in Figure 16. Values *include* node allocation times (typically a small % of corresponding total runtime), giving a practical estimate of reproduction cost. All experiments conducted are using 80Gb A100s with fast interconnect.

| Batch Size | GPUs | Time (h) | Compute/Run (GPUh) | Runs | Compute (GPUh) |
|---|---|---|---|---|---|
| $2 \times 290s$ | 2 | 225 | 550 | 2 | 1,110 |
| $4 \times 290s$ | 4 | 120 | 480 | 2 | 960 |
| $8 \times 290s$ | 8 | 72 | 576 | 1 | 576 |
| $16 \times 290s$ | 16 | 45 | 720 | 2 | 1,440 |
| $32 \times 290s$ | 32 | 33 | 1,056 | 4 | 4,224 |
| $64 \times 290s$ | 64 | 25 | 1,600 | 2 | 3,200 |
| **Total** | | | | | **11,510** |

### G.1 Additional experimental settings and detailed metrics

We present detailed comparison between models trained with and without EMA Scaling Rule in Figures 14 and 15 for supervised training and in Figures 16 and 17 for semi-supervised training.

First, we observe that if the Adam Scaling Rule does not hold perfectly[13] (there is a mismatch between trajectories for the model before pseudo-labels are involved) the EMA Scaling Rule also

---

[13]See Malladi et al. (2022) for a discussion on scenarios that lead to a breakdown of the Adam Scaling Rule.

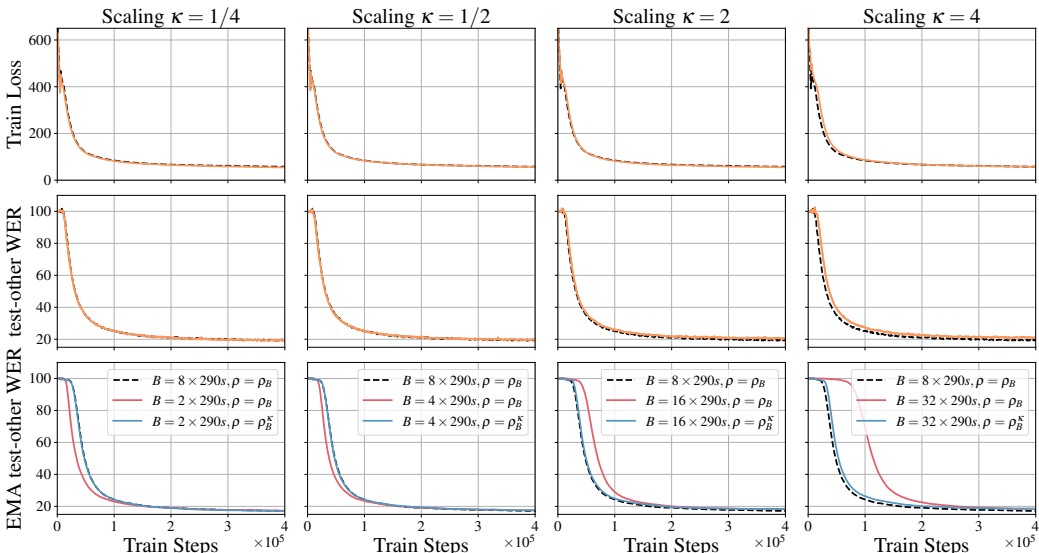

Figure 14: *Transformer Polyak-Ruppert averaging on LibriSpeech (trained on train-clean-100)* with different scalings $\kappa$. The baseline ($\kappa = 1$, black dashed) is trained with Adam and momentum $\rho_B = 0.99995$ at a *dynamic batch size $B = 8 \times 290s$*, which corresponds to a single train step on the $x$-axis. We investigate dynamic batch sizes down to $B = 2 \times 290s$ (left) and up to $B = 32 \times 290s$ (right), with (blue, $\rho = \rho_B^\kappa$), and without (red, $\rho = \rho_B$) the EMA Scaling Rule (model non-EMA is marked by orange). The Adam Scaling Rule (Malladi et al. (2022), Definition C.3) is used throughout. For momentum $\rho_B = 0.9999$ we observe similar trajectories for all models.

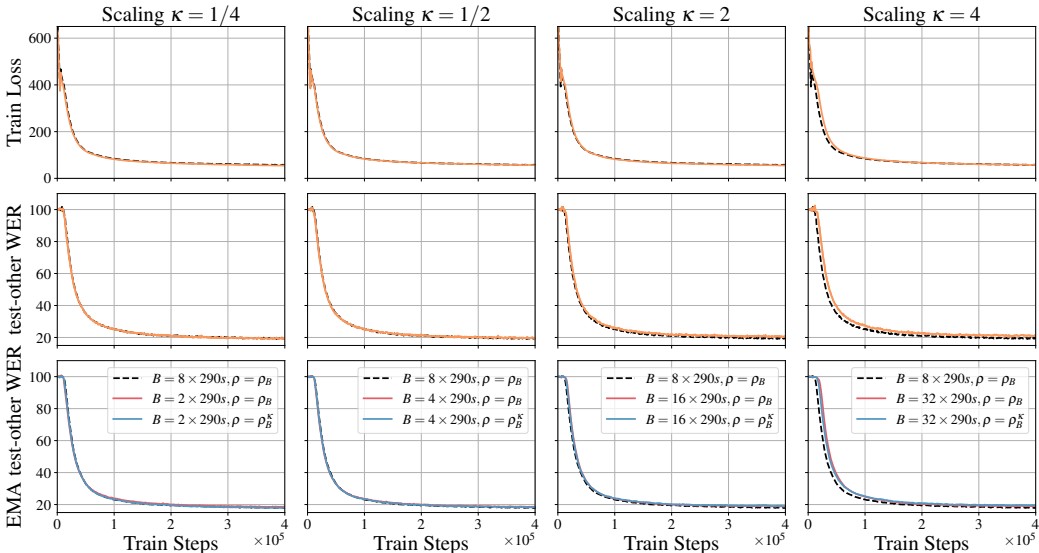

Figure 15: *Transformer Polyak-Ruppert averaging on LibriSpeech (trained on train-clean-100)* with different scalings $\kappa$. The baseline ($\kappa = 1$, black dashed) is trained with Adam and momentum $\rho_B = 0.999$ at a *dynamic batch size $B = 8 \times 290s$*, which corresponds to a single train step on the $x$-axis. We investigate dynamic batch sizes down to $B = 2 \times 290s$ (left) and up to $B = 32 \times 290s$ (right), with (blue, $\rho = \rho_B^\kappa$), and without (red, $\rho = \rho_B$) the EMA Scaling Rule (model non-EMA is marked by orange). The Adam Scaling Rule (Malladi et al. (2022), Definition C.3) is used throughout. If momentum $\rho_B$ is small and accumulation history is short we observe no any significant difference between models which all are matching the reference trajectory despite scaling $\kappa$.

gives discrepancies with the reference trajectory, however they are negligible compared to models trained without EMA Scaling Rule. For the semi-supervised training, to alleviate the difficulties with a breakdown of the Adam Scaling Rule for large $\kappa$ we postpone the pseudo-labeling process until the model reaches similar WER as the baseline. This allows us to align the initial model conditions for pseudo-labeling. In this scenario we are able to match the reference trajectory up to $\kappa = 8$.

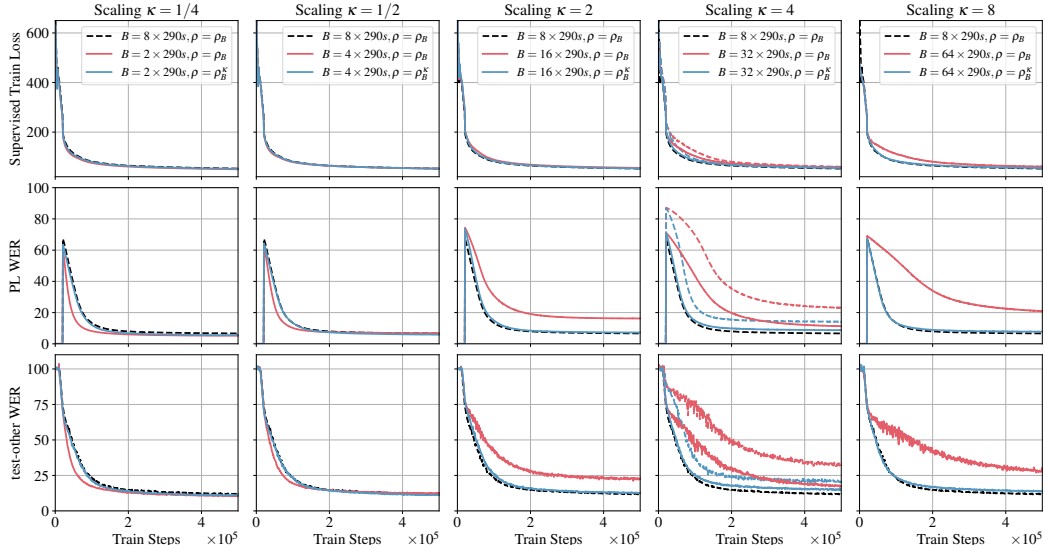

Figure 16: *Transformer pseudo-labeling on LibriSpeech (trained on train-clean-100 as labeled and the rest of LibriSpeech as unlabeled)* with different scalings $\kappa$. The baseline ($\kappa = 1$, black dashed) is trained with Adam at a *dynamic batch size* of $8 \times 290$ seconds, which corresponds to a single train step on the $x$-axis. The model EMA (*teacher*) is updated with momentum $\rho_B = 0.9999$. We investigate dynamic batch sizes down to $B = 2 \times 290s$ (left) and up to $B = 64 \times 290s$ (right), with (blue, $\rho = \rho_B^\kappa$) and without (red, $\rho = \rho_B$) the EMA Scaling Rule. The Adam Scaling Rule (Malladi et al. (2022), Definition C.3) is used throughout. For $\kappa \leq 2$, we start pseudo-labeling after 20k/$\kappa$ training steps; while for $\kappa > 2$, we start when pre-training WER matches the baseline WER (24k/$\kappa$ for $\kappa = 4$ and 29k/$\kappa$ for $\kappa = 8$). For $\kappa = 4$ we experimented with both variants: we start pseudo-labeling after 20k/$\kappa$ (dashed) and when pre-training WER matches the baseline WER (solid, 24k/$\kappa$).

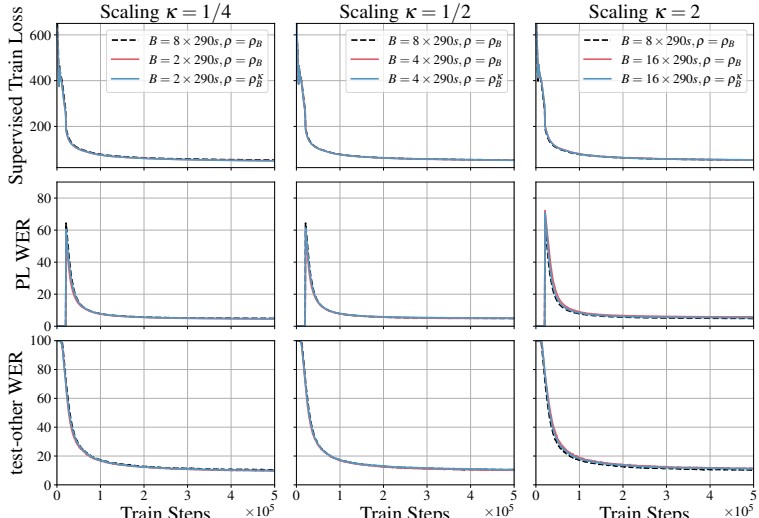

Figure 17: *Transformer pseudo-labeling on LibriSpeech (using train-clean-100 as labeled)* with different scalings $\kappa$. The baseline ($\kappa = 1$, black dashed) is trained with Adam at a *dynamic batch size* of $8 \times 290$ seconds, which corresponds to a single train step on the $x$-axis. The model EMA (*teacher*) is updated with momentum $\rho_B = 0.999$. We investigate dynamic batch sizes down to $B = 2 \times 290s$ (left) and up to $B = 16 \times 290s$ (right), with (blue, $\rho = \rho_B^\kappa$) and without (red, $\rho = \rho_B$) the EMA Scaling Rule. The Adam Scaling Rule is used throughout.

We note that this result reveals that errors for the Adam Scaling Rule *and* the EMA Scaling Rule are contributing, although the way in which they contribute is different, and one can dominate the other. We observe in Figure 16 that if the initial conditions of the models are similar (attained by using the same WER as a condition to begin pseudo-labeling) then the error from the EMA Scaling Rule dominates over that of the Adam Scaling Rule, causing a divergence in training dynamics.

Second, we observe in practice that the EMA Scaling Rule holds for both fixed batching (a sequence length in the batch can vary significantly) and for dynamic batching (when total number of frames in the batch is fixed, though padding still is accounted to the this amount). This shows that EMA Scaling Rule is applicable to sequential data too.

Third, we observe in Figures 15 and 17 that for smaller values of $\rho_B$, scaling with or without EMA Scaling Rule behave similarly, and reference trajectories match in the supervised and semi-supervised cases. However, if the momentum is too large, the *teacher* moves slowly and is uninformative, whereas if the momentum is too low, the *teacher* and the *student* are effectively be the same model, implying: i) the student will self-predict with high confidence, removing any benefits of distillation[14]; and ii) training instability or model divergence will happen in the low-resource settings (Likhomanenko et al., 2021a; Higuchi et al., 2022).

### G.2   Scaling to $\kappa = 16$ with Progressive Scaling

Finally, we aim to scale semi-supervised pseudo-labeling further to $\kappa = 16$. In this case we observe that Adam Scaling Rule does not hold in the pre-training phase and there is no model convergence. To overcome this, we apply Progressive Scaling (Definition 3.2). We pre-train models on supervised data with $\kappa = 8$ for 29k of reference steps (model EMA accumulation starts at 28k steps). We then scale to $\kappa = 16$ and begin pseudo-labeling. We see in Figure 18 that Progressive Scaling enables us to scale pseudo-labeling to $\kappa = 16$ with (middle) and without (left) the EMA Scaling Rule. Second, models *with* the EMA Scaling Rule track the baseline much closer than models without the EMA Scaling Rule, although a small gap is present. We further experimented with Progressive Scaling, postponed the transition condition to the $\kappa = 16$ until 75k reference steps. In Figure 18 (right), we see this scaled model tracks the reference trajectory, and so using a combination of the EMA Scaling Rule and Progressive Scaling, we are able to scale pseudo-labeling to $\kappa = 16$, corresponding to a dynamic batch size of $128 \times 290s$.

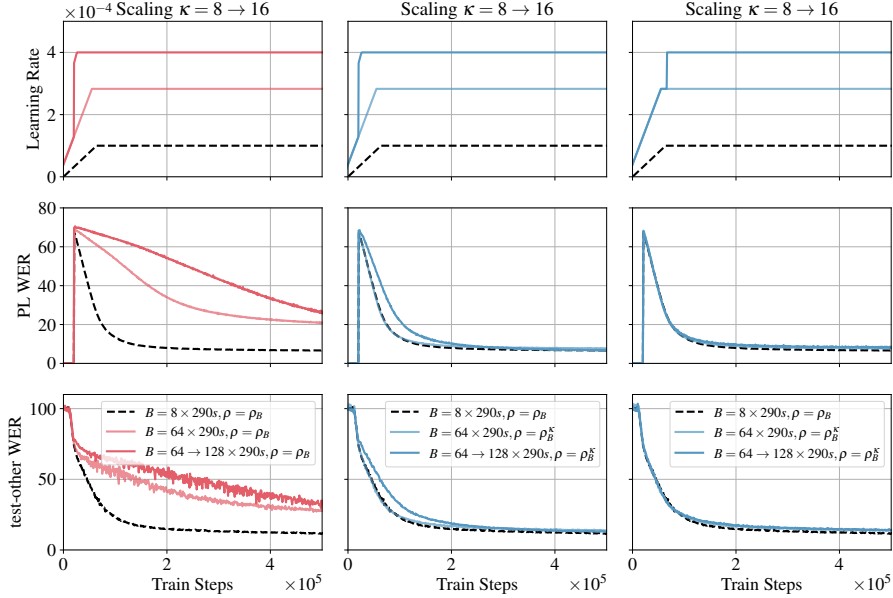

Figure 18: *Transformer pseudo-labeling on LibriSpeech (trained on train-clean-100 as labeled and the rest of LibriSpeech as unlabeled)* with different Progressive Scaling from $\kappa = 8$ to $\kappa = 16$ ($\kappa = 8 \rightarrow 16$). The baseline ($\kappa = 1$, black dashed) is trained with Adam at a *dynamic batch size* of $8 \times 290$ seconds, which corresponds to a single train step on the *x*-axis. The model EMA (*teacher*) is updated with momentum $\rho_B = 0.9999$. The scaling with $\kappa = 8$ is shown with lighter color for reference from Figure 16. We investigate dynamic batch sizes progressively from $B = 64 \times 290s$ to $B = 128 \times 290s$, with (blue, $\rho = \rho_B^\kappa$) and without (red, $\rho = \rho_B$) the EMA Scaling Rule. For reference (top) we show the learning rate schedule with Progressive Scaling. The Adam Scaling Rule (Malladi et al. (2022), Definition C.3) is used throughout. Left and middle correspon to Progressive Scaling with scale from $\kappa = 8$ to $\kappa = 16$ at 29k steps, while right corresponds to 75k steps.

---

[14]He et al. (2020) alleviated the problem with the proper amount of noise during *student* model training, whilst Xu et al. (2020) used beam-search decoding with a language model.

# H  Additional details and results for self-supervised image representation learning

**Organization**  This appendix is structured into three sections. We first give an overview of our chosen SSL method BYOL (Appendix H.1), our recipe for training BYOL using ResNet 18s (Appendix H.2), our recipe for training BYOL using Vision Transformers (ViTs) (Appendix H.3), ablations of normalization approaches that lead to the development of this recipe (Appendix H.4), and additional results corresponding to longer training duration (Appendix H.5) and further understanding the impact of Progressive Scaling (Appendix H.6).

Second, we demonstrate that the EMA Scaling Rule combined with Progressive Scaling can scale a ResNet-50 BYOL model trained with LARS to batch size 32,768 without performance drop, demonstrating the empirical utility of the tools we provide outside of their theoretical validity (Appendix H.10).

Finally, we show that it is possible to systematically scale DINO (Caron et al., 2021) using a combination of Progressive Scaling and the EMA Scaling Rule, providing a solution for researchers and practitioners wanting to train DINO at scale.

## H.1  Components of self-supervised learning

First, a key component of many SSL methods is the *stop-gradient* or StopGrad (Definition H.1).

**Definition H.1** (Stop Gradient/StopGrad($\cdot$)). *The* stop-gradient *operator StopGrad($\cdot$) prevents the flow of gradient information*

$$\frac{df(StopGrad(h(x;\omega));\theta)}{d\omega} \equiv 0 \tag{72}$$

*for all parametric functions $h$ and $f$ and for all parameters $\theta$ and $\omega$.*

Applying a *stop-gradient* is sometimes called *detaching* in the literature. Now, we introduce the update rule of our representative SSL method BYOL in Definition H.2.

**Definition H.2** (BYOL Update). *BYOL learns unsupervised features by minimizing the cosine distance between the predictions of a student backbone $f(\cdot;\theta)$ (typically a ResNet or Vision Transformer), projected through $h(\cdot;\omega)$ (typically a Multi-Layer Perceptron (MLP)), and the predictions of an EMA teacher $f(\cdot;\zeta)$ (Grill et al., 2020). The update for the parameters of BYOL is then*

$$(\theta_{t+1}, \omega_{t+1}) = (\theta_t, \omega_t) - \eta \times \frac{1}{B} \sum_{x \in \mathbb{B}} \nabla_{(\theta,\omega)} \mathcal{L}(x; \theta_t, \omega_t, \zeta_t) \tag{73}$$

$$\zeta_{t+1} = \rho\, \zeta_t + (1 - \rho)\, \theta_{t+1} \tag{74}$$

$$with \quad \mathcal{L}(x; \theta_t, \omega_t, \zeta_t) = \frac{1}{2} \cos\left[h(f(x_1; \theta_t); \omega_t), StopGrad(f(x_2; \zeta_t))\right] + (x_1 \leftrightarrow x_2), \tag{75}$$

*where $\cos(a, b) \equiv 1 - a \cdot b/(||a||\,||b||)$ is the cosine distance, and $x_1$ and $x_2$ are two views of a single variate $x$, often produced by augmentations, and $x_1 \leftrightarrow x_2$ denotes symmetrization over $x_1$ and $x_2$.*

As noted in Section 3.4, the BYOL EMA update (Equation 74) uses $\theta_{t+1}$ instead of our analyzed $\theta_t$ (Equation 4). The effect upon the overall EMA update is $O(\eta \times \beta_\rho)$ and so is captured by the EMA Scaling Rule (Definition 1.2).

One more piece of technology typically employed in SSL is a *tracking probe* (Definition H.3) which we will use to evaluate the performance of BYOL on downstream tasks of interest, for example, image classification.

**Definition H.3** (Tracking Probe/Linear Probe). *When optimizing model parameters $\omega_t$ of an SSL method, simultaneously optimize the parameters $\xi$ of a probe model $r(\cdot;\xi)$ under a downstream objective $\mathcal{L}^{(d)}$. For example, in classification, with data $x$ and samples $y$*

$$\mathcal{L}^{(d)}(x, y, \theta_t, \xi_t) = -\log P(y|r(StopGrad(h(x; \omega_t)); \xi)) \tag{76}$$

$$\mathcal{L}^{(total)}(x, y; \theta_t, \omega_t, \zeta_t, \xi_t) = \mathcal{L}(x; \theta_t, \omega_t, \zeta_t) + \mathcal{L}^{(d)}(x, y, \omega_t, \xi_t), \tag{77}$$

*The is a probe for the teacher, which is typically the better choice due to Polyak-Ruppert averaging effects (see Section 3.2). When the $r$ is a linear model, the tracking probe is called a linear probe.*

Table 9: BYOL ResNet-18 hyperparameters for CIFAR10

|  | ResNet-18 |
|---|---|
| Weight initialization | `kaiming_uniform` (He et al., 2015) |
| Backbone normalization | BatchNorm |
| Head normalization | BatchNorm |
| Synchronized BatchNorm over replicas | Yes |
| Learning rate schedule | Single Cycle Cosine |
| Learning rate warmup (epochs) | 20 |
| Learning rate minimum value | 0 |
| Training duration (epochs) | 100 |
| Optimizer | SGD |
| Optimizer scaling rule | SGD |
| Optimizer momentum | 0.9 |
| Gradient clipping | 0.1 |
| Base learning rate | 0.02 |
| Base batch size | 1024 |
| Base teacher momentum | 0.992 |
| Weight decay | $1 \times 10^{-6}$ |
| Weight decay scaling rule | None |
| Weight decay skip bias | Yes |
| Numerical precision | `tf32` |
| Augmentation stack | `BYOL CIFAR10` |

It is also typical to use a Batch Normalization layer *without* trainable affine terms before this linear layer as in He et al. (2022) to stabilize probe training. In this case, the running statistics can be absorbed into a definition of the linear layer weights and biases, and so this is still a *linear probe*, although we will call this a *pre-bn linear probe* to remove ambiguity.

## H.2 A ResNet-18 recipe for BYOL

**Hyperparameters** We present the base hyperparameters for training BYOL with a ResNet-18 backbone using SGD in Table 9. This recipe was developed by starting from a well-known BYOL ResNet-50 recipe (Grill et al., 2020), adapting the input augmentations for CIFAR10, and performing a search over learning rate choices for an SGD optimizer.

## H.3 A Vision Transformer recipe for BYOL

**Hyperparameters** We present the base hyperparameters for training BYOL with a ViT-B/16 backbone in Table 10. This recipe was developed by starting from a well-known supervised ViT-B/16 recipe (He et al., 2022) and performing a search over weight decay and learning rate hyperparameter choices. We find that BYOL performs well with heavy weight decay ($\lambda = 0.3$) and a low learning rate ($\eta = 10^{-3}$) at a base batch size $B = 4096$. The AdamW optimizer is used, and so for scaling to other batch sizes $\hat{B} = \kappa B$ we use the Adam Scaling Rule (Definition C.3)[15] We use a pre-bn linear probe as discussed in Appendix H.1. Finally, the performance of BYOL can be further improved by employing multicrop (Caron et al., 2020) by $\approx$ +2% in absolute test top-1 performance on ImageNet1k compared to without multicrop, however, as this is not our focus, we omit this from the presented recipe.

**Additional background** Achieving large scale SSL training with ViTs to large scale SSL training has been a long standing goal in the community. MoCo-v3 (Chen et al., 2021) enables the use of ViTs with contrastive learning, but achieves this through modifications of the ViT training procedures, including gradient freezing on the image patching layer, and re-introducing Batch Normalization to post-attention MLP layers. Despite these modifications, MoCo-v3 was only trained up to a batch size of 6144, where model performance begins to suffer (Chen et al., 2021). In Figure 6 we demonstrate that combining dynamic batch scaling (Appendix C.4) with the EMA Scaling Rule

---

[15]We note that Adam (Kingma & Ba, 2015) and AdamW (Loshchilov & Hutter, 2019) are equivalent in the limit of zero weight decay, and that the Adam Scaling Rule (Definition C.3) was derived with zero weight decay (Malladi et al., 2022).

Table 10: BYOL ViT-B/16 hyperparameters.

|  | BYOL ViT-B/16 |
|---|---|
| ImageNet1k Linear Probe Test Top-1 | 74.47% (Figure 19) |
| Weight initialization | `trunc_normal(.02)` |
| Backbone normalization | LayerNorm |
| Head normalization | BatchNorm |
| Synchronized BatchNorm over replicas | No |
| Learning rate schedule | Single Cycle Cosine |
| Learning rate warmup (epochs) | 40 |
| Learning rate minimum value | $1 \times 10^{-6}$ |
| Training duration (epochs) | 480 |
| Optimizer | AdamW |
| Optimizer scaling rule | Adam |
| Base $(\beta_1, \beta_2)$ | (0.9, 0.95) |
| Base learning rate | $1 \times 10^{-3}$ |
| Base batch size | 4096 |
| Base teacher momentum | 0.99 |
| Weight decay | 0.3 |
| Weight decay scaling rule | None |
| Weight decay skip bias | Yes |
| Numerical precision | `bf16` |
| Augmentation stack | BYOL (Grill et al., 2020) |
| Stochastic depth | 0.1 |

(Definition 1.2) enables BYOL to be trained using ViTs to batch sizes of 24,576 without any drop in performance compared to the reference batch size of 4096. We emphasize that the piecewise transitions in the schedules are important for preserving training dynamics.

## H.4 The role of Batch Normalization and Layer Normalization in BYOL with ViTs

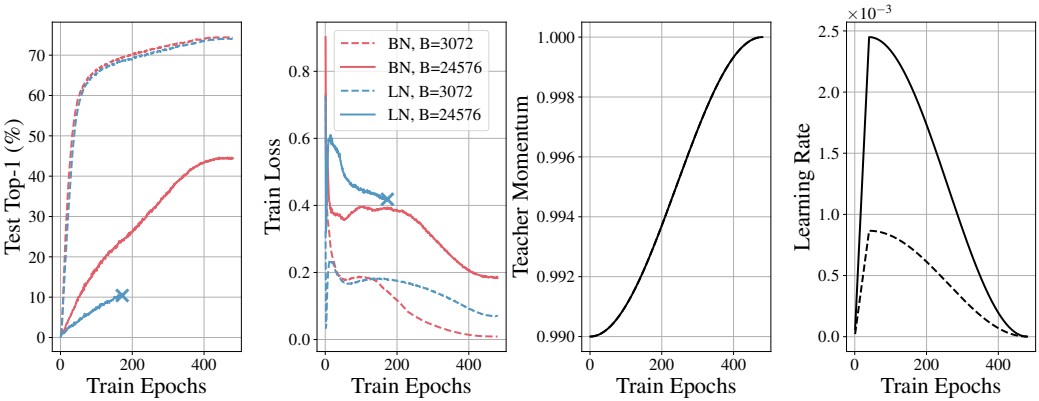

Figure 19: *BYOL ViT-B/16 on ImageNet1k for different scalings $\kappa$*. We present runs comparing LayerNorm (blue) to BatchNorm (red) in the projection and prediction heads of BYOL ViT models for batch size 3072 (dashed) and 24,576 (solid) *without the EMA Scaling Rule*. $\kappa = 1$ corresponds to $B = 4096$. In all scenarios the transformer backbone *only* uses LayerNorm. We truncate the training of the large batch size LayerNorm variant to preserve compute (indicated by ×).

Here we compare the roles of Batch Normalization (BatchNorm, Ioffe & Szegedy (2015)) and Layer Normalization (LayerNorm, Ba et al. (2016)) in the projection and prediction heads of BYOL (Grill et al., 2020) using ViTs.

It has been observed that BatchNorm plays a critical role in BYOL predictor and projector dynamics (Fetterman & Albrecht, 2020), and using either LayerNorm or *no normalization* significantly decreases model performance. Subsequently, it was demonstrated (Richemond et al., 2020) that competitive BYOL performance could be achieved through a combination of Group Normaliza-

tion (GroupNorm, Wu & He (2018)) and Weight Standardization (Qiao et al., 2019). Additionally, Richemond et al. (2020) showed that if BatchNorm is used in the backbone, one can use LayerNorm or *no normalization* in the predictor and projector without any performance drop.

In this work, we we show it is possible to train BYOL ViT using *only LayerNorm* across the backbone, projector and predictor (see Figure 19), decoupling BYOL's reliance on batch statistics, a desirable trait for a representation learning algorithm (Brock et al., 2021). At batch size 3072, using LayerNorm in the predictor and projector achieves competitive performance (74.10%), performing slightly worse than using BatchNorm (74.47%). At the larger batch size of 24,576, runs perform significantly worse as the EMA Scaling Rule was not applied.

## H.5  Longer training duration with incremental Progressive Scaling

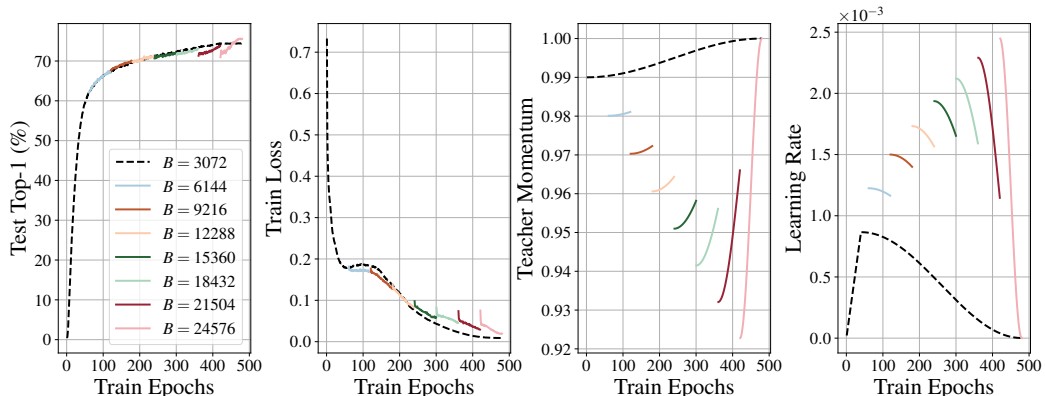

Figure 20: *BYOL ViT-B/16 on ImageNet1k* for different scalings $\kappa$. The baseline model ($\kappa = 0.75$, black dashed) uses batch size 3072 and teacher momentum $\rho_B = 0.99$. We increment the batch size by 3072 every 60 epochs to a final batch size of 24,576 using Progressive Scaling (Definition 3.2).

Here we use the same base hyperparameters as Table 10, except that we train for 480 instead of 300 epochs. To mitigate the student impulse phenomena discussed in Section 3.4, in Figure 20 we investigate increasing the batch size every 60 epochs using Progressive Scaling (Definition 3.2). We observe that this more gradual procedure enables closer tracking of the baseline train loss trajectory. Additionally, this procedure results in a scaled linear probe performance that outperforms the baseline (75.64% compared to the baseline performance of 74.47%). The same procedure can be applied to the LayerNorm variant discussed in Appendix H.4, which produces a similar result (75.09% compared to the baseline performance of 74.10%).

## H.6  Building intuition around Progressive Scaling and momentum sensitivity

Our final BYOL ViT results are to help build intuition around Progressive Scaling (Definition 3.2), as well as when the EMA Scaling Rule is most important. In Figure 21 we explore transitioning from the baseline batch size 4096 model to batch size 24,576 in a *single transition* after 60 epochs. After this transition, we continue training for 240 epochs for a range of momenta: $\rho \in \{0.8, 0.9, 0.95, 0.97, 0.9867, 0.994, 0.999\}$ *without* the EMA Scaling Rule.

We observe that after the transition, any $0.9 \leq \rho \leq 0.994$ produces a linear probe performance that matches or outperforms the baseline at the end of training. This indicates that after the initial training period, BYOL becomes less sensitive to the choice of teacher momentum. Note that without the initial 60 epochs of training with batch size 4096, *all models*, including those employing the EMA Scaling Rule diverge (see $B = 24,576$ in Figure 6).

We present an illustration for why this might happen in Figure 22. First, we see that using the EMA Scaling Rule *always* keeps the model within the acceptable momentum region. We also wee that *not* using the EMA Scaling Rule can keep the model within the acceptable momentum region for a range of batch sizes, depending on how large wide in momenta the acceptable region is at the base batch size. Finally, we see that the momentum value matters much more at low values of momenta

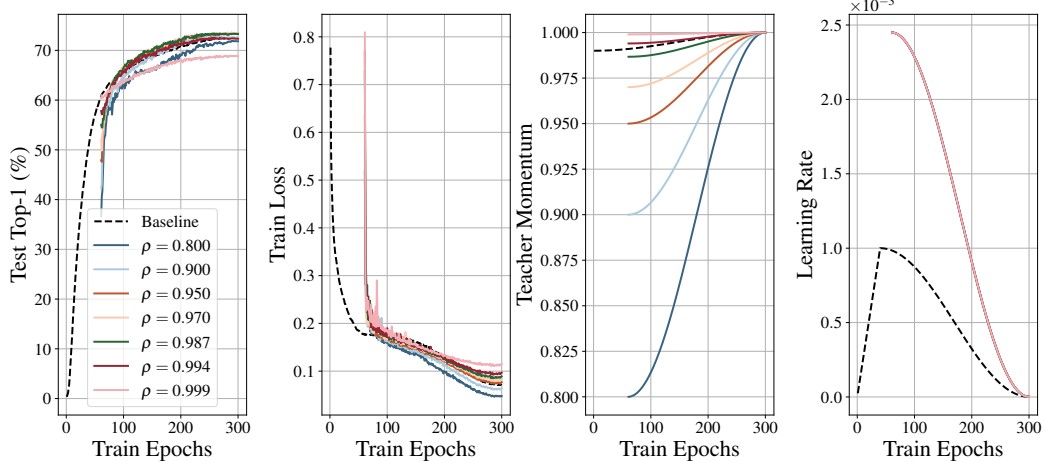

Figure 21: *BYOL ViT-B/16 on ImageNet1k for different momenta $\rho$*. The baseline model ($\rho = 0.99$, black dashed) uses batch size 4096. At the 60th epoch we apply Progressive Scaling (Definition 3.2) and transition to batch size 24576. We train for a further 240 epochs without EMA scaling for a range of momenta: $\rho \in \{0.9, 0.95, 0.97, 0.9867, 0.994\}$.

(the acceptable momentum region shrinks), whereas at large momenta, this region of acceptability widens.

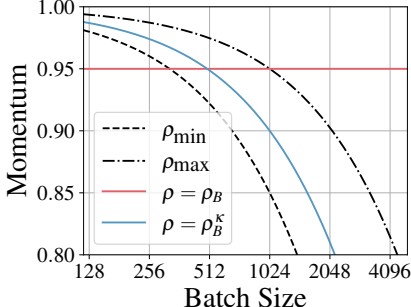

Figure 22: *A hypothetical scenario where there is an upper and lower limit for momenta qualitatively leading to the same result.* We assume at base batch size $B = 1024$ there is an upper ($\rho_{\max}$, black dashdot) and lower ($\rho_{\min}$, black dashed) limit for *valid momenta*. We show what happens if we start with $\rho_B = 0.95$ at a batch size of 4096, and scale with ($\rho = \rho_B^\kappa$, blue) and without ($\rho = \rho_B$, red) the EMA Scaling Rule.

## H.7 Compute usage for ViT BYOL investigation

We now summarize the compute usage for the BYOL ViT experiments. First we detail cost of reproducing Figure 6 in Table 11, Figure 19 in Table 12, and Figure 20 in Table 13.

Table 11: Compute usage for baseline ViT BYOL investigation in Figure 6. Values *include* node allocation times (typically a small % of corresponding total runtime), giving a practical estimate of reproduction cost. All experiments conducted are using 80Gb A100s, and experiments indicated by (†) have a faster interconnect.

| Batch Size | GPUs | Time (h) | Compute/Run (GPUh) | Runs | Compute (GPUh) |
|---|---|---|---|---|---|
| 4,096 | 32 | 16.6 | 531.4 | 2 | 1,062.7 |
| 8,192 | 48 | 14.1 | 678.0 | 2 | 1,356.1 |
| 16,384 | 96 | 8.3 | 800.3 | 2 | 1,600.6 |
| 24,576 | 128 | 6.6 | 850.1 | 4 | 3,400.4 |
| 32,768† | 176 | 4.1 | 721.7 | 2 | 1,443.4 |
| **Total** | | | | | **8,863.1** |

Table 12: Compute usage for ViT BYOL investigation into BatchNorm and LayerNorm variants in Figure 19. Values *include* node allocation times (typically a small % of corresponding total runtime), giving a practical estimate of reproduction cost. All experiments conducted are using 80Gb A100s, and were run for 480 epochs, except those indicated by (∗) were truncated early (see Figure 19 for more details).

| Batch Size | Normalization | GPUs | Time (h) | Compute (GPUh) |
|---|---|---|---|---|
| 3,072 | BatchNorm | 16 | 47.9 | 766.0 |
| 3,072 | LayerNorm | 16 | 48.0 | 768.7 |
| 24,576 | BatchNorm | 128 | 14.8 | 1900.4 |
| 24,576∗ | LayerNorm | 128 | 3.5 | 451.1 |
| **Total** | | | | **3,886.2** |

Table 13: Compute usage for ViT BYOL investigation into incremental scaling in Figure 20. Values *include* node allocation times (typically a small % of corresponding total runtime), giving a practical estimate of re-production cost. All experiments conducted are using 80Gb A100s for 60 epochs. Stage 0 corresponding to the baseline in Figure 20 is the run detailed in the first row of Table 12, using a batch size of 3,072, Batch Normalization, and 16 GPUs. Computing only the first 60 epochs of stage 0 corresponds to approximately 127.7 GPUh, which would bring the total cost of Figure 20 to 1,432.9 GPUh.

| Stage | Batch Size | GPUs | Time (h) | Compute (GPUh) |
|---|---|---|---|---|
| 1 | 6,144 | 32 | 3.5 | 113.0 |
| 2 | 9,216 | 48 | 3.1 | 149.8 |
| 3 | 12,288 | 64 | 2.8 | 176.0 |
| 4 | 15,360 | 80 | 2.3 | 186.5 |
| 5 | 18,432 | 96 | 2.1 | 202.9 |
| 6 | 21,504 | 112 | 2.1 | 235.8 |
| 7 | 24,576 | 128 | 1.9 | 241.3 |
| **Total** | | | | **1,305.2** |

Next, the cost of a single momentum ablation presented in Figure 21 is 240 epochs at batch size 24,576, which is $\approx 240/480 \times 1900.4$ GPUh $= 950.2$ GPUh, giving a total cost over seven runs of **6651.4 GPUh**.

Finally, providing a full view of the investigations carried out for the ViT BYOL is given in Table 14.

Table 14: Total compute usage for ViT BYOL investigations.

| | Compute (GPUh) |
|---|---|
| Baselines (Figure 6 and Table 11) | 8,863.1 |
| BatchNorm and LayerNorm (Figure 19 and Table 12) | 3,886.2 |
| Incremental scaling (Figure 20 and Table 13) | 1,305.2 |
| Momentum ablations (Figure 21) | 6,651.4 |
| All other compute, e.g. code development, runs with errors, and debugging | 84,984.1 |
| **Total** | **105,690.0** |

## H.8  ResNet-18 hyperparameter sensitivity analysis

To demonstrate that the EMA Scaling Rule works for a broad range of optimization hyperparameters (i.e. *beyond* those presented in Figure 5 and Section 3.4), we provide a sensitivity analysis for base teacher momentum $\rho_B$ and base learning rate $\eta_B$ in the challenging setting of BYOL.

**Base teacher momentum**  In Figure 23 we show the effect of changing the base teacher momentum $\rho_B$, defined at batch size 1024. The EMA Scaling Rule is robust to modifications of momentum down to $\rho_B \approx 0.946$ in this particular setting. Below $\rho_B \approx 0.946$, matching is poor, although the smallest momentum in this setting corresponds to $0.841^4 \approx 0.5$, which is a particularly small teacher momentum, and is unlike to provide utility over the using the target model (see Appendix E.2).

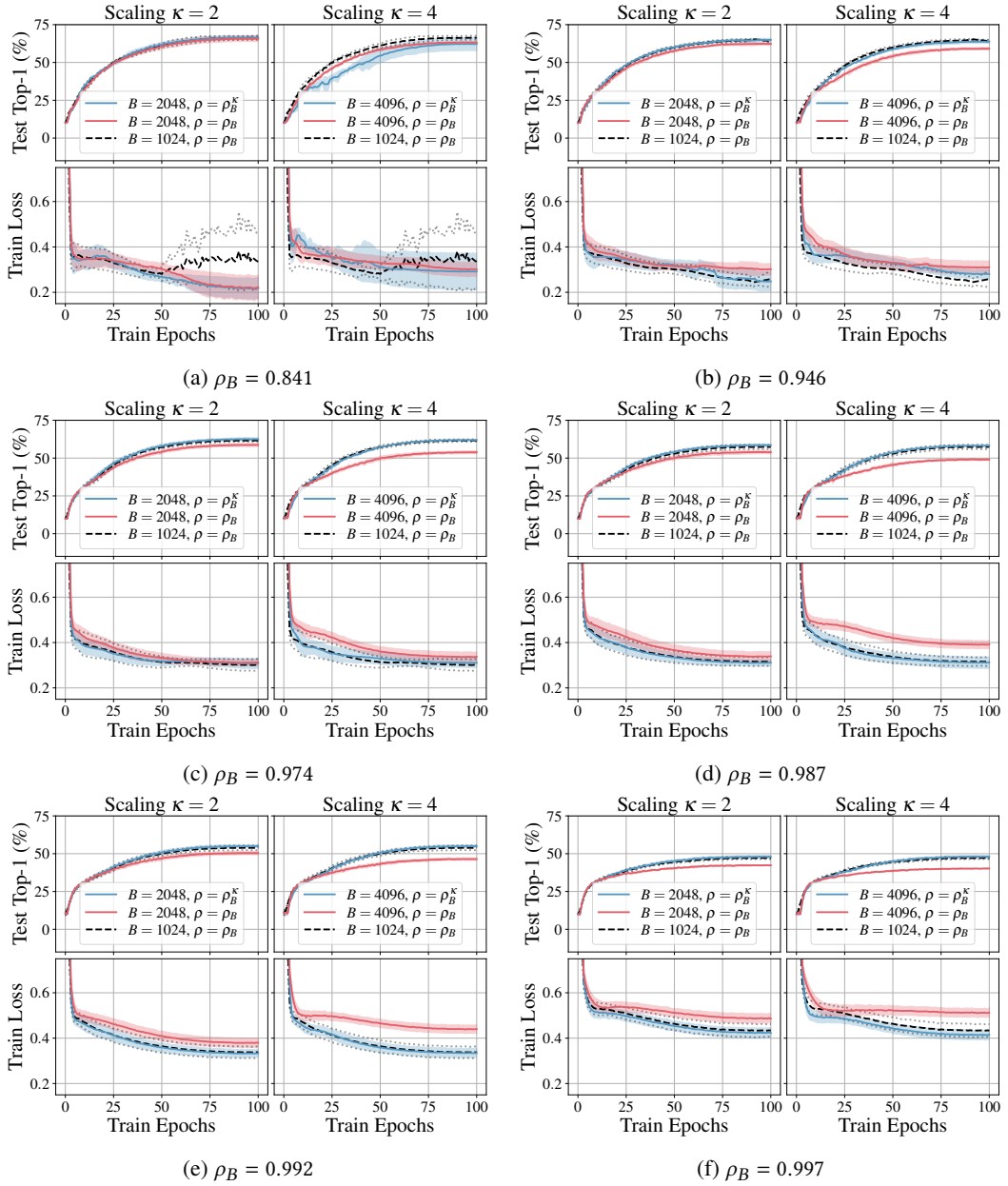

Figure 23: *ResNet-18 BYOL on CIFAR10 teacher momentum sensitivity* ($\eta_B = 0.08$) for scalings $\kappa \in \{2, 4\}$ and base teacher momenta $\rho_B \in \{0.841, 0.946, 0.974, 0.987, 0.992, 0.997\}$ defined at $\kappa = 1$. The baseline ($\kappa = 1$, black dashed) uses batch size 1024, and is scaled from batch size 2048 (left) to 4096 (right) with (blue, $\rho = \rho_B^\kappa$) and without (red, $\rho = \rho_B$) the EMA Scaling Rule. Bands indicate mean and standard deviation across three runs.

**Base learning rate** In Figure 24 we show the effect of changing the base learning rate $\eta_B$, defined at batch size 1024. The EMA Scaling Rule is robust over a wide range of learning rates. At the largest learning rate $\eta_B = 0.5$ matching starts to become poor at scaling $\kappa = 4$.

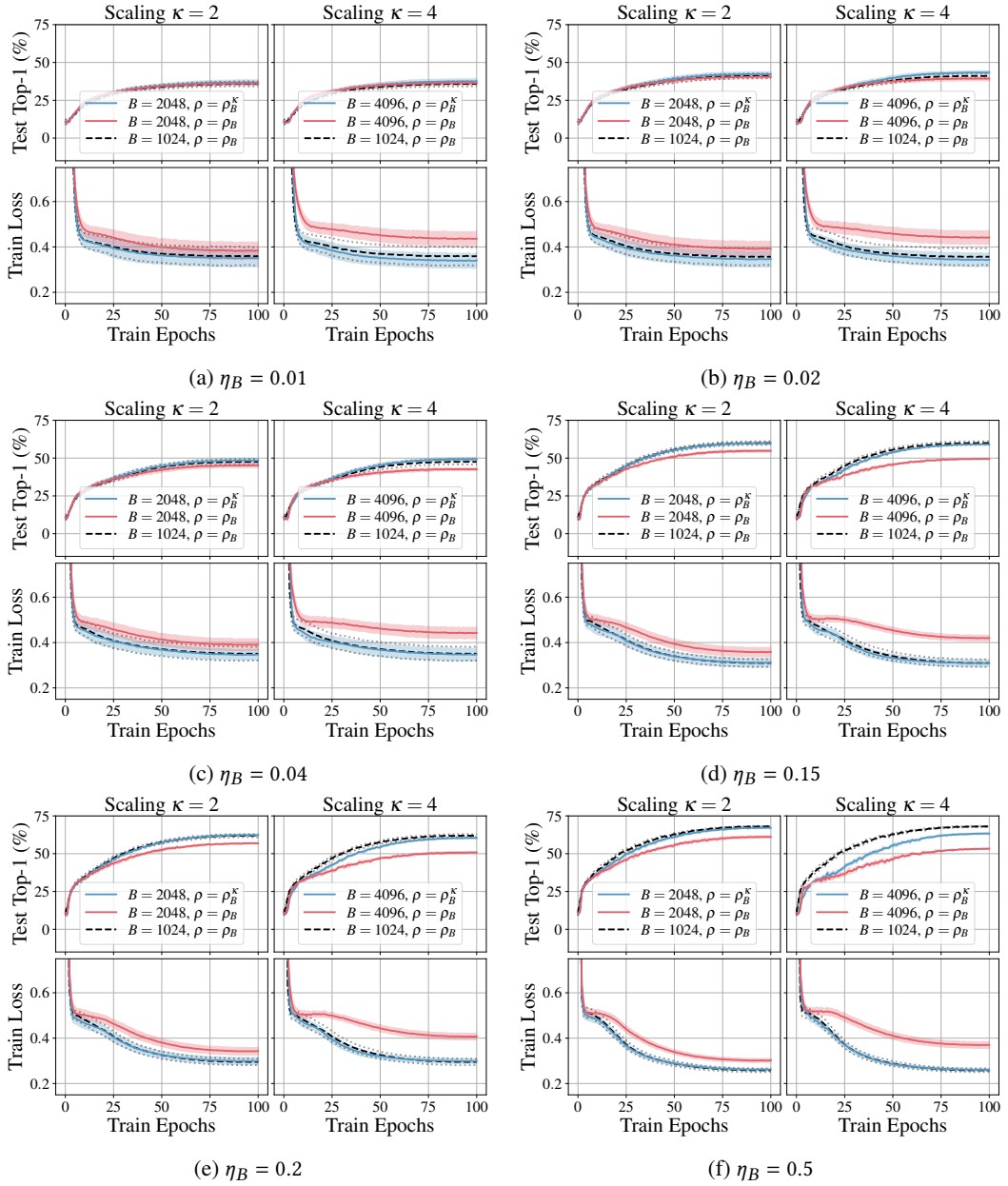

Figure 24: *ResNet-18 BYOL on CIFAR10 learning rate sensitivity* ($\rho_B = 0.992$) for scalings $\kappa \in \{2, 4\}$ and base learning rates $\eta_B \in \{0.01, 0.02, 0.04, 0.15, 0.20, 0.50\}$ defined at $\kappa = 1$. The baseline ($\kappa = 1$, black dashed) uses batch size 1024, and is scaled from batch size 2048 (left) to 4096 (right) with (blue, $\rho = \rho_B^\kappa$) and without (red, $\rho = \rho_B$) the EMA Scaling Rule. Bands indicate mean and standard deviation across three runs.

## H.9 ResNet-18 additional scaling analysis

To demonstrate that the EMA Scaling Rule works for a broad range of scalings $\kappa$ (i.e. *beyond* those presented in Figure 5 and Section 3.4), we investigate scaling down to $\kappa = 1/8$ in Figure 25. We see that the EMA Scaling Rule works well down to the small batch size of 128, although matching is not perfect. We suspect this is due to the presence of Batch Normalization layers in the ResNet-18 architecture, which underperform at small batch sizes (Ioffe & Szegedy, 2015). The synthetic analysis of Section 3.1 instead demonstrated the EMA Scaling Rule holding for scalings spanning factors of $\kappa$ that differ by 1024, with scaling error insensitive to the value of $\kappa$ for sufficiently low $\kappa$ (see Figure 1b).

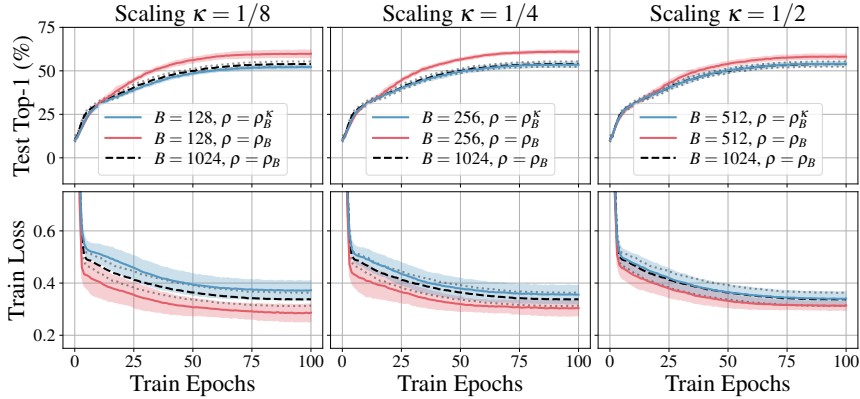

Figure 25: *ResNet-18 BYOL on CIFAR10 lower scaling analysis* ($\eta_B = 0.08$, $\rho_B = 0.992$) for scalings $\kappa \in \{1/8, 1/4, 1/2\}$. The baseline ($\kappa = 1$, black dashed) uses batch size 1024, and is scaled from batch size 128 (left) to 512 (right) with (blue, $\rho = \rho_B^\kappa$) and without (red, $\rho = \rho_B$) the EMA Scaling Rule. Bands indicate mean and standard deviation across three runs.

## H.10  Scaling a ResNet-50 BYOL using LARS and Progressive Scaling

Here we investigate whether Progressive Scaling and the EMA Scaling Rule can be used in practice where there is no known optimizer SDE approximation. We use the default 300 epoch configuration for BYOL (Grill et al., 2020) in Figure 26. We see that although trajectories during training do not match, we are able to match or surpass the linear probe performance of the BYOL baseline at the larger batch size if 32,768. *This indicates that the contributions of our work have practical utility beyond the theoretical constraints.*

The compute usage for the BYOL ResNet using LARS is detailed in Table 15.

Table 15: Compute usage for ResNet 50 LARS investigation in Figure 26. Values *include* node allocation times (typically a small % of corresponding total runtime), giving a practical estimate of reproduction cost. All experiments conducted are using 80Gb A100s.

| Batch Size | GPUs | Time (h) | Compute (GPUh) |
|---|---|---|---|
| $4,096 \rightarrow 32,768$ (120 Epochs) | 128 | 14.1 | 1809.8 |
| $4,096 \rightarrow 32,768$ (60 Epochs) | 128 | 12.9 | 1655.9 |
| $4,096$ | 16 | 32.8 | 524.9 |
| All other compute, e.g. code development, runs with errors, and debugging | | | 17,654.6 |
| **Total** | | | **21645.2** |

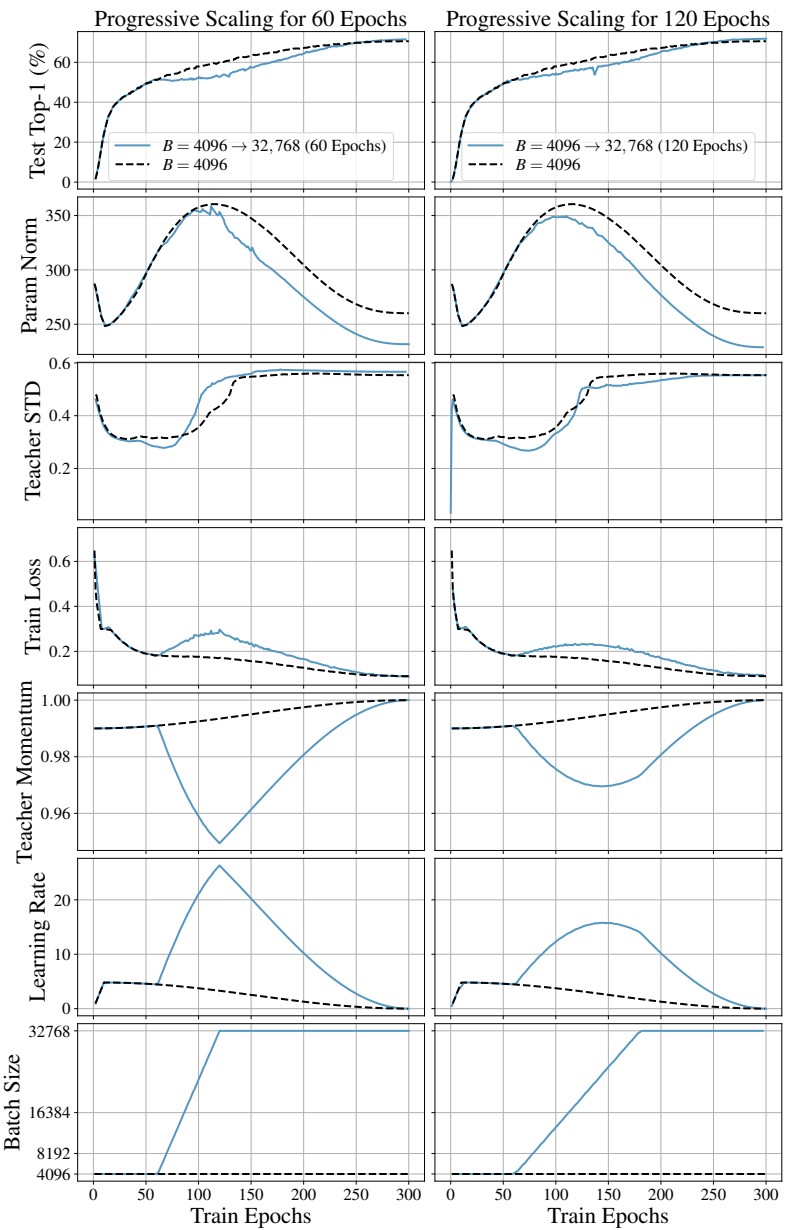

Figure 26: *ResNet50 BYOL on ImageNet1k using LARS* for different configurations of progressive scaling. The baseline (black dashed) uses batch size 4096 and momentum $\rho_B = 0.99$. We consider progressive scaling (blue) smoothly from epoch 60 for 60 epochs (left) and 120 epochs (right) up until batch size 32,768, scaling the learning rate linearly, and applying the EMA Scaling Rule.

## H.11 Preventing collapse phenomena in DINO at scale

Until now, our representatives SSL method has been BYOL for reasons discussed in Section 3.4. Here, we will turn our attention to DIstillation with NO labels (DINO) (Caron et al., 2021), which has the update rule presented in Definition H.4.

**Definition H.4** (DINO Update). *DINO learns unsupervised features by matching predictions over emergent pseudo-labels of a student backbone and head $f(\,\cdot\,;\theta)$ to those of an EMA teacher $f(\,\cdot\,;\zeta)$ through a cross-entropy guided distillation procedure. DINO has a additional centering procedure, which is a form of batch normalization with momentum $\rho_c = 0.9$ which we do not scale using the*

Table 16: DINO ViT-B/16 Training hyperparameters.

|  | DINO ViT-B/16 |
|---|---|
| CIFAR10 Linear Probe Top-1 ($\rho_B = 0.996$) | 85.38% |
| CIFAR10 Linear Probe Top-1 ($\rho_B = 0.992$) | 86.96% |
| Weight initialization | `trunc_normal(.02)` |
| Normalization | Layer Norm |
| Learning rate schedule | Single Cycle Cosine |
| Learning rate warmup (epochs) | 50 |
| Learning rate minimum value | $1 \times 10^{-6}$ |
| Training duration (epochs) | 280 |
| Optimizer | AdamW |
| Optimizer scaling rule | Adam |
| Base ($\beta_1, \beta_2$) | (0.9, 0.95) |
| Base learning rate | $3 \times 10^{-4}$ |
| Base batch size ($B$) | 1024 |
| Base teacher momentum ($\rho_B$) | 0.992 or 0.996 |
| Base weight decay | 0.04 |
| Weight decay scaling rule | Linear |
| Weight decay skip bias | Yes |
| Center Momentum | 0.9 |
| Center Momentum Scaling Rule | None |
| Precision | `bf16` |
| Augmentation stack | `DINO multi-crop` (Caron et al., 2020) |

*EMA Scaling Rule. The update for the parameters of DINO is*

$$\theta_{t+1} = \theta_t - \eta \times \frac{1}{B} \sum_{x \in \mathbb{B}} \nabla_\theta \mathcal{L}(x; \theta_t, \zeta_t, \mathbf{c}_t) \tag{78}$$

$$\zeta_{t+1} = \rho \, \zeta_t + (1 - \rho) \, \theta_{t+1} \tag{79}$$

$$\mathbf{c}_{t+1} = \rho_c \, \mathbf{c}_t + (1 - \rho_c) \, \mathbb{E}_{x'} \zeta(x') \tag{80}$$

$$\text{with} \quad \mathcal{L}(x; \theta_t, \zeta_t, \mathbf{c}_t) = H\big(f(x_1, \theta_t), f(x_2, \zeta_t) - \mathbf{c}_t\big) + (x_1 \leftrightarrow x_2), \tag{81}$$

*where $H(\mathbf{a}, \mathbf{b}) \equiv -\sum_{m=1}^{M} p_m(\mathbf{a}) \log p_m(\mathbf{b})$ is the cross-entropy between categorical distributions over $M$ (emergent pseudo-)classes given logits $\mathbf{a}, \mathbf{b} \in \mathbb{R}^M$, $x_1$ and $x_2$ are two views of a single variate $x$, often produced by augmentations, and $x_1 \leftrightarrow x_2$ denotes symmetrization over $x_1$ and $x_2$.*

In practice, DINO employs multi-crop (Caron et al., 2021). We omit this detail for clarity of presentation, although we *do* use multi-crop in the experiments that follow.

Our interest DINO is due to the difficulty in its optimization[16], and in particular, preventing collapse phenomena in DINO at batch sizes above 1024, which is an open research problem. In this section, we will show that a combination of the EMA Scaling Rule (Definition 1.2) and Progressive Scaling (Definition 3.2) enable training of DINO beyond batch size 1024 without sacrificing performance.

**Hyperparameters** Base hyperparameters are presented in Table 16.

**Results** In Figures 27 and 28 we show the results obtained training DINO on CIFAR-10 with $\rho_B = 0.996$ and $\rho_B = 0.992$ respectively at the reference batch size of 1024. We employ smooth Progressive Scaling (Definition 3.2) between epochs 120 and 180.

At batch size 2048, the training loss matches the reference *only* when the EMA Scaling Rule is applied, whereas the run *without* the scaling rule diverges from the reference. The impact of this divergence is emphasized as we consider the larger batch size of 4096. Here, there is also a gap *with* the EMA Scaling Rule, however is approximately three times smaller than the gap *without* the EMA Scaling Rule.

---

[16]For an example, see https://github.com/facebookresearch/dino/issues/43#issuecomment-881453515.

Additionally, we observe that using $\rho_B = 0.992$ yields higher Top-1 accuracy over $\rho_B = 0.996$, and in our experiments, using the EMA Scaling Rule *always* performs better in terms of linear probe performance than not using the scaling rule.

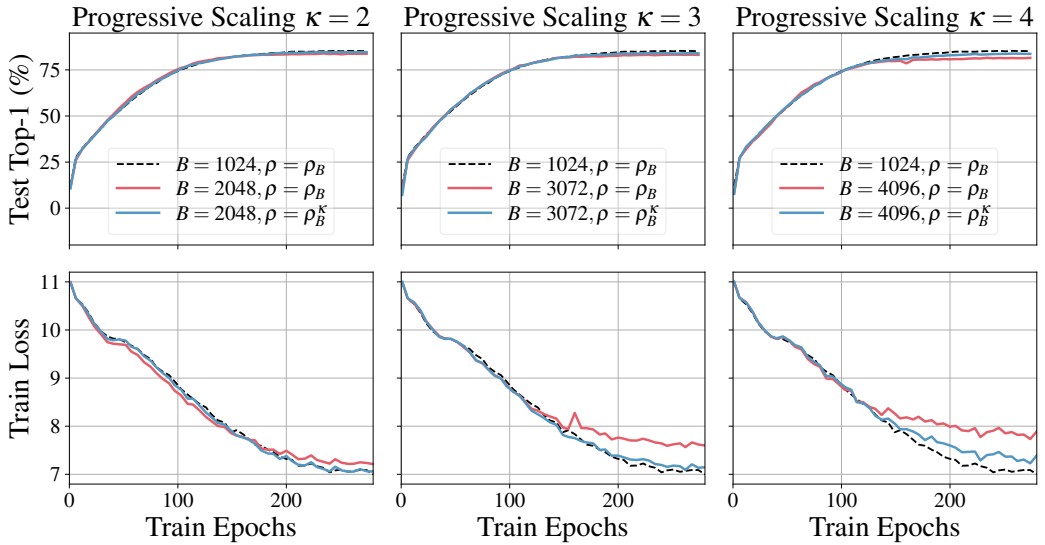

Figure 27: *DINO ViT-B/16 on CIFAR-10* for different scalings $\kappa$ and base teacher momentum $\rho_B = 0.996$. The baseline model ($\kappa = 1$, black dashed) uses batch size 1024 and center momentum $\rho_c = 0.9$, and is scaled up from batch size 2048 (left) to 4096 (right) with (blue, $\rho = \rho_B^\kappa$) and without (red, $\rho = \rho_B$) the EMA Scaling Rule. Between epochs 100 and 180 we scale the batch size using progressive scaling (Definition 3.2).

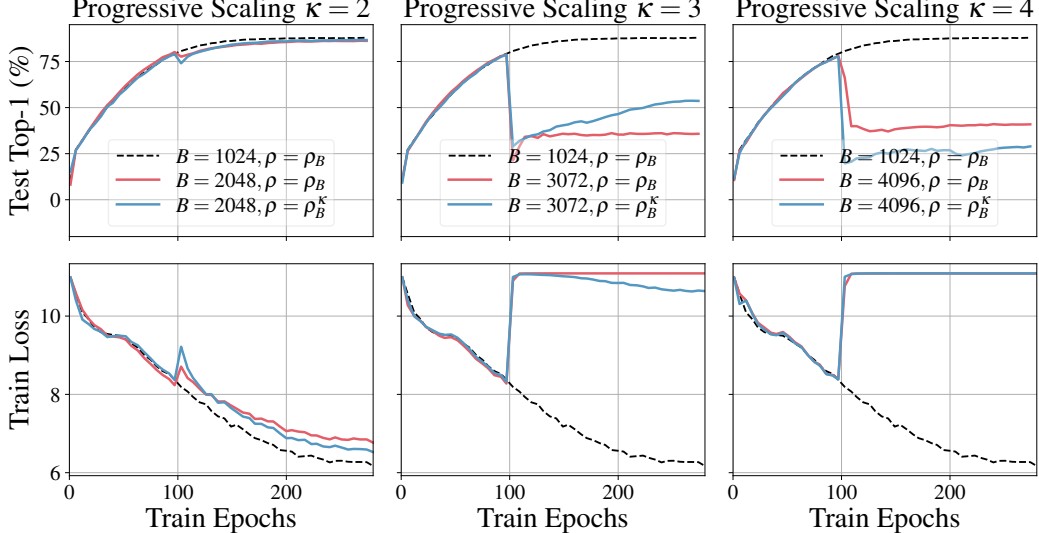

Figure 28: *DINO ViT-B/16 on CIFAR-10* for different scalings $\kappa$ and base teacher momentum $\rho_B = 0.992$. The baseline model ($\kappa = 1$, black dashed) uses batch size 1024 and center momentum $\rho_c = 0.9$, and is scaled up from batch size 2048 (left) to 4096 (right) with (blue, $\rho = \rho_B^\kappa$) and without (red, $\rho = \rho_B$) the EMA Scaling Rule. Between epochs 100 and 180 we scale the batch size using progressive scaling (Definition 3.2).

In Figure 29 we show how the hyperparameters $\rho$, $B$ and learning rate change with the progressive scaling in Definition 3.2.

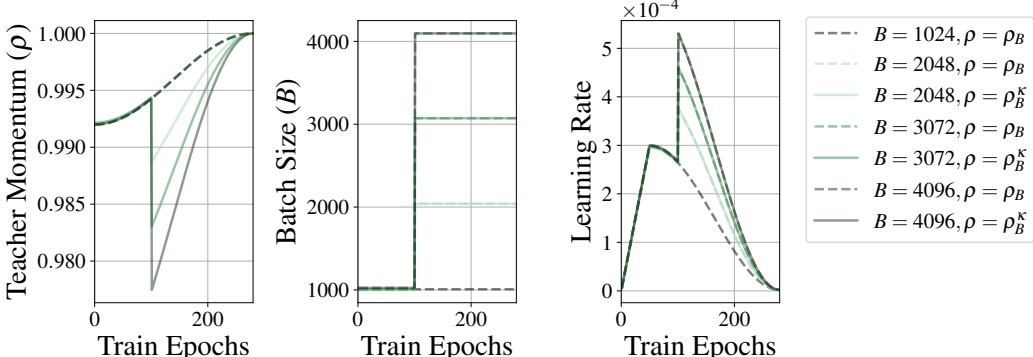

Figure 29: *DINO ViT-B/16 on CIFAR-10* for different scalings $\kappa$ and base teacher momentum $\rho_B = 0.992$. We show how the hyperparameters $\rho$, $B$ and learning rate change with the Progressive Scaling in Definition 3.2. These hyperparameters correspond to the training runs in Figure 28. Those for Figure 27 are identical, with the exception of $\rho$ that starts at 0.996 instead of 0.992.

We also attempted to use a sharp batch size transition (Figures 30 and 31), which leads to the collapse pheonomena observed in prior work. This collapse happens with and without the EMA Scaling Rule. We suspect this is due to dynamics specific to DINO's early phase that are even more challenging to replicate under discretization than those of BYOL.

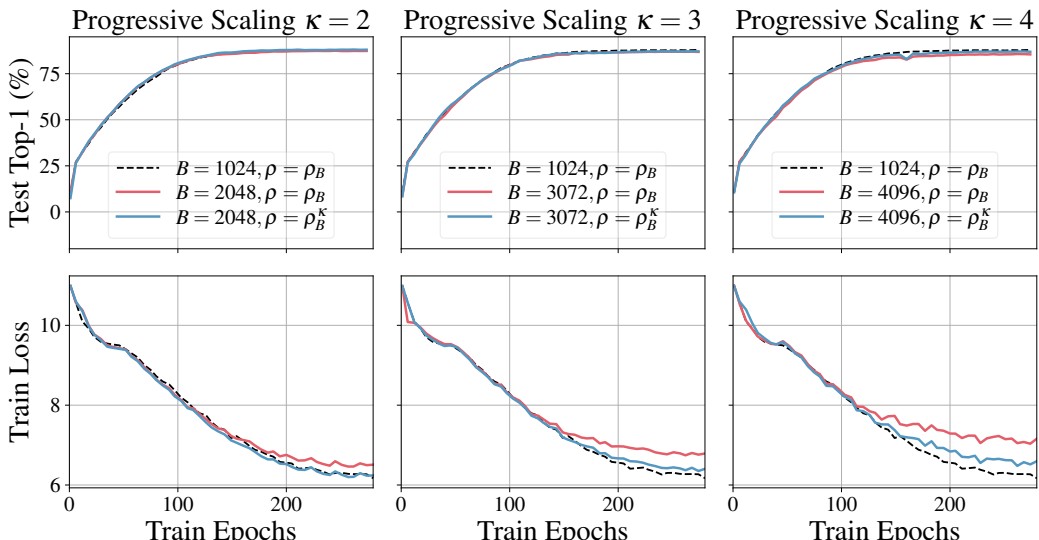

Figure 30: *DINO ViT-B/16 on CIFAR-10* for different scalings $\kappa$ and base teacher momentum $\rho_B = 0.992$. The baseline model ($\kappa = 1$, black dashed) uses batch size 1024 and center momentum $\rho_c = 0.9$, and is scaled up from batch size 2048 (left) to 4096 (right) with (blue, $\rho = \rho_B^\kappa$) and without (red, $\rho = \rho_B$) the EMA Scaling Rule. Progressive Scaling is employed with a sharp transition at epoch 100, leading to a collapse phenomenon.

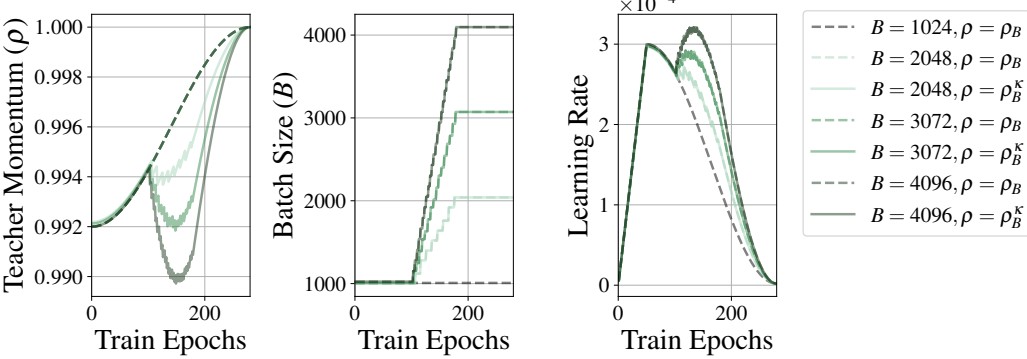

Figure 31: *DINO ViT-B/16 on CIFAR-10* with $\rho_B = 0.992$ and a sharp transition in batch size at epoch 100. We show how the hyperparameters $\rho$, $B$ and learning rate change with sudden scaling. These hyperparameters correspond to the training runs in Figure 30.

**Compute**  The compute usage for the DINO investigations is detailed in Table 17.

Table 17: Compute usage for DINO investigations. Values *include* node allocation times (typically a small % of corresponding total runtime), giving a practical estimate of reproduction cost. All experiments conducted are using 80Gb A100s.

| Batch Size | GPUs | Time (h) | Compute/Run (GPUh) | Runs | Compute (GPUh) |
|---|---|---|---|---|---|
| $1,024$ | 24 | 6.8 | 163.5 | 2 | 327.0 |
| $1,024 \rightarrow 2,048$ | 40 | 4.6 | 182.4 | 1 | 182.4 |
| $1,024 \rightarrow 3,072$ | 48 | 4.0 | 189.9 | 1 | 189.9 |
| $1,024 \rightarrow 4,096$ | 64 | 3.3 | 212.3 | 1 | 212.3 |
| $1,024 \rightarrow 2,048$ (100 Epochs) | 40 | 4.8 | 190.6 | 4 | 762.3 |
| $1,024 \rightarrow 3,072$ (100 Epochs) | 48 | 4.0 | 192.5 | 4 | 769.9 |
| $1,024 \rightarrow 4,096$ (100 Epochs) | 64 | 3.6 | 232.1 | 4 | 928.2 |
| All other compute, e.g. code development, runs with errors, and debugging | | | | | 38239.2 |
| **Total** | | | | | **41,611.1** |

Our results in this section show it is possible to scale DINO to large batch sizes *without* sacrificing performance by using *both* the EMA Scaling Rule and Progressive Scaling, providing the batch size schedule of Progressive Scaling is not sudden.

# I  Additional details on numerical stability

A general analysis of overflow and underflow of the EMA Update (Definition 1.1) or EMA Scaling Rule (Definition 1.2) for different momenta $\rho$, particularly for IEE-754 floating point values, is beyond the scope of this work due to non-linearity from mechanisms like gradual underflow (IEEE, 2019).

In our setting, do not suffer from practical overflow or underflow issues through exponentiation when applying the EMA Scaling Rule, as FP32 precision allows a maximum $\rho = 1 - \varepsilon$, or minimum $\rho = \varepsilon$ with $\varepsilon \approx 1.2 \times 10^{-7}$. Take self-supervised image representation learning (Section 3.4) as a baseline, with $\kappa = 1$ corresponding to batch size $B = 4096$ with momentum $\rho_B = 0.996$. The maximum value of $\rho$ corresponds to scaling $\kappa = \log(\rho_B)/\log(\varepsilon) \approx 1/(32K)$, give a batch size less than one, while the minimum value of $\rho$ corresponds to scaling $\kappa = \log(\rho_B)/\log(1-\varepsilon) \approx 4K$, giving a batch size $B \approx 8M$ which is beyond current hardware feasibility, and beyond the breakdown of known optimizer scaling rules (Li et al., 2021).

To examine how momentum may induce numerical errors in practice during training, we train a linear regression model with a Polyak-Ruppert average Definition 3.1, and and track the difference between FP32 model weights and weights in i) BF16; ii) FP16; and iii) a second FP32 run, which act as a proxy for overflow and underflow. In Figure 32 we plot these differences using

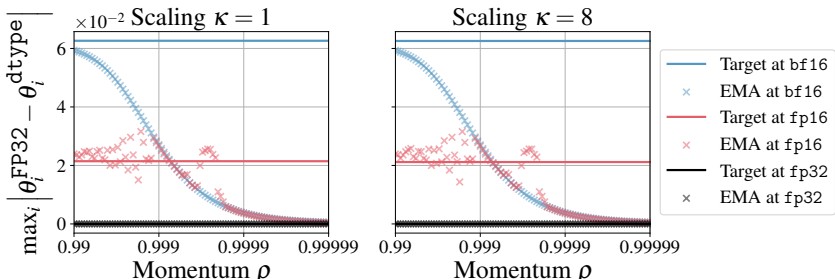

Figure 32: Numerical precision of target and EMA networks compared to an FP32 reference on a regression task for a range of momenta.

the maximum absolute difference between model parameters, where the maximum is taken over individual weights

$$\text{MaxAbsDiff}(\texttt{dtype}) = \max_{i=1}^{P} \left| \theta_i^{\texttt{FP32}} - \theta_i^{\texttt{dtype}} \right|, \tag{82}$$

where $P$ is the number of parameters in the model. We observe that when model weights and EMA weights are FP16 (never done in practice), an increasing variance happens for FP16 as the value of the momentum $\rho$ approaches 0.99999, whereas BF16 and FP32 are stable. We stress that all experiments presented in the paper store weights for target model *and* EMA in FP32 and use automatic-mixed precision to cast them to BF16 during training, and so do not encounter momentum-induced overflow or underflow.

## J    Contributions

All authors contributed to writing this paper, designing the experiments, discussing results at each stage of the project.

**Preliminary work**    Derivation of the EMA Scaling Rule with learning rate $\eta = 0$, initial synthetic and self-supervised ImageNet1k experiments done by Dan Busbridge.

**EMA scaling rules for constant gradients**    Original proof of Equation 4 and the form of $\delta(\eta, \rho, \kappa)$ in Equation 54 done by Eeshan Gunesh Dhekane. Final proof presented in Appendix E.1 done by Dan Busbridge, verified by Eeshan Gunesh Dhekane and Pierre Ablin.

**EMA approximation theorems with SDEs**    Proofs of validity of EMA Scaling Rule in the SDE limit presented in Section 2.2 and Appendix D done by Pierre Ablin.

**Polyak-Ruppert averaging in a simple setting**    Design of noisy parabola setting of Section 3.1 and initial experiments done by Russ Webb. Design of $\rho^*$-optimality search (Equation 10), final experiments and analysis of Section 3.1 and Appendix F.1 done by Dan Busbridge.

**Polyak-Ruppert averaging on image classification**    ResNetv2-50 reproduction (Table 4) and baseline momentum identification done by Jason Ramapuram. Final ImageNet1k experiments and analysis of Section 3.2 and Appendices F.2 and F.3 done by Dan Busbridge.

**Automatic speech recognition**    Experiments and analysis of automatic speech recognition using Polyak-Ruppert averaging (Section 3.2) and continuous pseudo-labeling (Section 3.3 and Appendix G), as well as design choice of a seed model to start pseudo-labeling (aligning quality of the seed models for different batch size settings before pseudo-labeling process) done by Tatiana Likhomanenko.

**Self-supervised image representation learning**    BYOL ResNet-18 recipe (Table 9) and experiments on CIFAR10 using SGD (Figure 5), and BYOL ResNet-50 experiments using LARS (Appendix H.10) done by Dan Busbridge. BYOL ResNet 50 baseline implementation and BYOL ViT

recipe (Table 10) done by Jason Ramapuram. BYOL ViT exploratory ablations done by Eeshan Gunesh Dhekane and Jason Ramapuram. All final BYOL ViT experiments and analysis (Figure 6 and Appendices H.4 to H.6) done by Jason Ramapuram. Baseline DINO reproduction done by Dan Busbridge. DINO experiments and analysis (Appendix H.11) done by Xavier Suau Cuadros.

**Progressive Scaling** Progressive Scaling (Definition 3.2 and Algorithm 1) is proposed by Dan Busbridge based on discussions with Xavier Suau Cuadros, Tatiana Likhomanenko, Jason Ramapuram, Russ Webb, and the authors of Malladi et al. (2022). Adaptation of progressive scaling to semi-supervised learning in automatic speech recognition (Appendix G.2) done by Tatiana Likhomanenko, and to self-supervised learning in vision done by Dan Busbridge and Jason Ramapuram for BYOL (Figures 5 and 6 and Appendices H.4, H.5 and H.10) and Xavier Suau Cuadros for DINO (Appendix H.11).

**Limiting behavior of Polyak-Ruppert averaging** Original proof of limiting behavior of Polyak-Ruppert averaging done by Eeshan Gunesh Dhekane. Final proof presented in Appendix E.2 done by Dan Busbridge, verified by Eeshan Gunesh Dhekane.

**Numerical stability analysis** Polyak-Ruppert experiment (Figure 32) using linear regression for various floating point precisions done by Jason Ramapuram.

**Implementation details** Investigations carried out in two distributed, scalable frameworks: Jax for automatic speech recognition experiments, done by Tatiana Likhomanenko; and PyTorch for all remaining investigations, done by Dan Busbridge, Xavier Suau Cuadros, Eeshan Gunesh Dhekane, Jason Ramapuram and Russ Webb. Initial implementation of progressive scaling experiments for incremental-style strategies (e.g. Appendix H.5) showing feasibility done by Jason Ramapuram, and subsequent progressive scaling implementations for smooth strategies (e.g. Appendices H.10 and H.11) done by Dan Busbridge and Xavier Suau Cuadros.

