**Compute**   *[This section has been redacted to preserve anonymity during the peer-review process. If this work is accepted, the full details compute used for these experiments, including: the experiments presented, hyperparameter optimization, and the development process, will be provided.]*

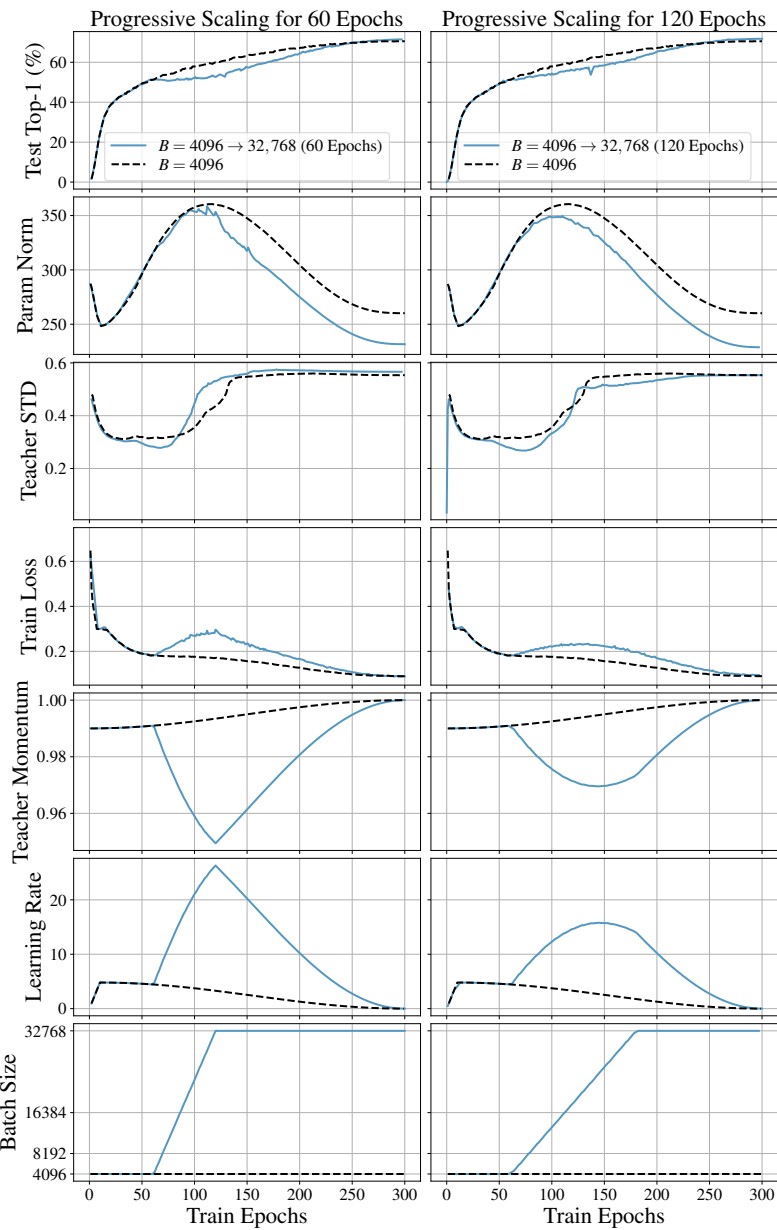

Figure 23: *ResNet50 BYOL on ImageNet1k using LARS* for different configurations of progressive scaling. The baseline (black dashed) uses batch size 4096 and momentum $\rho_B = 0.99$. We consider progressive scaling (blue) smoothly from epoch 60 for 60 epochs (left) and 120 epochs (right) up until batch size 32,768, scaling the learning rate linearly, and applying the EMA Scaling Rule.

## H.7 Preventing collapse phenomena in DINO at scale

Until now, our representatives SSL method has been BYOL for reasons discussed in Section 3.4. Here, we will turn our attention to DIstillation with NO labels (DINO) (Caron et al., 2021), which has the update rule presented in Definition H.4.

**Definition H.4** (DINO Update). *DINO learns unsupervised features by matching predictions over emergent pseudo-labels of a student backbone and head $f(\,\cdot\,;\theta)$ to those of an EMA teacher $f(\,\cdot\,;\zeta)$ through a cross-entropy guided distillation procedure. DINO has a additional centering procedure, which is a form of batch normalization with momentum $\rho_c = 0.9$ which we do not scale using the*

Table 7: DINO ViT-B/16 Training hyperparameters.

|  | DINO ViT-B/16 |
| --- | --- |
| CIFAR10 Linear Probe Top-1 ($\rho_B = 0.996$) | 85.38% |
| CIFAR10 Linear Probe Top-1 ($\rho_B = 0.992$) | 86.96% |
| Weight initialization | `trunc_normal(.02)` |
| Normalization | Layer Norm |
| Learning rate schedule | Single Cycle Cosine |
| Learning rate warmup (epochs) | 50 |
| Learning rate minimum value | $1 \times 10^{-6}$ |
| Training duration (epochs) | 280 |
| Optimizer | AdamW |
| Optimizer scaling rule | Adam |
| Base ($\beta_1, \beta_2$) | (0.9, 0.95) |
| Base learning rate | $3 \times 10^{-4}$ |
| Base batch size ($B$) | 1024 |
| Base teacher momentum ($\rho_B$) | 0.992 or 0.996 |
| Base weight decay | 0.04 |
| Weight decay scaling rule | Linear |
| Weight decay skip bias | Yes |
| Center Momentum | 0.9 |
| Center Momentum Scaling Rule | None |
| Precision | `bf16` |
| Augmentation stack | DINO Multi-crop |

*EMA Scaling Rule. The update for the parameters of DINO is*

$$\theta_{t+1} = \theta_t - \eta \times \frac{1}{B} \sum_{x \in \mathbb{B}} \nabla_\theta \mathcal{L}(x; \theta_t, \zeta_t, c_t) \tag{78}$$

$$\zeta_{t+1} = \rho \, \zeta_t + (1 - \rho) \, \theta_{t+1} \tag{79}$$

$$c_{t+1} = \rho_c \, c_t + (1 - \rho_c) \, \mathbb{E}_{x'} \zeta(x') \tag{80}$$

$$\textit{with} \quad \mathcal{L}(x; \theta_t, \zeta_t, c_t) = H\big(f(x_1, \theta_t), f(x_2, \zeta_t) - c_t\big) + (x_1 \leftrightarrow x_2), \tag{81}$$

*where $H(a, b) \equiv -\sum_{m=1}^{M} p_m(a) \log p_m(b)$ is the cross-entropy between categorical distributions over M (emergent pseudo-)classes given logits $a, b \in \mathbb{R}^M$, $x_1$ and $x_2$ are two views of a single variate x, often produced by augmentations, and $x_1 \leftrightarrow x_2$ denotes symmetrization over $x_1$ and $x_2$.*

In practice, DINO employs multi-crop (Caron et al., 2021). We omit this detail for clarity of presentation, although we *do* use multi-crop in the experiments that follow.

Our interest DINO is due to the difficulty in its optimization[13], and in particular, preventing collapse phenomena in DINO at batch sizes above 1024, which is an open research problem. In this section, we will show that a combination of the EMA Scaling Rule (Definition 1.1) and Progressive Scaling (Definition 3.2) enable training of DINO beyond batch size 1024 without sacrificing performance.

**Hyperparameters** Base hyperparameters are presented in Table 7.

**Compute** *[This section has been redacted to preserve anonymity during the peer-review process. If this work is accepted, the full details compute used for these experiments, including: the experiments presented, hyperparameter optimization, and the development process, will be provided.]*

**Results** In Figures 24 and 25 we show the results obtained training DINO on CIFAR-10 with $\rho_B = 0.996$ and $\rho_B = 0.992$ respectively at the reference batch size of 1024. We employ smooth Progressive Scaling (Definition 3.2) between epochs 120 and 180.

At batch size 2048, the training loss matches the reference *only* when the EMA Scaling Rule is applied, whereas the run *without* the scaling rule diverges from the reference. The impact of this

---

[13]For an example, see `https://github.com/facebookresearch/dino/issues/43#issuecomment-881453515`.

1234 divergence is emphasized as we consider the larger batch size of 4096. Here. there is also a gap *with*
1235 the EMA Scaling Rule, however is approximately three times smaller than the gap *without* the EMA
1236 Scaling Rule.

1237 Additionally, we observe that using $\rho_B = 0.992$ yields higher Top-1 accuracy over $\rho_B = 0.996$, and
1238 in our experiments, using the EMA Scaling Rule *always* performs better in terms of linear probe
1239 performance than not using the scaling rule.

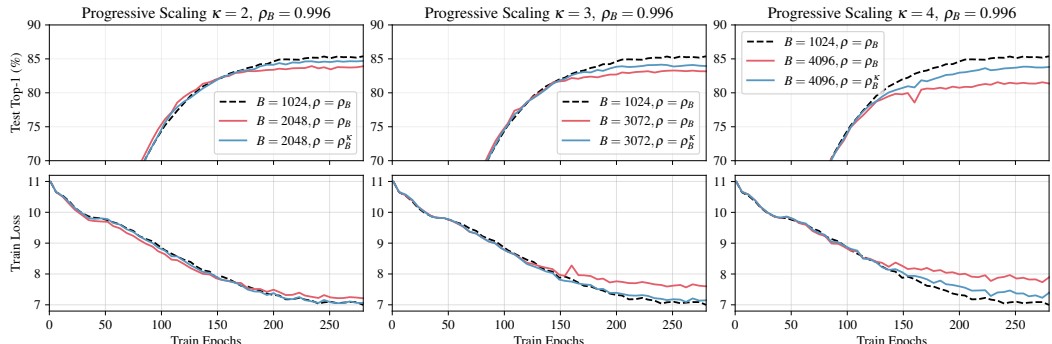

Figure 24: *DINO ViT-B/16 on CIFAR-10* for different scalings $\kappa$ and base teacher momentum $\rho_B = 0.996$. The baseline model ($\kappa = 1$, black dashed) uses batch size 1024 and center momentum $\rho_c = 0.9$, and is scaled up from batch size 2048 (left) to 4096 (right) with (blue, $\rho = \rho_B^\kappa$) and without (red, $\rho = \rho_B$) the EMA Scaling Rule. Between epochs 100 and 180 we scale the batch size using progressive scaling (Definition 3.2).

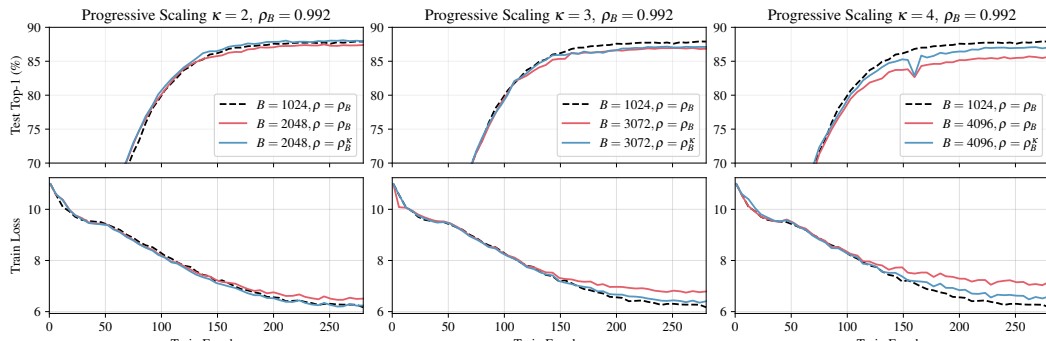

Figure 25: *DINO ViT-B/16 on CIFAR-10* for different scalings $\kappa$ and base teacher momentum $\rho_B = 0.992$. The baseline model ($\kappa = 1$, black dashed) uses batch size 1024 and center momentum $\rho_c = 0.9$, and is scaled up from batch size 2048 (left) to 4096 (right) with (blue, $\rho = \rho_B^\kappa$) and without (red, $\rho = \rho_B$) the EMA Scaling Rule. Between epochs 100 and 180 we scale the batch size using progressive scaling (Definition 3.2).

1240 In Figure 26 we show how the hyperparameters $\rho$, $B$ and learning rate change with the progressive
1241 scaling in Definition 3.2.

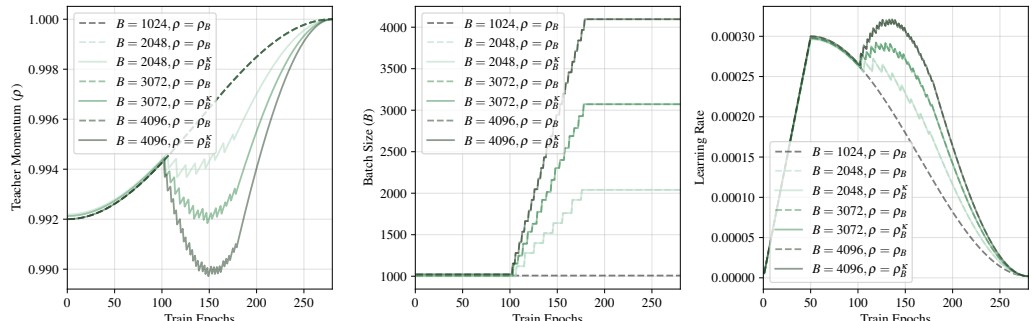

Figure 26: *DINO ViT-B/16 on CIFAR-10 for different scalings $\kappa$ and base teacher momentum $\rho_B = 0.992$.* We show how the hyperparameters $\rho$, $B$ and learning rate change with the Progressive Scaling in Definition 3.2. These hyperparameters correspond to the training runs in Figure 25. Those for Figure 24 are identical, with the exception of $\rho$ that starts at 0.996 instead of 0.992.

We also attempted to use a sharp batch size transition (Figures 27 and 28), which leads to the collapse pheonomena observed in prior work. This collapse happens with and without the EMA Scaling Rule. We suspect this is due to dynamics specific to DINO's early phase that are even more challenging to replicate under discretization than those of BYOL.

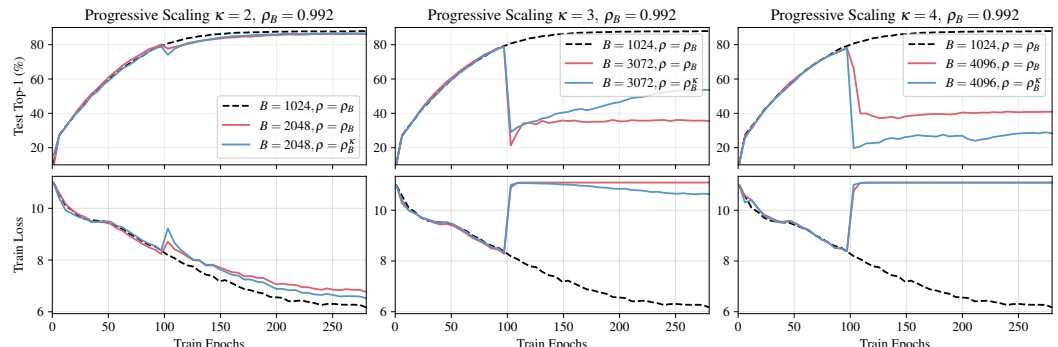

Figure 27: *DINO ViT-B/16 on CIFAR-10 for different scalings $\kappa$ and base teacher momentum $\rho_B = 0.992$.* The baseline model ($\kappa = 1$, black dashed) uses batch size 1024 and center momentum $\rho_c = 0.9$, and is scaled up from batch size 2048 (left) to 4096 (right) with (blue, $\rho = \rho_B^\kappa$) and without (red, $\rho = \rho_B$) the EMA Scaling Rule. Progressive Scaling is employed with a sharp transition at epoch 100, leading to a collapse phenomenon.

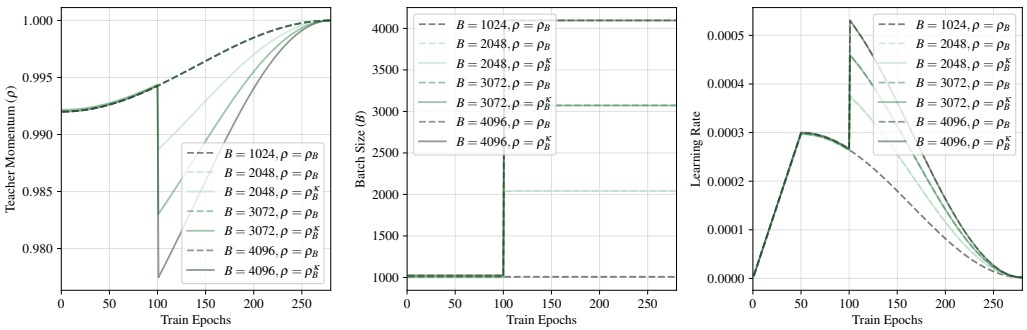

Figure 28: *DINO ViT-B/16 on CIFAR-10 with $\rho_B = 0.992$ and a sharp transition in batch size at epoch 100.* We show how the hyperparameters $\rho$, $B$ and learning rate change with sudden scaling. These hyperparameters correspond to the training runs in Figure 27.

Our results in this section show it is possible to scale DINO to large batch sizes *without* sacrificing performance by using *both* the EMA Scaling Rule and Progressive Scaling, providing the batch size schedule of Progressive Scaling is not sudden. This resolves an open problem in SSL research.