# OpenReview forum: "How to Scale Your EMA"
_NeurIPS.cc/2023/Conference — NeurIPS 2023 spotlight_

### Official Review · Reviewer_nnfC · 2023-07-06

**Soundness:** 3 good
**Presentation:** 4 excellent
**Contribution:** 3 good
**Rating:** 7
**Confidence:** 4

**Summary:**

In duality to previous work on learning rate scalings, the authors present a novel scaling strategy for the momentum parameter in teacher-student settings, i.e. how to adapt the momentum in the learning dynamics when the batch-size is changed. The work puts an emphasis on the large batch-size setting, which is of particular interest due to its computational efficiency in practice. The obtained scaling is validated through a theoretical analysis and numerous experiments in different settings demonstrate that (a slight modification of) the scaling indeed preserves performance at large batch-sizes.


**Strengths:**

1. The paper﻿ is very well-written and easy to follow. The problem of large batch-size training is very relevant as it enables more efficient training (especially when distributing across multiple GPUs) and the resulting scaling rule is very simple. I believe that such insights can be of tremendous use to practicioners, especially if hyperparameter tuning is too expensive, especially in the setting of SSL where the setups are usually already very brittle to hyperparameter changes.
2. The experimental setup is quite broad, including various student-teacher settings such as Polyak-Ruppert averaging, pseudo-labeling and self-supervised learning.
3. The technique is nicely motivated through a theoretical analysis, strengthening the particular choice of scaling further. ﻿


**Weaknesses:**

Some important aspects of the experimental setup are not well explained:

1. In the real data experiments (i.e. Fig 2, 3, 4, 5, 6), what does “No EMA Scaling Rule” entail? Does it still apply the “Learning-rate Scaling Rule”? If it does not, then it would be very hard to draw any conclusions regarding "EMA Scaling” as it is not clear how much of the “heavy-lifting” the learning rate scaling is doing. I hope the authors can clarify this.
2. One of the main motivation of the works is to make methods like SSL more computationally effective by ensuring that training them with large batch-sizes does not deterioate performance. In order to make it work in practice, the authors need to resort to “progressive scaling”, where the batch-size is only gradually increased after the warm-up phase. How long is the warm-up period exactly (during which a small batch size is used) and how quickly is the batch-size scaled up afterwards? How much time is lost compared to the “EMA Scaling without Progression”?
3. Another aspect which remained a bit mysterious is the effect of the baseline momentum. For Polyak-Ruppert averaging for instance, its role seems very crucial. If e.g. rho_base were set to 1, the teacher would always be equal to the student and at inference time I would not expect any sort of batch size effect, given that the learning rate scaling correctly works. I would appreciate if the authors could elaborate more on this initial choice of momentum.


**Questions:**

1. The CIFAR10 results in Fig. 5 seem surprisingly bad (around 55%-58%).
2. Are you using a linear or a K-NN probe for the SSL experiments?
3. After the warm-up phase with progressive scaling, how well is the model already performing at this point? I.e. how much is still gained in the large (or maximal) batch-size regime?
4. I find it very surprising that the model is so sensitive to the momentum parameter in Polyak-Ruppert averaging. How well is the student performing at the end of training and how much is actually gained by averaging? I would be very surprised if the non-ideally averaged teacher would perform inferior to the student at the end of training.


**Limitations:**

The authors have addressed the limitations of their work.

---

> ### Author Rebuttal · Authors · 2023-08-09
>
> Thank you for taking the time to read our work. We are happy you found the paper easy to read, appreciate the empirical and theoretical validation of the EMA Scaling Rule, and recognize the broad applicability of the result to many modern ML settings.
>
> ### Weaknesses
>
> #### *1. No EMA Scaling Rule*
> We will clarify that this means the optimizer is still scaled (i.e. SGD Scaling Rule for learning rate is applied), while no scaling rule is used for the EMA, i.e. the same momentum is used for different batch sizes. This means that the effect we see is due only to the EMA parameter, and not an unscaled optimizer.
>
> #### *2. Computational effectiveness of Progressive Scaling*
> If one needs to use Progressive Scaling, a greater wall-clock benefit is realized an earlier transition, but note that one does not need to begin scaling from $\kappa=1$. E.g. for pseudo-labeling experiments in the case of scaling $\kappa=16$, 15% of training time is done first at $\kappa=8$ followed by 85% of training time at $\kappa=16$ (see Appendix, Figure 18), maintaining most of the wall-clock time benefit. We also note that for self-supervised learning, one of our main use cases, performance benefits are given by long training duration; the time cost of Progressive Scaling can be made negligible by training for longer.
>
> The warm-up phase should happen once the optimizer SDE approximation is valid. Malladi et al. 2021 showed the Adam Scaling Rule is only valid after finite training and a gradient noise criterion is met. A practical option could be to track this criterion, and begin Progressive Scaling when it is met. In this work, our choices are empirical, and found that the beginning Progressive Scaling after the learning rate warmup phase has completed works well.
> #### *3. Effect of baseline momentum*
> The baseline momentum at a given batch size affects model performance. In all our experiments, the baseline momentum is taken from reference recipes (and cited accordingly), or are found using max-performance hyperparameter search. We, now also present a baseline momentum sensitivity analysis in the rebuttal pdf (Figure 1) which we will include in the paper.
>
> On pathological choices of base momentum, there are two. The first is $\rho_B = 1$, which means that the EMA model (teacher) is always equal to its initial state, independent of scaling. This is the same as saying the EMA model is infinitely slow, which no change in discretization modification can alter. The second pathological is $\rho_B = 0$, which means that the EMA model is always equal to the target model (student). This is the same as saying the EMA model is infinitely fast, again which no change in discretization modification can alter. No momentum $rho_B$ can be scaled onto either of these values exactly, but only asymptotically, i.e. $\lim_{\kappa\to\infty}\rho_B^\kappa = 0$ and $\lim_{\kappa\to0}\rho_B^\kappa = 1$ for $0<\rho_B<1$, with the latter limit corresponding to the SDE limit, providing other quantities are scaled appropriately. Each pathological choice does not have the properties of a moving average, and produces trivial results; there is only initial-state dependence, or there is no EMA respectively.
>
> ### Questions
> #### *1. CIFAR10 results are not competitive*
> We agree that these results are not competitive. The small model (ResNet 18) and short training duration were chosen to produce an experimental setup that enabled quick iteration, and the SGD optimizer was chosen as it has as known optimizer scaling rule. This allowed us to precisely probe the claims of the EMA Scaling Rule in the challenging setting of BYOL, where the EMA network provides the learning signal for the whole optimization process.
>
> Competitive results can be achieved by: using a larger model (e.g. ResNet 50), training for longer, or using an optimizer like LARS. Performing these changes gives us a model consistent with the reference BYOL implementation (App. H.6, Fig. 23).
>
> We also note that the EMA Scaling Rule should also work for sub-optimal hyperparameter settings, which this experiment confirms. We will clarify in the text why we chose this particular setting.
> #### *2. Which probe do you use in SSL?*
> We will clarify we are using a linear problem in all SSL experiments.
> #### *3. In Progressive Scaling, how much is gained in the large batch regime?*
> If Progressive Scaling is successful, the model will reach a final performance that is the same as the unscaled model; there is no particular gain one can expect after Progressive Scaling, as one could progressively scale at any point (scaling at the start would give 100% of the benefit, and scaling at the end would give 0%). In our pseudo-labeling settings, we see WER improve from 20% to 10% (Fig. 18) after warmup, and in SSL we see BYOL ViT accuracy increase from 64% to 75% (Fig. 23) after warmup for the last Progressive Scale change.
> #### *4. Sensitivity to momentum in Polyak-Ruppert averaging*
> In Polyak-Ruppert averaging, we do see gains compared to the student. In vision we see a gain of $\sim0.3$% (Table 4) and in ASR we improve 1-2% in WER (see App. Fig 14, 15), which is significant. The dramatic fall-off in Fig. 2 happens as the training duration of 90 epochs is not large, and is taken from the reference (Goyal et al. 2017). When scaling happens, with each increase in $\kappa$ the EMA network should to move exponentially faster in order to adapt with the new discretization. For scaling $\kappa\geq 4$, the EMA network without the EMA Scaling Rule moves too slowly to take advantage of any learning the target model has done.
>
> ### Summary
>
> Again, we appreciate the time you took reading our work and sharing your valuable comments and questions. We will integrate our responses, along with additional clarifications, into the paper. These enhancements will contribute to the overall readability, clarity and completeness.

---

> > ### Comment · Reviewer_nnfC · 2023-08-15
> > **Response**
> >
> > I thank the authors for their efforts and clear explanations! My concerns are addressed.

---

### Official Review · Reviewer_6nqM · 2023-07-06

**Soundness:** 3 good
**Presentation:** 2 fair
**Contribution:** 1 poor
**Rating:** 6
**Confidence:** 4

**Summary:**

This work proposes a scaling rule for EMA in optimization and empirically evaluates its effectiveness across a range of tasks and optimizers. Authors find that proposed method enables large batch training using 24k batch sizes with a 6x wall-clock time reduction.

**Strengths:**

- Idea of formulating a scaling rule for EMA is novel and not one I have seen much in the recent deep learning / optimization literature
- Theoretical motivations are clear to understand
- Strong empirical study

**Weaknesses:**

- Applicability to modern optimizers is still not clear. Despite the authors discussing some connections to Adam, I think an in-depth discussion of how the theory/scaling rule relates to Adam is not fully clear.

**Questions:**

- My main question is regarding the motivation behind such a scaling rule for EMA. Have the authors verified that such a scaling rule is infact needed since for most practical problems, hyper-parameters are just fixed and no such rule is used, still giving a good performance for most models. Is there a fundamental gap (that has potential to improve model quality or speed substantially) the authors are trying to address here?
- It looks like there is still some tuning required for the scaling rule to work? Can the authors clarify how much and sensitivity to larger models and different architectures?

**Limitations:**

- Key questions regarding broad applicability of this method are not answered (such as generality of the solution to varying model types, model scales, etc). Therefore it is not clear how impactful this method is.

---

> ### Author Rebuttal · Authors · 2023-08-09
>
> Thank you for taking the time to read our manuscript. We are are happy you find our formulation novel, the theoretical motivations clear, and the study design strong.
>
> We begin by clarifying a potential misunderstanding. Our work is devoted to the behaviour of exponential moving averages (EMAs) of model parameters, and *not* EMAs used within gradient-based optimizers such as Adam. Investigations for this direction already exist (see e.g. Malladi et al. (2022)). Understanding the dynamics of Adam is *not* a purpose of this work. We do conduct some experiments where we take an EMA of the sequence of parameters produced by the Adam optimizer (e.g. Fig. 3, 4), and discuss some related theoretical aspects in Appendix D.
>
> We note that the review contains concerns regarding the broad applicability of an EMA Scaling Rule (ESR), and contains a number of questions we will now address, beginning with a restatement of our contribution.
>
> An optimizer scaling rule allows the same training dynamics to be produced at different experimental scales (batch sizes). A common use is producing similar models in less wall-clock time. Our work derives an ESR, extending this training speedup to methods which use model parameters exponential moving averages, which include (but are not limited to) state-of-the-art semi-supervised methods, like continuous pseudo-labeling, and self-supervised methods like BYOL and DINO.
>
> ### Weaknesses
>
> #### *Applicability to modern optimizers*
> Our work does not deal with the momentum in gradient optimizers (Footnote 1), and scaling rules for Adam are already well-studied (Section 4). Our focus is the momentum of the model parameter EMA, whose scaling is independent to an optimizer choice (Line 104 & Appendix D), and is guaranteed, providing the gradient optimizer scaling is known (Appendix B, Limitation 2).
>
> We empirically verified that the EMA Scaling Rule works with SGD (with/without momentum), Adam, AdamW, and LARS optimizers, which are popular modern ML choices. Moreover, we empirically focus on different domains (toy data, vision, speech) and different models (ResNet and variants of Transformers). Although some of these optimizers do contain an EMA (such as Adam), these are not model parameter EMAs, and are not the focus of our work.
>
> ### Questions
>
> #### *What is the practical requirement for an EMA Scaling Rule?*
>
> For example, Fig. 2 shows EMA tracking performance goes to zero under batch scaling when going up only $4\times$ in batch size if the ESR is not used. One could find the ESR-predicted value through a hyperparameter search at the larger batch size, but this comes with large computational cost compared to already knowing the value. We note that our method still requires a *good momentum value* to be known at some computational scale - so at least one search, or one reference recipe is required. After this, the good value at a new scale can be determined by the ESR.
>
> #### *Is there tuning required for the EMA Scaling Rule to work?*
>
> We discuss in Limitations (App. B) scenarios when the ESR will and will not work. In summary, when the corresponding optimizer scaling rule is known, and discretization error is not large, then the EMA Scaling Rule will work, independent of any other hyperparameter choices. We have presented additional hyperparameter ablations in the rebuttal pdf highlighting this (Fig. 1, 2, 3).
>
> ### Limitations
>
> #### *Broad applicability of method*
> We have restated that our objective is to study the behavior of the model parameter EMA, which is independent of optimizer choice, model class, and problem domain. We also note that, beyond the theoretical guarantees, we empirically validated the EMA Scaling Rule for a large array of models, domains and modern optimizer choices. We hope that the combination of these two answers this point.
>
> ### Summary
>
> Thank you again for taking the time to review our work. We hope a restatement of our overall goal, and the reemphasis of the wide empirical validation we have performed  addresses the limitation you have raised regarding broad applicability.

---

> ### Author Response · Authors · 2023-08-15
> **Check for outstanding concerns**
>
> Dear Reviewer 6nqM,
>
> We thank you again for taking the time to read our manuscript and providing feedback. We hope the rebuttal and additional experiments we provided were helpful.
>
> Please let us know if there are still outstanding concerns regarding broad applicability, or any other concerns raised in your review. We will be very happy to address them.
>
> Thank you,
>
> The Authors

---

> ### Comment · Reviewer_6nqM · 2023-08-21
> **Feedback after author feedback**
>
> Thanks to the authors for their response and detailed clarifications. I did another pass on the paper and based on author clarifications, I find the contributions to be well-supported empirically and theoretically. I still have some concerns about the presentation and organization of the paper, but overall I am willing to raise my score as my technical concerns have been mostly addressed.

---

### Official Review · Reviewer_dfpr · 2023-07-07

**Soundness:** 3 good
**Presentation:** 2 fair
**Contribution:** 3 good
**Rating:** 7
**Confidence:** 4

**Summary:**

The exponential moving average (EMA) of model parameters is used in many contexts, including to improve robustness in supervised learning, and as an important component of self-supervised learning. Given model parameters $\theta_t$ at iteration $t$, the exponential moving average $\zeta_t$ is updated as $\zeta_{t+1} = \rho \zeta_t + (1 - \rho) \theta_t$, where $\rho$ is the EMA coefficient. This paper proposes a scaling rule that dictates how $\rho$ should be modified to preserve training dynamics when the batch size is modified. In particular, when the batch size $B$ is scaled by a factor of $\kappa$ yielding $\kappa B$, the EMA coefficient should be exponentiated to the power of $\kappa$, yielding $\rho^\kappa$.

The authors justify this rule by analogy to the learning rate vs batch size rule from (Goyal et al., 2017), under the assumption that the gradients change extremely slowly between successive steps in training. They also give a justification using an SDE interpretation.

Empirically, they evaluate the EMA scaling rule on several real-world tasks, including supervised learning for image classification and speech recognition, semi-supervised learning for speech recognition, and self-supervised image representation learning. In each case, they compare the loss curves of: 1) a baseline model trained with batch size $B$ and EMA coefficient $\rho$; 2) a model trained with batch size $\kappa B$ and the same EMA coefficient $\rho$; and 3) a model trained with batch size $\kappa B$ and modified EMA coefficient $\rho^\kappa$. They show that using the scaling rule $\rho^\kappa$ produces more similar loss curves to the baseline model than the un-modified $\rho$.

**Strengths:**

* The authors present a toy example where they apply the EMA to a noisy quadratic problem, where the "batch size" is simulated by adjusting the gradient variance. On this task, they compute the optimal EMA coefficient $\rho^*$ for various scaling factors $\kappa$ up to 1024, and show that the $\rho^\kappa$ coefficient nearly matches $\rho^*$ with respect to the approximation error. I appreciate the inclusion of such a toy example.

* The empirical evaluation considers diverse tasks, including both vision (ImageNet1K classification with a ResNet) and language (speech recognition with a Transformer) tasks. These experiments also consider different optimizers (SGD and Adam).

* The authors also evaluate progressive scaling, in which they increase the batch size during training and simultaneously scale the LR and EMA coefficient. They found that in one case (training a BYOC Vision Transformer), this was necessary to obtain decent performance.

* The plots are generally well-made and clear, with good labels.

**Weaknesses:**

* One of the main drawbacks of the paper is that it is not clearly written. The core idea is very simple, and should be simply explained. However, most of the paper is needlessly obfuscated, making it hard to read. The majority of the definitions (2.1, 2.2, 2.3, and 3.1) are not novel, and are basic concepts that would not typically be presented as formal definitions. This reads as though the authors are trying to add math and formality to the paper, when it is not necessary.

* The plots in Figure 1(a) are not clearly presented, as the legend overlaps with more than half the figure.

* While the experiments use several architectures and datasets, they all measure the same thing, comparing the loss values of models trained with either $\{ (\kappa B, \rho^\kappa), (\kappa B, \rho), (B, \rho) \}$. There is very little diversity in this respect. Are there other things that can be done with the ability to match scaling behaviors, like doing hyperparameter optimization in the small-batch regime (which could be run in parallel on multiple devices rather than requiring a large amount of compute for a single experiment) to find the optimal $\rho$, and then scale it to the large-batch setting using $\rho^\kappa$?

* For the Transformer experiment in Figure 3, it is hard to see the long-term behavior of the loss curves. It would help if the y-axis was log-scaled. Here, the red curve ($\rho$ unmodified) seems to do just as well as $\rho^\kappa$, especially in terms of final performance.

* Throughout the paper, the notation $\times$ is used for simple scalar multiplication. This is non-standard in ML literature, and confusing to the reader, as $\times$ looks like a cross product. There is no reason to write $\eta \times \frac{1}{B}$ rather than simply $\frac{\eta}{B}$.

* In Eq. 6, it is unclear what $\Theta$ and $Z_t$ are, as they are not defined.

* Probable typo in L163: "scaled additive $b > 0$" --> "scaling $b > 0$"

* The base results (black dashed lines) in the real-world experiments do not seem to have bands for the mean and std over three runs (only the red and blue curves have bands).

**Minor**

* The math font looks non-standard.

* I do not think that the abstract does a good job describing the exponential moving average.

* The term "model EMA" should be more clearly defined at the beginning, as "the EMA of the model parameters."

* In the abstract, it is not clear what is meant by "Prior works have treated the model EMA separately from optimization."

* In the abstract, what is meant by "optimally a $6\times$ wall-clock time reduction"? Is this the actual wall-clock reduction achieved in practice, or under some idealized scenarios? This should be clarified.

* L32: "These weights are updated through by a momentum hyperparameter." --> Bad grammar "through by," and it does not make sense to say that the weights are updated "by a momentum hyperparameter."

* To improve clarity, the introduction should introduce the mathematical form of the EMA, $\theta' = \alpha \theta' + (1 - \alpha) \theta'$. The expression for the EMA needs to be introduced before Definition 1.1 because otherwise it is not clear what $\rho$ corresponds to and how it is used.

* Definition 1.1 should simply be labeled Definition 1, because there is no Definition 1.2. In Def. 1.1, it is confusing to say "scale other optimizers." Should this be "scale other optimization hyperparameters"?

* On Line 64, write out "self-supervised learning" instead of saying "SSL" because in this context it could be misinterpreted as standing for "semi-supervised learning."

* What is the purpose of including the third row in Eq. 4, which keeps the gradient $g$ unchanged? Is this just to highlight the assumption that the gradient changes very slowly, so that the first two rows hold?

* Typo: L115, "typically use to" --> "typically used to"

* L278 Typo: "a model EMA alters" --> "a model EMA that alters"

* L307 Typo: "only when by the EMA" --> "only by using the EMA"

* L210 Typo: "In summary, ASR" --> "In summary, this ASR"


**Questions:**

* In many of the experiments, $\rho$ is very large: in Figure 1(a), $\rho=0.9999$, in Figure 2 $\rho = 0.9999$, in Figure 3 $\rho = 0.9995$, and in Figure 4 $\rho=0.9999$. Why are these EMA coefficients so large? Are these optimal values at the base batch size, and how were they chosen? How do these experiments behave when $\rho$ is smaller, like $\rho=0.9$? As a more general point, because $\rho$ is exponentiated, it seems that there could be numerical overflow or underflow issues depending on the initial value of $\rho$ and the scaling $\kappa$.

* In Eq. 5, what form does the gradient noise distribution $\mathcal{E}\_{\sigma}(\theta_k, \zeta_k)$ take? Is it Gaussian? If so, it would be better to use standard notation $\mathcal{N}$. What does it mean that the gradient noise distribution is "assumed to be zero-mean and variance $\Sigma(\theta_k, \zeta_k)$ independent of $\sigma$"? If the variance is independent of $\sigma$, why use the notation $\mathcal{E}_{\sigma}$?

* See also questions from the weaknesses box.


**Limitations:**


* For the real-world experiments, it would be important to report scaling behavior beyond the factors that are shown in the paper. Why is the maximum scaling factor $\kappa = 8$? Why not continue the scaling to see if the scaling rule continues to hold, and determine where it breaks down? Does it work if $\kappa \in \{ 16, 32, 64, 128, \dots \}$? If the reason for not using larger factors is that the base model already used a large minibatch size ($B=1024$), then why not train a base model with a small batch size $B=16$ such that large $\kappa$ can be evaluated? In addition, because the scaling rule is meant to hold for both increasing and decreasing the batch size, why not have an example reproducing the results of a model trained with $B= 2048$ using a much smaller batch size like $B=16$? This would be a factor of $128$ decrease in batch size.

* This also relates to the question about numerical overflow/underflow, because if the original EMA coefficient is $\rho = 0.9999$ and $\kappa = \frac{1}{128}$ then the adapted coefficient would be $\rho^{\frac{1}{128}} = 0.99999922$. What are the limits to the approach, and what are its limitations?

---

> ### Author Rebuttal · Authors · 2023-08-09
>
> Thank you for taking the time to read our work. We are happy you found the plots clear, appreciate the mixture of controlled toy and diverse real world investigations, and find utility in enabling large batch training of BYOL with Vision Transformers.
>
> ### Weaknesses
> #### *Obfuscation*
> The purpose of numbered definitions is to avoid ambiguity for readers as there are many concepts, algorithms and scaling rules in the text. Each definition is accompanied with appropriate context, and relevant citations where it was introduced by prior works.
> #### *Plots in main text all the same thing*
> The choice of reported metrics is three-fold: 1) they are used in prior works studying scaling rules, e.g. Maladi et al. 2021; 2) a unified comparison across models and domains makes it clear the EMA Scaling Rule is agnostic to them; and 3) These metrics are sufficient to show training dynamics preservation - validating the main claims of the paper. Additional domain-specific metrics are reported in the Appendices.
> #### *Other applications of the EMA Scaling Rule*
> Where hyperparameters are not from prior works (cited when this is the case), the $\rho$-optimization procedure you describe is exactly what we did. E.g. in supervised vision, take 8 GPUs and optimise $\rho_B$ at low batch size. Then scale to larger batch sizes (e.g. across 32 GPUs, with the same per-gpu batch size), and verify the scaling law by comparing $\rho_B$ to the predicted momentum $\rho_B^\kappa$ ($\kappa=4$ in this example). Total compute budget (GPUh) is the same, whereas training wall-clock time is reduced at higher batch sizes.
> #### *Fig 3*
> We note that final performance is not the main property we are interested in (we agree that in Fig 3, with/without the EMA Scaling Rule have similar end performance). Our goal is to match the whole trajectory, which does not happen without the EMA Scaling Rule. See also App. H.5 and Fig 22 for a detailed discussion.
> #### *Eq 6*
> $\Theta$ and $Z_t$ are SDE variables.
> #### *L163*
> $b$ is the coefficient of noise scaled by the gradient norm. Noise is additive so $b$ is the “scaled additive noise” coefficient.
> #### *Bands for baseline*
> We will show baseline bands, see rebuttal pdf for examples.
>
> ### Minor
> #### *Purpose of 3rd row in Eq. 4*
> This is the constant gradient within-time interval assumption of Goyal et al. 2017. This row contributes to gradient updates of the tracking model on the top row through repeated applications of the stepping matrix. We present a full breakdown of this in App. E.1.
>
> ### Questions
> #### *Choice of momenta*
> Momenta are chosen for performance and are model/domain dependent. A popular choice for SSL methods is $\rho_B=0.996$ at $B=4096$ (or equivalently $\rho_B=0.9997$ at $B=256$) (Grill et al. 2020). We stress batch size must be considered when determining if a momentum is large (App. C.1). Momenta in our paper are taken from reference recipes and cited, or determined by max-performance hyperparameter search. We also note that the toy quadratic experiments do cover a large momentum range, with the smallest momentum evaluated $0.99^{1024}\approx 0.00003$ in Fig. 7, far from 1 (EMA Scaling Rule breaks here). Finally, we present a momentum sensitivity analysis in the rebuttal pdf (Figure 1).
> #### *Overflow/underflow*
> Analysis of overflow/underflow, particularly for [IEE-754 floating point values](https://standards.ieee.org/ieee/754/6210/), is non-linear due mechanisms like gradual underflow.
>
> In our setting, $\rho$ will not suffer from practical overflow or underflow issues through exponentiation, as FP32 precision allows a maximum $\rho=1-\epsilon$, or minimum $\rho=\epsilon$, $\epsilon\approx 1.2\times 10^{-7}$. Taking SSL as a baseline, this corresponds to $\kappa\approx 32K$ or $\kappa\approx 4K$ respectively, the former being below $B=1$ and the latter giving $B\approx 9M$ which is beyond current hardware feasibility, and beyond the breakdown of known optimizer scaling rules.
>
> To examine how momentum may induce numerical errors in practice, we present Figure 4 in the rebuttal pdf. This shows a linear regression model with a Polyak-Ruppert average. We study $\rho\in[0.99, 0.99999]$ and track the maximum absolute difference between FP32 model weights and weights in {BF16, FP16, a second FP32 run}. This acts as a proxy for overflow and underflow. When model weights and EMA weights are FP16 (never done in practice) we observe an increasing variance only for FP16 as the value of $\rho$ becomes closer to 0.99999 — BF16 and FP32 are stable here. We stress all experiments presented in the paper store weights (network and EMA) in FP32 and use automatic-mixed precision to cast them to BF16 during training, so never encounter overflow/underflow induced by momentum.
>
> #### *Scaling behavior*
> We reported real-world experiments for batch sizes up until matching breaks in all cases (Fig 2, 5, 16, 18) while for toy problems — scaling ranges up to $\kappa=1024$ (see App. F.1). We now provide a scaling sensitivity analysis in the rebuttal pdf (Fig. 3) where scaling is matched down to $\kappa=1/8$ (batch normalization used in the ResNet 18 results in slight mismatch at low batch size). Moreover, $\kappa$ is only notational; the batch size or discretization level matters for SDE approximation: e.g. scaling from batch 1024 to 4096 ($\kappa=4$) means that we can scale also from batch 1 to 4096 ($\kappa=4096$) as this limit is guaranteed by the properties of the SDE, providing there are no confounding effects like batch normalization. Experiments using $B=1$ will be $4096\times$ slower to run in practice, thus impractical to perform for real-world problems.
>
> ### Summary
>
> Thank you again for taking the time to read our work, as well as for the many pointers, suggestions, and typo highlighting.  We will incorporate these changes in the paper revision to improve readability, as well as the additional content prompted by your feedback.

---

> > ### Comment · Reviewer_dfpr · 2023-08-15
> > **Response to Rebuttal**
> >
> > I have read the other reviews and the authors' rebuttal. I thank the authors for their responses, and for performing additional experiments in the rebuttal PDF. I raised my score to 7. I think that the EMA scaling rule is interesting and the experiments are good. The writing is still more obfuscated than is necessary, and the paper could benefit from some minor rewriting, which hopefully can be done for the final version.

---

### Official Review · Reviewer_bHsu · 2023-08-01

**Soundness:** 3 good
**Presentation:** 3 good
**Contribution:** 3 good
**Rating:** 7
**Confidence:** 2

**Summary:**

This paper talks about the scaling rule for model EMA, and proposes that the momentum coefficient of EMA update should be scaled exponentially with the batch size instead of linearly. The authors first informally derive such rule (Equation 4) under the assumption that the gradients change slowly for a few steps. Then the authors theoretically validate such rule by considering the SDE limit of the optimization process (Equation 6) and prove that the optimization trajectory of different batch sizes could still have small difference after we exponentiate by the batch size ratio $\kappa$.

For the experiments, the authors first use a noisy parabola and verify that exponentiating the model momentum term would approximate well with the empirical optimal momentum, as illustrated by Figure 1.b. The authors also consider ResNet and transformer for a supervised task, and the results are the same. However, there is a growing gap between the exponentiated momentum with large batch versus the original momentum with small batch (Figure 2), which is attributed to the breakdown of Adam scaling rule. In the final self-supervised experiments, the authors found a performance gap between the scaled model with EMA and the baseline, and thereby introduce progressive scaling which slowly increases batch size during the early stage of training to match the training dynamics at larger batch size.

**Strengths:**

The paper is well-written and easily understandable, and the experiment section is well-designed that contain both simple convex task and a broad spectrum of deep learning tasks. The idea is built on top of Malladi et al. (2022)'s work but the contribution is clear. The application of scaling model EMA is broad, and I believe this paper make a nice contribution behind it.

**Weaknesses:**

In figure 6 with subfigure progressive scaling kappa = 6, after applying the progressive scaling, both purple (without EMA scaling) and orange line (with EMA scaling) performs roughly the same. However, the other subfigures (that varies $\kappa$) in figure 6 show that EMA scaling is helpful besides the progressive scaling. I might misunderstand but could you provide an explanation why $\kappa$=6 would be quite different from $\kappa$=4 and $\kappa$=8?

In line 210, section 3.2, could you elaborate more on the intuiton behind the sentence: *Inspecting the train loss leads us to conclude this is due to a breakdown of the Adam Scaling Rule*? Such clarification would be non-trivial as the gap is growing when we increase the scaling factor.

An additional ablation study on both momentum coefficient + learning rate:
The model EMA momentum coefficient used in this paper is too close to 1. It would be nice to have an ablation study on the EMA scaling rule vs. the optimal scaling factor when we vary the EMA momentum coefficient away from 1.
Similarly, we would expect when the learning rate increases, the EMA scaling rule will break down, but it would be nice to quantify such phase change even with a convex task.

Minor:
In footnote 5, the eqn 68 might actually refer to eqn 74.

**Questions:**

The questions are raised above.

**Limitations:**

The authors acknowledge that there is still a $O(\eta * \beta_\rho)$ difference for the scaled model with the unscaled one, but didn't have an ablation of the impacts from the learning rate and the impacts from the base momentum coefficient. It would be nice to see such study somewhere.

---

> ### Author Rebuttal · Authors · 2023-08-09
>
> Thank you for taking the time to read our work. We are happy you found the paper easy to read, acknowledge the contributions made, and see the broad applicability of an EMA Scaling Rule.
>
> ### Weaknesses
>
> #### *Figure 6 - ViT scaling*
> We also note that the use of an EMA Scaling Rule is more important in some circumstances than others. Intuitively, the EMA Scaling Rule is less important when the value of the EMA momentum is less important (such as Polyak Ruppert averaging).
>
> To investigate in detail, we conducted the sensitivity analysis in Appendix H.5. We found that for good performance, it is sufficient to use a momentum within an acceptable region $\rho_{\text{min}}\leq \rho \leq \rho_{\text{max}}$, where $\rho_{\text{min}}$ and $\rho_{\text{max}}$ each achieve a minimum performance. Note that this is different to choosing a momentum whose training dynamics are the same as one from a reference batch size $B$ (the central goal of the EMA Scaling Rule). Application of the EMA Scaling Rule shows that the range of acceptable momenta actually increases with scaling (see Fig. 22).
>
> This explains the observation for Figure 6, $\kappa=6$: after 60 epochs at $B=4096$, the reference momentum $\rho_B$ enters the acceptable region for $B=24576$; it is not necessary to apply the EMA Scaling Rule in this specific setting to get good performance. We stress however, this strategy is not guaranteed to work, and the momentum given by EMA Scaling Rule will always be at least as good as this choice.
>
> We will clarify text around Figure 6 and more strongly highlight Appendix H.5.
>
> #### *How do you know the Adam Scaling Rule is broken?*
>
> In this experiment, we have used both optimizer scaling (Adam in this case), as well as EMA Scaling. This particular section concerns Polyak-Ruppert averaging, which lets us check independently whether the Adam Scaling Rule and the EMA Scaling Rule are holding. In Appendix G we present Figure 14 which includes Train Loss and test-other WER. Both quantities depend *only* on the target model, and are *not* related to the model parameter EMA. For $\kappa>2$ (i.e. $\kappa=4$ shown), the loss and WER curves deviate from the baseline. This is a breakdown of the Adam Scaling Rule as only the Adam optimizer is involved in the training dynamics leading to these quantities.
>
> #### *Additional momentum and learning rate analysis*
>
> We agree studies of the validity of the EMA Scaling Rule for a wider range of hyperparameters are very useful. In the quadratic setting, analysis of $\rho_B\in\{0.99, 0.999\}$ is presented in Appendix F.1 alongside higher-dimensional counterparts.
>
> In the rebuttal pdf, we present a learning rate $0.01\leq \eta_B \leq 0.5$ and momentum $0.841\leq \rho_B \leq 0.997$ ($B=1024$) sensitivity analysis for the challenging setup of BYOL + SGD in Figures 1 and 2. Note that the lower bound corresponds to  $\rho=0.5$ at $\kappa=4$ ($B=4096$), and is a very low momentum value. We see that the EMA Scaling Rule is robust to modifications of the learning rate, and has good scaling behaviour down to momenta of $\rho_B=0.946$ at $B=1024$ ($\rho=0.8$ at $B=4096$) in this particular setting.
>
> These results and related additional analysis will be included in a new version of the paper.
>
> #### *Minor: In footnote 5, ...*
> Thank you for finding this, it will be corrected in a new version of the paper.
>
> ### Limitations
>
> #### *Additional momentum and learning rate analysis*
> See above.
>
> ### Summary
>
> Again, thank you for the time you took reading our work and providing your valuable questions. We will integrate our responses, along with the requested sensitivity analysis into the paper, which we hope will address the raised limitation. These enhancements will contribute to the overall readability and provide a complete full picture for future readers.

---

> > ### Comment · Reviewer_bHsu · 2023-08-21
> > **Thanks for the rebuttal**
> >
> > I have read the authors' rebuttal and they have addressed all of my concerns. Thanks for the detailed explanation and well-designed experiments! I am willing to raise my score to accept.

---

### Author Rebuttal · Authors · 2023-08-09

We thank all our reviewers for taking the time to review our work. We will adapt the paper based on their insightful comments, feedback, and questions.

There were a number of questions prompting additional investigations. We provide here a pdf containing the results of those investigations. Its contents are:
- Figure 1: A momentum sensitivity analysis
- Figure 2: A learning rate sensitivity analysis
- Figure 3: A scaling analysis to low batch sizes
- Figure 4: A numerical precision analysis.

Specific context and discussions are presented in the corresponding review rebuttals.

Many thanks,

The authors

---

### Decision · Program_Chairs · 2023-09-21

**Decision:**

Accept (spotlight)

**Comment:**

Reviewers were unanimously enthusiastic about this quality of this study of neural iterate averaging (a smoothing tool that shows up usefully in many parts of deep learning, and is not fully understood for neural nets), deriving an exponential scaling rule via an analysis in the SDE limit, and performing an extensive empirical validation of this scaling. The authors addressed reviewers' questions comprehensively. The insights and results in this work are crisp, and have the potential for broad algorithmic impact; thus, I recommend a spotlight distinction.